# ADVERSARIAL COUNTERFACTUAL ENVIRONMENT MODEL LEARNING

## ABSTRACT

A good model for action-effect prediction, i.e., the environment model, is essential for sample-efficient policy learning, in which the agent can take numerous free trials to find good policies. Currently, the model is commonly learned by fitting historical transition data through empirical risk minimization (ERM). However, we discover that simple data fitting can lead to a model that will be totally wrong in guiding policy learning due to the selection bias in offline dataset collection. In this work, we introduce weighted empirical risk minimization (WERM) to handle this problem in model learning. A typical WERM method utilizes inverse propensity scores to re-weight the training data to approximate the target distribution. However, during the policy training, the data distributions of the candidate policies can be various and unknown. Thus, we propose an adversarial weighted empirical risk minimization (AWRM) objective that learns the model with respect to the worst case of the target distributions. We implement AWRM in a sequential decision structure, resulting in the GALILEO model learning algorithm. We also discover that GALILEO is closely related to adversarial model learning, explaining the empirical effectiveness of the latter. We apply GALILEO in synthetic tasks and verify that GALILEO makes accurate predictions on counterfactual data. We finally applied GALILEO in real-world offline policy learning tasks and found that GALILEO significantly improves policy performance in real-world testing.

## 1 INTRODUCTION

A good environment model is important for sample-efficient decision-making policy learning techniques like reinforcement learning (RL) (James & Johns, 2016). The agent can take trials with this model to find better policies, then the costly real-world trial-and-errors can be saved (James & Johns, 2016; Yu et al., 2020) or completely waived (Shi et al., 2019). In this process, the core of the models is to answer queries on counterfactual data unbiasedly, that is, given states, correctly answer *what might happen* if we were to carry out actions *unseen* in the training data (Levine et al., 2020).

Requiring counterfactual queries makes the environment model learning essentially different from standard supervised learning (SL) which directly fits the offline dataset. In real-world applications, the offline data is often collected with selection bias, that is, for each state, each action might be chosen unfairly. Seeing the example in Fig. 1(a), to keep the ball following a target line, a behavior policy will use a smaller force when the ball's location is closer to the target line. When a dataset is collected with selection bias, the association between the (location) states and (force) actions will make SL hard to identify the correct causal relationship of the states and actions to the next states respectively. Then when we query the model with counterfactual data, the predictions might be catastrophic failures. In Fig. 1(c), it mistakes that smaller forces will increase the ball's next location.

Generally speaking, the problem corresponds to a challenge of training the model in one dataset but testing in another dataset with a shifted distribution (i.e., the dataset generated by counterfactual queries), which is beyond the SL's capability as it violates the independent and identically distributed (*i.i.d.*) assumption. The problem is widely discussed in causal inference for individual treatment effects (ITEs) estimation in many scenarios like patients' treatment selection (Imbens, 1999; Alaa & van der Schaar, 2018). ITEs are the effects of treatments on individuals, which are measured by treating each individual under a uniform policy and evaluate the effect differences. Practical solutions use weighted empirical risk minimization (WERM) to handle this problem (Jung et al., 2020; Shimodaira, 2000; Hassanpour & Greiner, 2019). In particular, they estimate an inverse propensity score (IPS) to re-weight the training data to approximate the data distribution under a uniform policy. Then a model is trained under the reweighted data distribution. The distribution-shift problem is solved as ITEs estimation and model training are under the same distribution.

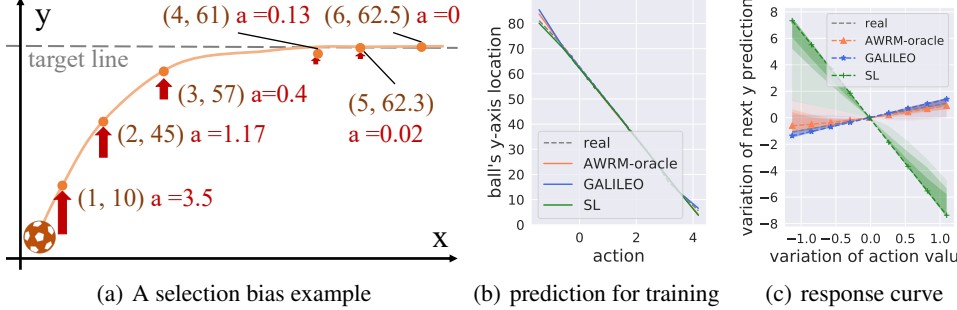

(a) A selection bias example    (b) prediction for training    (c) response curve

Figure 1: An example of selection bias and predictions under counterfactual queries. Subfigure (a) shows how the data is collected: a ball locates in a 2D plane whose position is $(x_t, y_t)$ at time $t$. The ball will move to $(x_{t+1}, y_{t+1})$ according to $x_{t+1} = x_t + 1$ and $y_{t+1} \sim \mathcal{N}(y_t + a_t, 2)$. Here, $a_t$ is chosen by a control policy $a_t \sim \mathcal{N}((\phi - y_t)/15, 0.05)$ parameterized by $\phi$, which tries to keep the ball near the line $y = \phi$. In Subfigure (a), $\phi$ is set to $62.5$. Subfigure (b) shows the collected training data (grey dashed line) and the two learned models' prediction of the next position of $y$. All the models discovered the relation that the corresponding next $y$ will be smaller with a larger action. **However, the truth is not because the larger $a_t$ causes a smaller $y_{t+1}$, but the policy selects a small $a_t$ when $y_t$ is close to the target line.** When we estimate the response curves by fixing $y_t$ and reassigning action $a_t$ with other actions $a_t + \Delta a$, where $\Delta a \in [-1, 1]$ is a variation of action value, the model of SL will exploit the association and give opposite responses, while in AWRM and its practical implementation GALILEO, the predictions are closer to the ground truths. The result is in Subfigure (c), where the darker a region is, the more samples are fallen in.

The selection bias can be regarded as an instance of the problem called "distributional shift" in offline model-based RL, which has also received great attention (Levine et al., 2020; Yu et al., 2020; Kidambi et al., 2020; Chen et al., 2021). However, previous methods, where *naive supervised learning* is used for environment model learning, ignore the problem in environment model learning    to Reviewer rQ79
but handling the problem by suppressing the policy exploration and learning in risky regions. Although these methods have made great progress in many tasks, so far, how to learn a better environment model that can alleviate the problem for faithful offline policy optimization has rarely been discussed.

In this work, for faithful offline policy optimization, we introduce WERM to environment model learning. The extra challenge of model learning for policy optimization is that we have to query numerous different policies' feedback besides the uniform policy for finding a good policy. Thus the target data distribution to reweight can be various and unknown. To solve the problem, we propose an objective called adversarial weighted empirical risk minimization (AWRM). AWRM introduces adversarial policies, of which the corresponding counterfactual dataset has the maximal prediction error of the model. For each iteration, the model is learned to be as small prediction risks as possible under the adversarial counterfactual dataset. However, the adversarial counterfactual dataset cannot be obtained in the offline setting, thus we derive an approximation of the counterfactual data distribution queried by the optimal adversarial policy and use a variational representation to give a tractable solution to learn a model from the approximated data distribution. As a result, we derive a practical approach named **G**enerative **A**dversarial off**LI**ne counterfactua**L E**nvironment m**O**del learning (GALILEO) for AWRM. Fig. 2 shows the difference in the prediction errors learned by these algorithms. We also discover that GALILEO is closely related to existing generative-adversarial model learning techniques, explaining the effectiveness of the latter.

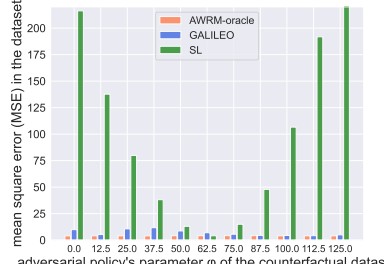

Figure 2: An illustration of the prediction error in counterfactual datasets. The prediction risks is measured with mean square error (MSE). The error of SL is small only in training data ($\phi = 62.5$) but becomes much larger in the dataset "far away from" the training data. AWRM-oracle selects the oracle worst counterfactual dataset for training for each iteration (pseudocode is in Alg. 1) which reaches small MSE in all datasets and gives correct response curves (Fig. 1(c)). GALILEO approximates the optimal adversarial counterfactual data distribution based on the training data and model. Although the MSE of GALILEO is a bit larger than SL in the training data, in the counterfactual datasets, the MSE is on the same scale as AWRM-oracle.

Experiments are conducted in two synthetic and two realistic environments. The results in the synthetic environments show that GALILEO can reconstruct correct responses for counterfactual queries. The evaluation results in two realistic environments also demonstrate that GALILEO has

better ability in counterfactual query compared with baselines. We finally search for a policy based on the learned model in a real-world online platform. The policy significantly improves performance in concerned business indicators.

## 2 RELATED WORK

We give related adversarial algorithms for model learning in the following and leave other related work in Appx. F. GANTIE (Yoon et al., 2018) uses a generator to fill counterfactual outcomes for each data pair and a discriminator to judge the source (treatment group or control group) of the filled data pair. The generator is trained to minimize the output of the discriminator. GANITE is trained until the discriminator cannot determine which of the components is the factual outcome. Bica et al. (2020) propose SCIGAN to extend GANITE to continuous treatment effect estimation (a.k.a., dosage-response estimation) via a hierarchical discriminator architecture. In real-world applications, environment model learning based on Generative Adversarial Imitation Learning (GAIL) has also been adopted for sequential decision-making problems (Ho & Ermon, 2016). GAIL is first proposed for policy imitation (Ho & Ermon, 2016), which uses the imitated policy to generate trajectories by interacting with the environment. The policy is learned with the trajectories through RL which maximizes the cumulative rewards given by the discriminator. Shi et al. (2019); Chen et al. (2019); Shang et al. (2019) use GAIL for environment model learning by regarding the environment model as the generator and the behavior policy as the "environment" in standard GAIL. These studies empirically demonstrate that adversarial model learning algorithms have better generalization ability for counterfactual queries, while our study reveals the connection between adversarial model learning and the WERM through IPS. Our derived practical algorithm GALILEO is closely related to the existing adversarial model learning algorithms, explaining the effectiveness of the latter.

## 3 PRELIMINARIES

### 3.1 SINGLE-STEP INDIVIDUALIZED TREATMENT EFFECTS ESTIMATION AND WEIGHTED EMPIRICAL RISKS MINIMIZATION

We first introduce individualized treatment effects (ITEs) estimation (Rosenbaum & Rubin, 1983), which can be regarded as the scenario in which the environment model has only a single step. ITEs are typically defined as $ITE(x) := \mathbb{E}[M^*(y|x,1)|A = 1, X = x] - \mathbb{E}[M^*(y|x,0)|A = 0, X = x]$, where $y$ is the feedback of the environment $M^*(y|x,a)$, $X$ denotes the state vector containing pre-treatment covariates (such as age and weight), $A$ denotes the treatment variable which is the action intervening to the state $X$, and $A$ should be sampled from a uniform policy. In the two-treatment scenario, $A$ is in $\{0, 1\}$ where 1 is the action to intervene and 0 is the action to do nothing. A correct ITEs estimation should be done in Randomized Controlled Trials (RCT) in which we have the same probability of samples of $A = 1$ and $A = 0$ for each $X$. Here we use lowercase $x, a$ and $y$ to denote samples of random variables $X, A$ and $Y$, and use $\mathcal{X}, \mathcal{A}$ and $\mathcal{Y}$ to denote space of the samples. In practice, we prefer to estimate ITEs under observational studies. In observational studies, datasets are pre-collected from the real world by a behavior policy such as a human-expert policy. In this case, a common approach for estimating the ITEs can be $\hat{ITE}(x_i) = a_i(y_i^F - M(x_i, 1 - a_i)) + (1 - a_i)(M(x_i, 1 - a_i) - y_i^F)$ (Shalit et al., 2017) in deterministic prediction, where $x_i$ and $y_i^F$ denote the covariate and factual feedback of the $i$-th sample, and $M \in \mathcal{M}$ denotes an approximated feedback model. $\mathcal{M}$ is the space of the model. In this formulation, the training set is an empirical factual data distribution $P_F = \{(x_i, a_i)\}_i^n$ and the testing set is an empirical counterfactual data distribution $P_{CF} = \{(x_i, 1 - a_i)\}_i^n$. If $a$ does not sample from a discrete uniform policy, i.e., the policy has selection bias, $P_F$ and $P_{CF}$ will be two different distributions, which violate the i.i.d. assumption of standard supervised learning. In stochastic prediction, $\hat{ITE}(x) = \mathbb{E}[M(y|x,1)] - \mathbb{E}[M(y|x,0)]$ and the counterfactual distribution for testing is the dataset with action sampling from a uniform policy.

Generally speaking, in ITEs estimation, the risks of queries under counterfactual data are caused by the gap between the policy in training and testing data distributions. Without further processing, minimizing the empirical risks cannot guarantee the counterfactual-query risks being minimized. Assuming that the policy in training data $\mu$ satisfies $\mu(a|x) > 0, \forall a \in \mathcal{A}, \forall x \in \mathcal{X}$ (often named overlap assumption), a classical solution to handle the above problem is weighted empirical risk minimization (WERM) through an inverse propensity scoring (IPS) term $\omega$ (Shimodaira, 2000; Assaad et al., 2021; Hassanpour & Greiner, 2019; Jung et al., 2020):

**Definition 3.1.** *The learning objective of WERM through IPS is formulated as*

$$\min_{M \in \mathcal{M}} L(M) = \min_{M \in \mathcal{M}} \mathbb{E}_{x,a,y \sim p^\mu_{M^*}} [\omega(x,a)\ell(M(y|x,a),y)],$$

$$s.t. \quad \omega(x,a) = \frac{\beta(a|x)}{\mu(a|x)}, \tag{1}$$

*where $\beta$ and $\mu$ denote the policies in testing and training domains, and $p^\mu_{M^*}$ is the joint probability $p^\mu_{M^*}(x,a,y) := \rho_0(x)\mu(a|x)M^*(y|x,a)$ in which $\rho_0(x)$ is the distribution of state. $\mathcal{M}$ is the model space. $\ell$ is a loss function for model learning.*

The $\omega$ is also known as importance sampling (IS) weight, which corrects the sampling bias. In this objective, $\omega$ is to align the training data distribution to the testing data. By selecting different $\hat{\omega}$ to approximate $\omega$ to learn the model $M$, current environment model learning algorithms for ITEs estimation are fallen into the framework. In standard supervised learning and some works for ITEs estimation (Wager & Athey, 2018; Weiss et al., 2015), $\hat{\omega}(x,a) = 1$ as the distribution-shift problem is ignored. In Shimodaira (2000); Assaad et al. (2021); Hassanpour & Greiner (2019), $\omega = \frac{1}{\hat{\mu}}$ (i.e., $\beta$ a uniform policy) for balancing treatment and control group, where $\hat{\mu}$ is an approximation of behavior policy $\mu$. Note that it is a reasonable weight in ITEs estimation: ITEs are defined to evaluate the effect of each state between treatment and control behavior under a uniform policy.

## 3.2 SEQUENTIAL DECISION-MAKING SETTING

Decision-making processes in a sequential environment are often formulated into Markov Decision Process (MDP) (Sutton & Barto, 1998). MDP depicts an agent interacting with the environment through actions. In the first step, states are sampled from an initial state distribution $x_0 \sim \rho_0(x)$. Then at each time-step $t \in \{0, 1, 2, ...\}$, the agent takes an action $a_t \in \mathcal{A}$ through a policy $\pi(a_t|x_t) \in \Pi$ based on the state $x_t \in \mathcal{X}$, then the agent receives a reward $r_t$ from a reward function $r(x_t, a_t) \in \mathbb{R}$ and transits to the next state $x_{t+1}$ given by a transition function $M^*(x_{t+1}|x_t, a_t)$ built in the environment. $\Pi$, $\mathcal{X}$, and $\mathcal{A}$ denote the policy, state, and action spaces.

## 4 METHOD

In this section, we first propose a new offline model-learning objective based on Def. 3.1 for policy optimization tasks in Sec. 4.1; In Sec. 4.2, we derive a tractable solution to the proposed objective; Finally, we give a practical implementation in Sec. 4.3.

## 4.1 PROBLEM FORMULATION

For offline policy optimization, we require the environment model to have generalization ability in counterfactual queries since we need to query numerous different policies' correct feedback from $M$. Referring to the formulation of WERM through IPS in Def. 3.1, policy optimization requires $M$ to minimize counterfactual-query risks under numerous unknown different policies rather than *a specific target policy $\beta$*. More specifically, the question is: If $\beta$ is unknown and can be varied, how should we reduce the risks in counterfactual queries? In this article, we call the model learning problem in this setting "counterfactual environment model learning" and propose a new objective to handle the problem. To be compatible with multi-step environment model learning, we first define a generalized WERM through IPS based on Def. 3.1.

**Definition 4.1.** *Given the MDP transition function $M^*$ that satisfies $M^*(x'|x,a) > 0, \forall x \in \mathcal{X}, \forall a \in \mathcal{A}, \forall x' \in \mathcal{X}$ and $\mu$ satisfies $\mu(a|x) > 0, \forall a \in \mathcal{A}, \forall x \in \mathcal{X}$, the learning objective of generalized WERM through IPS is formulated as*

$$\min_{M \in \mathcal{M}} L(M) = \min_{M \in \mathcal{M}} \mathbb{E}_{x,a,x' \sim \rho^\mu_{M^*}} [\omega(x,a,x')\ell_M(x,a,x')],$$

$$s.t. \quad \omega(x,a,x') = \frac{\rho^\beta_{M^*}(x,a,x')}{\rho^\mu_{M^*}(x,a,x')}, \tag{2}$$

*where $\rho^\mu_{M^*}$ is the training data distribution (collected by policy $\mu$), $\rho^\beta_{M^*}$ is the testing data distribution (collected by policy $\beta$). We define $\ell_M(x,a,x') := \ell(M(x'|x,a), x')$ for brevity.*

In an MDP, given any policy $\pi$, $\rho^\pi_{M^*}(x,a,x') = \rho^\pi_{M^*}(x)\pi(a|x)M^*(x'|x,a)$ where $\rho^\pi_{M^*}(x)$ denotes the occupancy measure of $x$ for policy $\pi$, which can be defined as $\rho^\pi_{M^*}(x) := (1-\gamma)\mathbb{E}_{x_0 \sim \rho_0}[\sum_{t=0}^{\infty} \gamma^t \Pr(x_t = x|x_0, M^*)]$ (Sutton & Barto, 1998; Ho & Ermon, 2016) where

$\Pr^{\pi}[x_t = x|x_0, M^*]$ is the state visitation probability that the policy $\pi$ visits $x$ at time-step $t$ by executing in the environment $M^*$ and starting at the state $x_0$, and $\gamma \in [0, 1]$ is the discount factor. Here we also define $\rho_{M^*}^{\pi}(x, a) := \rho_{M^*}^{\pi}(x)\pi(a|x)$ for simplicity. In this definition, $\omega$ can be rewritten as: $\omega(x, a, x') = \frac{\rho_{M^*}^{\beta}(x)\beta(a|x)M^*(x'|x,a)}{\rho_{M^*}^{\mu}(x)\mu(a|x)M^*(x'|x,a)} = \frac{\rho_{M^*}^{\beta}(x,a)}{\rho_{M^*}^{\mu}(x,a)}$. In single-step environments, for any policy $\pi$, $\rho_{M^*}^{\pi}(x) = \rho_0(x)$. Then we have $\omega(x, a, x') = \frac{\rho_0(x)\beta(a|x)}{\rho_0(x)\mu(a|x)} = \frac{\beta(a|x)}{\mu(a|x)}$, and the objective is degraded to Eq. 1. Therefore, Def. 3.1 is a special case of this generalized form.

Since $\beta$ for counterfactual queries is unknown and can be varied in policy optimization, to reduce the risks in counterfactual queries in this scenario, we introduce adversarial policies which can induce the worst performance of the model predictions and propose to optimize WERM under the adversarial policies. In particular, we propose **A**dversarial **W**eighted empirical **R**isk **M**inimization (AWRM) based on Def. 4.1 to handle this problem.

**Definition 4.2.** *Given the MDP transition function $M^*$, the learning objective of adversarial Weighted empirical risk minimization through IPS is formulated as*

$$\hat{M}^* = \min_{M \in \mathcal{M}} \max_{\beta \in \Pi} L(\rho_{M^*}^{\beta}, M) = \min_{M \in \mathcal{M}} \max_{\beta \in \Pi} \mathbb{E}_{x,a,x' \sim \rho_{M^*}^{\mu}}[\omega(x, a|\rho_{M^*}^{\beta})\ell_M(x, a, x')],$$

$$s.t. \quad \omega(x, a|\rho_{M^*}^{\beta}) = \frac{\rho_{M^*}^{\beta}(x, a)}{\rho_{M^*}^{\mu}(x, a)}, \tag{3}$$

*where the re-weighting term $\omega(x, a|\rho_{M^*}^{\beta})$ is conditioned on the data distribution $\rho_{M^*}^{\beta}$ of the adversarial policy $\beta$. In the following, we will ignore $\rho_{M^*}^{\beta}$ and use $\omega(x, a)$ for brevity.*

According to the definition of MDP, $\omega(x, a, x') = \omega(x, a)$ since the transition probability in the ratio can be canceled. Eq. 3 minimizes the maximum model loss under all counterfactual data distributions $\rho_{M^*}^{\beta}(x, a, x'), \beta \in \Pi$ to guarantee the generalization ability for counterfactual data queried by policies in $\Pi$.

## 4.2 TRACTABLE AWRM SOLUTION

In this section, we propose a tractable solution to optimize Eq. 3. The full derivations can be found in the Appx. A. We choose $\ell_M$ to be the negative log-likelihood loss and derive the solution to Eq. 3:

$$L(\rho_{M^*}^{\beta}, M) = \mathbb{E}_{x,a \sim \rho_{M^*}^{\mu}}\left[\omega(x, a|\rho_{M^*}^{\beta})\mathbb{E}_{M^*}(-\log M(x'|x, a))\right],$$

where $\mathbb{E}_{M^*}[\cdot]$ denotes $\mathbb{E}_{x' \sim M^*(x'|x,a)}[\cdot]$. *The core problem is how to construct the data distribution $\rho_{M^*}^{\beta^*}$ of the best-response policy $\beta^*$ in $M^*$ as it is costly to get extra data from $M^*$ in real-world applications.* Instead of deriving the optimal $\beta^*$, our solution is to offline estimate the optimal adversarial distribution $\rho_{M^*}^{\beta^*}$ with respect to $M$, then we can construct a surrogate objective to optimize $M$ without directly querying the real environment $M^*$.

### 4.2.1 OPTIMAL ADVERSARIAL DATA DISTRIBUTION APPROXIMATION

Ideally, given any $M$, it is obvious that the optimal $\beta$ is the one that makes $\rho_{M^*}^{\beta}(x, a)$ assign all densities to the point that has the largest negative log-likelihood. However, searching for the maximum is impractical, especially in continuous space. To give a relaxed but tractable solution, we add an $L_2$ regularizer to the original objective Eq. 3:

$$\min_{M \in \mathcal{M}} \max_{\beta \in \Pi} \bar{L}(\rho_{M^*}^{\beta}, M) = \min_{M \in \mathcal{M}} \max_{\beta \in \Pi} \mathbb{E}_{x,a \sim \rho_{M^*}^{\mu}}\left[\omega(x, a)\mathbb{E}_{M^*}[-\log M(x'|x, a)]\right] - \frac{\alpha}{2}\|\rho_{M^*}^{\beta}(\cdot, \cdot)\|_2^2,$$

$$\tag{4}$$

where $\alpha$ denotes the regularization coefficient of $\rho_{M^*}^{\beta}$ and $\|\rho_{M^*}^{\beta}(\cdot, \cdot)\|_2^2 = \int_{\mathcal{X},\mathcal{A}}(\rho_{M^*}^{\beta}(x, a))^2 \mathrm{d}a\mathrm{d}x$.

Then we can approximate the optimal distribution $\rho_{M^*}^{\bar{\beta}^*}$ via Lemma. 4.3.

**Lemma 4.3.** *Given any $M$ in $\bar{L}(\rho_{M^*}^{\beta}, M)$, the distribution of the ideal best-response policy $\bar{\beta}^*$ satisfies:*

$$\rho_{M^*}^{\bar{\beta}^*}(x, a) = \frac{1}{\alpha_M}(D_{KL}(M^*(\cdot|x, a), M(\cdot|x, a)) + H_{M^*}(x, a)), \tag{5}$$

*where $D_{KL}(M^*(\cdot|x, a), M(\cdot|x, a))$ is the Kullback-Leibler (KL) divergence between $M^*(\cdot|x, a)$ and $M(\cdot|x, a)$, $H_{M^*}(x, a)$ denotes the entropy of $M^*(\cdot|x, a)$, and $\alpha_M$ is the regularization coefficient $\alpha$ in Eq. 4 and also as a normalizer of Eq. 5.*

Intuitively, $\rho_{M^*}^{\bar{\beta}^*}$ has larger densities on the data where the divergence between the approximation model and the real model (i.e., $D_{KL}(M^*(\cdot|x,a), M(\cdot|x,a))$) is larger or the stochasticity of the real model (i.e., $H_{M^*}$) is larger. However, the integral process of $D_{KL}$ in Eq. 5 is intractable in the offline setting as it explicitly requires the conditional probability function of $M^*$. Our solution to solve the problem is utilizing the offline dataset $\mathcal{D}_{\text{real}}$ as the empirical *joint* distribution $\rho_{M^*}^{\mu}(x,a,x')$ and adopting practical techniques for distance estimation on two joint distributions, like GAN (Goodfellow et al., 2014; Nowozin et al., 2016), to approximate Eq. 5. To adopt that solution, we should first transform Eq. 5 into a form under joint distributions. Without loss of generality, we introduce an intermediary policy $\kappa$, of which $\mu$ can be regarded as a specific instance. Then we have $M(x'|x,a) = \rho_M^{\kappa}(x,a,x')/\rho_M^{\kappa}(x,a)$ for any $M$ if $\rho_M^{\kappa}(x,a) > 0$. Assuming $\forall x \in \mathcal{X}, \forall a \in \mathcal{A}, \rho_{M^*}^{\kappa}(x,a) > 0$ if $\rho_{M^*}^{\bar{\beta}^*}(x,a) > 0$, which will hold when $\kappa$ overlaps with $\mu$, then Eq. 5 can transform to:

to Reviewer bcJS

$$\frac{1}{\alpha_0(x,a)} \left( \int_{\mathcal{X}} \rho_{M^*}^{\kappa}(x,a,x') \log \frac{\rho_{M^*}^{\kappa}(x,a,x')}{\rho_M^{\kappa}(x,a,x')} \mathrm{d}x' - \rho_{M^*}^{\kappa}(x,a) \left( \log \frac{\rho_{M^*}^{\kappa}(x,a)}{\rho_M^{\kappa}(x,a)} - H_{M^*}(x,a) \right) \right),$$

where $\alpha_0(x,a) = \alpha_M \rho_{M^*}^{\kappa}(x,a)$. We notice that the form $\rho_{M^*}^{\kappa} \log \frac{\rho_{M^*}^{\kappa}}{\rho_M^{\kappa}}$ is the integrated function in reverse KL divergence, which is an instance of $f$ function in $f$-divergence (Ali & Silvey, 1966). Replacing that form with $f$ function, we obtain a generalized representation of $\rho_{M^*}^{\bar{\beta}^*}$:

$$\bar{\rho}_{M^*}^{\bar{\beta}^*} := \frac{1}{\alpha_0(x,a)} \left( \int_{\mathcal{X}} \rho_{M^*}^{\kappa}(x,a,x') f\left( \frac{\rho_M^{\kappa}(x,a,x')}{\rho_{M^*}^{\kappa}(x,a,x')} \right) \mathrm{d}x' - \rho_{M^*}^{\kappa}(x,a) \left( f\left( \frac{\rho_M^{\kappa}(x,a)}{\rho_{M^*}^{\kappa}(x,a)} \right) - H_{M^*}(x,a) \right) \right), \tag{6}$$

where $f : \mathbb{R}_+ \to \mathbb{R}$ is a convex and lower semi-continuous (l.s.c.) function. $\bar{\rho}_{M^*}^{\bar{\beta}^*}$ gives a generalized representation of the optimal adversarial data distribution to maximize the error of the model. Based on Eq. 6, we have a surrogate objective of AWRM which can avoid querying $M^*$ to construct $\rho_{M^*}^{\beta^*}$:

**Theorem 4.4.** *Let $\bar{\rho}_{M^*}^{\bar{\beta}^*}$ as the data distribution of the best-response policy $\bar{\beta}^*$ in Eq. 4 under model $M_\theta$ parameterized by $\theta$, then we can find the optimal $\theta^*$ of $\min_\theta \max_{\beta \in \Pi} \bar{L}(\rho_{M^*}^{\beta}, M_\theta)$ (Eq. 4) via iteratively optimizing the objective $\theta_{t+1} = \min_\theta \bar{L}(\bar{\rho}_{M^*}^{\bar{\beta}^*}, M_\theta)$, where $\bar{\rho}_{M^*}^{\bar{\beta}^*}$ is approximated via the last-iteration model $M_{\theta_t}$. Based on Corollary A.7, we derive an upper bound objective for $\min_\theta \bar{L}(\bar{\rho}_{M^*}^{\bar{\beta}^*}, M_\theta)$:*

$$\theta_{t+1} = \min_\theta \mathbb{E}_{\rho_{M^*}^{\kappa}} \left[ \frac{-1}{\alpha_0(x,a)} \log M_\theta(x'|x,a) \underbrace{\left( f\left( \frac{\rho_{M_{\theta_t}}^{\kappa}(x,a,x')}{\rho_{M^*}^{\kappa}(x,a,x')} \right) - f\left( \frac{\rho_{M_{\theta_t}}^{\kappa}(x,a)}{\rho_{M^*}^{\kappa}(x,a)} \right) + H_{M^*}(x,a) \right)}_{W(x,a,x')} \right],$$

*where $\mathbb{E}_{\rho_{M^*}^{\kappa}}[\cdot]$ denotes $\mathbb{E}_{x,a,x' \sim \rho_{M^*}^{\kappa}}[\cdot]$, $f$ is a l.s.c function satisfying $f'(x) \leq 0, \forall x \in \mathcal{X}$, and $\alpha_0(x,a) = \alpha_{M_{\theta_t}} \rho_{M^*}^{\kappa}(x,a)$.*

Thm. 4.4 approximately achieve AWRM by using $\kappa$ and a pseudo-reweighting module $W$. $W$ assigns learning propensities for data points with larger differences between distributions $\rho_{M_{\theta_t}}^{\kappa}$ and $\rho_{M^*}^{\kappa}$. By adjusting the weights, the learning process will exploit subtle errors in any data point, whatever how many proportions it contributes, to correct potential generalization errors on counterfactual data.

### 4.2.2 TRACTABLE SOLUTION TO THM. 4.4

In Thm. 4.4, the term $f\left( \frac{\rho_{M_{\theta_t}}^{\kappa}(x,a,x')}{\rho_{M^*}^{\kappa}(x,a,x')} \right) - f\left( \frac{\rho_{M_{\theta_t}}^{\kappa}(x,a)}{\rho_{M^*}^{\kappa}(x,a)} \right)$ is still intractable. To solve the problem, first, we resort to the first-order approximation of $f$. Given some $u \in (1 - \xi, 1 + \xi), \xi > 0$, we have

$$f(u) \approx f(1) + f'(u)(u - 1), \tag{7}$$

where $f'$ is the first-order derivative of $f$. By Taylor's formula and the fact that $f'(u)$ of the generator function $f$ is bounded in $(1 - \xi, 1 + \xi)$, the approximation error is no more than $\mathcal{O}(\xi^2)$. Let $u = \frac{p(x)}{q(x)}$ in Eq. 7, the pattern $f(\frac{p(x)}{q(x)})$ in Thm. 4.4 can be converted to $f'(\frac{p(x)}{q(x)})(\frac{p(x)}{q(x)} - 1) + f(1)$. $\frac{p(x)}{q(x)}$ can be approximated by sampling from datasets and $f'(\frac{p(x)}{q(x)})$ can be approximated by the corresponding variational representation $T_{\varphi^*}$ according to Lemma A.9 (Nowozin et al., 2016) (see

Appendix for details). Based on any specific $f$-function, we can represent $T_\varphi$ with a discriminator $D_\varphi$. However, Thm. 4.4 holds only when $\forall x \in \mathcal{X}, f'(x) \le 0$. It can be verified that the instance $f(u) = u \log u - (u+1) \log(u+1)$, $T_\varphi(u) = \log D_\varphi(u)$ satisfy this condition (see Tab. 2 in the Appendix). We select that instance for the tractable solution. In summary, by computing $f(\cdot)$ with first-order approximation (Eq. 7) and leveraging the variational representation $T_{\varphi^*} = \log D_{\varphi^*}$ to approximate $f'(\frac{\rho^\kappa_{M_{\theta_t}}}{\rho^\kappa_{M^*}})$, we can optimize the surrogate objective in Thm. 4.4 via:

$$\theta_{t+1} = \max_\theta \; \mathbb{E}_{\rho^\kappa_{M_{\theta_t}}} \left[ A_{\varphi_0^*, \varphi_1^*}(x, a, x') \log M_\theta(x'|x, a) \right] + \mathbb{E}_{\rho^\kappa_{M^*}} \left[ (H_{M^*}(x, a) - A_{\varphi_0^*, \varphi_1^*}(x, a, x')) \log M_\theta(x'|x, a) \right]$$

$$s.t. \quad \varphi_0^* = \arg\max_{\varphi_0} \mathbb{E}_{\rho^\kappa_{M^*}} \left[ \log D_{\varphi_0}(x, a, x') \right] + \mathbb{E}_{\rho^\kappa_{M_{\theta_t}}} \left[ \log(1 - D_{\varphi_0}(x, a, x')) \right] \tag{8}$$

$$\varphi_1^* = \arg\max_{\varphi_1} \mathbb{E}_{\rho^\kappa_{M^*}} \left[ \log D_{\varphi_1}(x, a) \right] + \mathbb{E}_{\rho^\kappa_{M_{\theta_t}}} \left[ \log(1 - D_{\varphi_1}(x, a)) \right],$$

where $A_{\varphi_0^*, \varphi_1^*}(x, a, x') = \log D_{\varphi_0^*}(x, a, x') - \log D_{\varphi_1^*}(x, a)$, $\mathbb{E}_{\rho^\kappa_M}[\cdot]$ is a simplification of $\mathbb{E}_{x, a, x' \sim \rho^\kappa_M}[\cdot]$ and $\varphi_0$ and $\varphi_1$ are the parameters of $T_{\varphi_0}$ and $T_{\varphi_1}$. Based on Lemma A.9, we have $f'(\frac{\rho^\kappa_{M_{\theta_t}}(x, a, x')}{\rho^\kappa_{M^*}(x, a, x')}) \approx \log D_{\varphi_0^*}(x, a, x')$ and $f'(\frac{\rho^\kappa_{M_{\theta_t}}(x, a)}{\rho^\kappa_{M^*}(x, a)}) \approx \log D_{\varphi_1^*}(x, a)$. Note that in the process, we ignore the term $\alpha_0(x, a)$ for simplifying the objective. The discussion on the impacts of removing $\alpha_0(x, a)$ is left in App. B. In Eq. 8, the pseudo-reweighting module $W$ in Thm. 4.4 is split into two terms in the RHS of the equation. The first term is a generative adversarial training objective regarding $M_\theta$ as the generator, while the second term is WERM through $H_{M^*} - A_{\varphi_0^*, \varphi_1^*}$.

### 4.3 PRACTICAL IMPLEMENTATION

To give a practical implementation of the solution, extra two assumptions introduced in the process of modeling should be handled:

**First**, the approximation of Eq. 7 holds only when $\frac{p(x)}{q(x)}$ is close to 1, which might not be satisfied. To handle the problem, we inject a naive supervised learning loss and replace the second term of the objective Eq. 8 when the expectation of $D_{\varphi_0}$ in $\rho^\kappa_{M_{\theta_t}}$ is far away from $0.5$ ($f'(1) = \log 0.5$);

**Second**, the overlap assumption of $\kappa$. In practice, we need to use the real-world data to construct the distribution $\rho^\kappa_{M^*}$ and the generative data to construct $\rho^\kappa_{M_{\theta_t}}$. In the offline model-learning setting, we only have a real-world dataset $\mathcal{D}$ collected by the behavior policy $\mu$. We can learn a policy $\hat\mu \approx \mu$ via behavior cloning based on $\mathcal{D}$ (Pomerleau, 1991; Ho & Ermon, 2016) and let $\hat\mu$ be the policy $\kappa$. Then we can regard $\mathcal{D}$ as the empirical data distribution of $\rho^\kappa_{M^*}$ and the trajectories collected by $\hat\mu$ in the model $M_{\theta_t}$ as the empirical data distribution of $\rho^\kappa_{M_{\theta_t}}$. But the assumption $\forall x \in \mathcal{X}, \forall a \in \mathcal{A}, \mu(a|x) > 0$ might not be satisfied. To handle the problem, in behavior cloning, we model $\hat\mu$ with a Gaussian distribution and constrain the lower bound of the variance with a small value $\epsilon_\mu > 0$ to keep the assumption holding. Besides, we add small Gaussian noises $\mathcal{N}(0, \epsilon_D)$ to the inputs of $D_\varphi$ to handle the mismatch between $\rho^\mu_{M^*}$ and $\rho^{\hat\mu}_{M^*}$ due to $\epsilon_\mu$.

to Reviewer bcJS

In Eq. 8, $H_{M^*}$ is unknown in advance. In practice, we use $H_{M_\theta}$ to estimate it. More specifically, the neural network of $M_\theta$ can be modeled with a Gaussian distribution. The variance of the Gaussian distribution is modeled with global variables $\Sigma$ for each dimension of output. We estimate $H_{M^*}$ with the closed-form solution of Gaussian entropy through $\Sigma$.

to Reviewer bcJS

Based on the above techniques, we propose **G**enerative **A**dversarial off**LI**ne counterfactua**L** **E**nvironment m**O**del learning (GALILEO) for environment model learning. GALILEO can be adopted in both single-step and sequential environment model learning. The detailed implementation and comparison to previous adversarial methods are in Appx. E.

## 5 EXPERIMENTS

In this section, we first conduct experiments in synthetic environments (Bica et al., 2020) to verify GALILEO on counterfactual queries [1] and the compatibility of GALILEO in sequential and single-step environments. We select Mean Integrated Square Error MISE $= \mathbb{E}\left[ \int_\mathcal{A} (M^*(y|x, a) - M(y|x, a))^2 \, da \right]$ as the metric, which is a commonly used metric to measure the accuracy in counterfactual queries by considering the prediction errors in the whole action space. Then we analyze the benefits of the implementation techniques described in Sec. 4.3 and

---

[1] the code will be released after the paper is published.

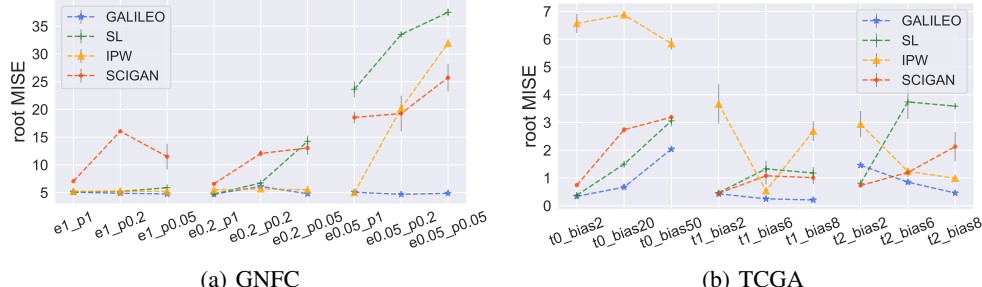

(a) GNFC        (b) TCGA

Figure 3: Illustration of the performance in GNFC and TCGA. The grey bar denotes the standard error ($\times 0.3$ for brevity) of 3 random seeds.

the problems without them. Finally, we deploy GALILEO in two complex environments: MuJoCo in Gym (Todorov et al., 2012) and a real-world food-delivery platform to test the performance of GALILEO in difficult tasks. The algorithms compared are: (1) **Supervised Learning (SL)**: using standard empirical risk minimization for model learning; (2) **Inverse Propensity Weighting (IPW)** (Spirtes, 2010): a standard implementation of WERM based IPS; (3) **eStimating the effects of Continuous Interventions using GANs (SCIGAN)** (Bica et al., 2020): an adversarial algorithms for model learning used for causal effect estimation, which can be roughly regarded as a partial implementation of GALILEO (Refer to Appx. E.2). We give a detailed description in Appx. G.2.     to Reviewer rQ79

### 5.1 TEST IN SYNTHETIC ENVIRONMENTS

**Test in sequential environments** Since there does not exist a task specifically designed for sequential environment model learning with selection bias, we construct a task, General Negative Feedback Control (GNFC), which can represent a classic type of task with policies having selection bias. Fig. 1(a) is also an example of GNFC. We give detailed motivation, the effect of selection bias, and other details in Appx. G.1.1. We construct tasks on GNFC by adding behavior policies $\mu$ with different scales of uniform noise $U(-e, e)$ with different probabilities $p$. In particular, with $e \in \{1.0, 0.2, 0.05\}$ and $p \in \{1.0, 0.2, 0.05\}$, we construct 9 tasks and name them with the format of "e*_p*". For example, e1_p0.2 is the task with behavior policy injecting with $U(-1, 1)$ with 0.2 probability. The results of GNFC tasks are summarized in Fig. 3(a) and the detailed results can be found in Tab. 8. The results show that the property of the behavior policy (i.e., $e$ and $p$) dominates the generalization ability of the baseline algorithms. When $e = 0.05$, almost all of the baselines fail and give a completely opposite response curve (see Fig. 4(a) and Appx. H.2). IPW still perform well when $0.2 \le e \le 1.0$ but fails when $e = 0.05, p <= 0.2$. We also found that SCIGAN can reach a better performance than other baselines when $e = 0.05, p <= 0.2$, but the results in other tasks are unstable. GALILEO is the only algorithm that is robust to the selection bias and outputs correct response curves in all of the tasks. Based on the experiment, we also indicate that the commonly used overlap assumption is unreasonable to a certain extent especially in real-world applications since it is impractical to inject noises into the whole action space. The problem of overlap assumption being violated should be taken into consideration otherwise the algorithm will be hard to use in practice if it is sensitive to the noise range.

**Test in single-step environments** Previous experiments on counterfactual environment model learning are based on single-step semi-synthetic data simulation (Bica et al., 2020). Since GALILEO is compatible with single-step environment model learning, we select the same task named TCGA in Bica et al. (2020) to test GALILEO. Based on three synthetic response functions in TCGA, we construct 9 tasks by choosing different parameters of selection bias on $\mu$ which is constructed with beta distribution, and design a coefficient $c$ to control the selection bias of the beta distribution. We name the tasks with the format of

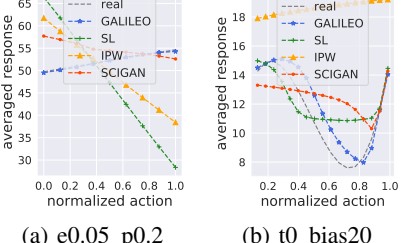

(a) e0.05_p0.2     (b) t0_bias20

Figure 4: Illustration of the averaged response curves.

"t?_bias?". For example, t1_bias2 is the task with the first response functions and $c = 2$. The detail of TCGA is in Appx. G.1.2. The results of TCGA tasks are summarized in Fig. 3(b) and the detailed results can be found in Tab. 9 in Appendix. We found the phenomenon in this experiment is similar to the one in GNFC, which demonstrates the compatibility of GALILEO to single-step environments. We also found that the results of IPW are unstable in this experiment. It might be because the behavior policy is modeled with beta distribution while the propensity score $\hat{\mu}$ is modeled with Gaussian distribution. Since IPW directly reweight loss function with $\frac{1}{\hat{\mu}}$, the results are sensitive

Table 1: Results of policy performance directly optimized through standard SAC (Haarnoja et al., 2018) using the learned dynamics models and deployed in MuJoCo environments. MAX-RETURN is the policy performance of SAC in the MuJoCo environments, and "avg. norm." is the averaged normalized return of the policies in the 9 tasks, where the returns are normalized to lie between 0 and 100, where a score of 0 corresponds to the worst policy, and 100 corresponds to MAX-RETURN.

| Task | Hopper | | | Walker2d | | | HalfCheetah | | | avg. norm. |
|---|---|---|---|---|---|---|---|---|---|---|
| Horizon | H=10 | H=20 | H=40 | H=10 | H=20 | H=40 | H=10 | H=20 | H=40 | / |
| GALILEO | $13.0 \pm 0.1$ | $33.2 \pm 0.1$ | $53.5 \pm 1.2$ | $11.7 \pm 0.2$ | $29.9 \pm 0.3$ | $61.2 \pm 3.4$ | $0.7 \pm 0.2$ | $-1.1 \pm 0.2$ | $-14.2 \pm 1.4$ | **51.1** |
| SL | $4.8 \pm 0.5$ | $3.0 \pm 0.2$ | $4.6 \pm 0.2$ | $10.7 \pm 0.2$ | $20.1 \pm 0.8$ | $37.5 \pm 6.7$ | $0.4 \pm 0.5$ | $-1.1 \pm 0.6$ | $-13.2 \pm 0.3$ | 21.1 |
| IPW | $5.9 \pm 0.7$ | $4.1 \pm 0.6$ | $5.9 \pm 0.2$ | $4.7 \pm 1.1$ | $2.8 \pm 3.9$ | $14.5 \pm 1.4$ | $\mathbf{1.6 \pm 0.2}$ | $\mathbf{0.5 \pm 0.8}$ | $\mathbf{-11.3 \pm 0.9}$ | 19.7 |
| SCIGAN | $12.7 \pm 0.1$ | $29.2 \pm 0.6$ | $46.2 \pm 5.2$ | $8.4 \pm 0.5$ | $9.1 \pm 1.7$ | $1.0 \pm 5.8$ | $1.2 \pm 0.3$ | $-0.3 \pm 1.0$ | $-11.4 \pm 0.3$ | 41.8 |
| MAX-RETURN | $13.2 \pm 0.0$ | $33.3 \pm 0.2$ | $71.0 \pm 0.5$ | $14.9 \pm 1.3$ | $60.7 \pm 11.1$ | $221.1 \pm 8.9$ | $2.6 \pm 0.1$ | $13.3 \pm 1.1$ | $49.1 \pm 2.3$ | 100.0 |

to the error on $\hat{\mu}$. GALILEO also models $\hat{\mu}$ with Gaussian distribution but the results are more stable since GALILEO does not re-weight through $\hat{\mu}$ explicitly.

**Response curve visualization** We plot the averaged response curves which are constructed by equidistantly sampling action from the action space and averaging the feedback of the states in the dataset as the averaged response. Parts of the results in Fig. 4 (all curves can be seen in Appx. H.2). For those tasks where baselines fail in reconstructing response curves, GALILEO not only reaches a better MISE score but reconstructs almost exact responses.

**Ablation studies** In Sec. 4.3, we introduce several techniques to develop a practical GALILEO algorithm. Based on task e0.2_p0.05 of GNFC, we give the ablation studies to investigate the effects of these techniques. As the main-body space is limited, we leave the results in Appx. H.3.

## 5.2 TEST IN COMPLEX ENVIRONMENTS

**In MuJoCo tasks** MuJoCo is a benchmark task in Gym (Todorov et al., 2012; Brockman et al., 2016) where we need to control a robot with specific dynamics to complete some tasks (e.g., standing or running). We select 3 environment from D4RL (Fu et al., 2020) to construct our model learning tasks. We compare it with a standard transition model learning algorithm used in the previous offline model-based RL algorithms (Yu et al., 2020; Kidambi et al., 2020), which is a variant of supervised learning. We name the method OFF-SL. Besides, we also implement IPW and SCIGAN as the baselines. In D4RL benchmark, only the "medium" tasks is collected with a fixed policy, i.e., the behavior policy is with 1/3 performance to the expert policy), which is most matching to our proposed problem. So we train models in datasets HalfCheetah-medium, Walker2d-medium, and Hopper-medium. We trained the models with the same gradient steps and saved the models.

to Reviewer bcJS, j5p5, and rQ79

We first verify the generalization ability of the models by adopting them into offline model-based RL. Instead of designing sophisticated tricks to suppress policy exploration and learning in risky regions as current offline model-based RL algorithms (Yu et al., 2020; Kidambi et al., 2020) do, we just use the standard SAC algorithm Haarnoja et al. (2018) to exploit the models for policy learning to strictly verify the ability of the models. Unfortunately, we found that the compounding error will still be inevitably large in the 1,000-step rollout, which is the standard horizon in MuJoCo tasks, leading all models to fail to derive a reasonable policy. To better verify the effects of models on policy optimization, we learn and evaluate the policies with three smaller horizons: $H \in \{10, 20, 40\}$.

The results are listed in Tab. 1. We first averaged the normalized return (refer to "avg. norm.") under each task, and we can see that the policy obtained by GALILEO is significantly higher than other models (the improvements are 24% to 161%). At the same time, we found that SCIGAN performed better in policy learning, while IPW performed similarly to SL. This is in line with our expectations, since IPW only considers the uniform policy as the target policy for debiasing, while policy optimization requires querying a wide variety of policies. Minimizing the prediction risks only under a uniform policy cannot yield a good environment model for policy optimization. Besides, in IPW, the cumulative effects of policy on the state distribution are ignored. On the other hand, SCIGAN, as a partial implementation of GALILEO (refer to Appx. E.2), also roughly achieves AWRM and considers the cumulative effects of policy on the state distribution, so its overall performance is better; In addition, we find that GALILEO achieves significant improvement in 6 of the 9 tasks. But in HalfCheetah, IPW works slightly better. However, compared with MAX-RETURN, it can be found that all methods fail to derive reasonable policies because their policies' performances are far away from the optimal policy. By further visualizing the trajectories, we found that all the learned policies just keep the cheetah standing in the same place or even going backward. This phenomenon is also similar to the results in MOPO (Yu et al., 2020). In MOPO's experiment in the medium datasets, the truncated-rollout horizon used in Walker and Hopper for policy training is set to 5, while HalfCheetah has to be set to *the minimal value: 1*. These phenomena indicate that HalfCheetah may still have unknown problems, resulting in the generalization bottleneck of

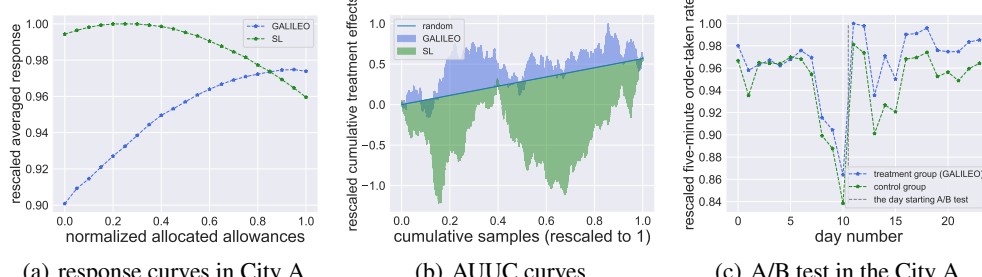

| (a) response curves in City A | (b) AUUC curves | (c) A/B test in the City A |

Figure 5: An illustration of the performance in BAT tasks. In Fig. 5(a) demonstrate the averaged response curves of the SL and GALILEO model in City A. It is expected to be monotonically increasing through our prior knowledge. In Fig. 5(b), the model with larger areas above the "random" line makes better predictions in randomized-controlled-trials data (Betlei et al., 2020). Fig. 5(c) shows the daily responses in the A/B test in City A. The complete results are in Appx. H.6.

the models. Besides, we also test the prediction error of the learned model in corresponding unseen "expert" and "medium-replay" datasets. The detailed results are in Appx. H.5.

**In a real-world platform** We finally deploy GALILEO in a real-world large-scale food-delivery platform. The goal of the platform is to balance the demand from take-out food orders and the supply of delivery clerks, i.e., helping delivery clerks fulfill more orders by giving reasonable strategies. We focus on a Budget Allocation task to the Time period (BAT) in the platform (see Appx. G.1.3 for details). The goal of the BAT task is to handle the imbalance problem between the demanded orders from customers and the supply of delivery clerks in different time periods by allocating reasonable allowances to those time periods. The core challenge of the environment model learning in BAT tasks is similar to the challenge in Fig. 1. Specifically, the behavior policy in BAT tasks is a human-expert policy, which tends to increase the budget of allowance in the time periods with a lower supply of delivery clerks, otherwise tends to decrease the budget (Fig. 12 gives a real-data instance of this phenomenon).

We first learn a model to predict the supply of delivery clerks (measured by fulfilled order amount) on given allowances. Although the SL model can efficiently fit the offline data, the tendency of the response curve is easily to be incorrect. As can be seen in Fig. 5(a), with a larger budget of allowance, the prediction of the supply is decreased in SL, which obviously goes against our prior knowledge. This is because, in the offline dataset, the corresponding supply will be smaller when the allowance is larger. It is conceivable that if we learn a policy through the model of SL, the optimal solution is canceling all of the allowances, which is obviously incorrect in practice. On the other hand, the tendency of GALILEO's response is correct. Fig. 13 plots all the results in 6 cities.

Second, we conduct randomized controlled trials (RCT) in one of the testing cities. Using the RCT samples, we can evaluate the generalization ability of the model predictions via Area Under the Uplift Curve (AUUC) (Betlei et al., 2020), which measure the correctness of the sort order of the model prediction in RCT samples. The AUUC further show that GALILEO gives a reasonable sort order on the supply prediction (see Fig. 5(b)) while the standard SL technique fails to complete this task.

Finally, we search for the optimal policy via the cross-entropy method planner (Hafner et al., 2019) based on the learned model and deploy the policy in a real-world platform. The results of A/B test in City A is shown in Fig. 5(c). It can be seen that after the day of the A/B test, the treatment group (deploying our policy) significant improve the five-minute order-taken rate than the baseline policy (the same as the behavior policy). In summary, *the policy improves the supply from 0.14 to 1.63 percentage points to the behavior policies in the 6 cities*. The details of these results are in Appx. H.6.

## 6 DISCUSSION AND FUTURE WORK

In this work, we propose AWRM which handles the generalization challenges of the counterfactual environment model learning. By theoretical modeling, we give a tractable solution to handle AWRM and propose GALILEO. GALILEO is verified in synthetic environments, complex robot control tasks, and a real-world platform, and shows great generalization ability on counterfactual queries.

Giving correct answers to counterfactual queries is important for policy learning. We hope the work can inspire researchers to develop more powerful tools for counterfactual environment model learning. The current limitation lies in: There are several simplifications in the theoretical modeling process (further discussion is in Appx. B), which can be modeled more elaborately . Besides, experiments on MuJoCo indicate that these tasks are still challenging to give correct predictions on counterfactual data. These should also be further investigated in future work.

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

# Appendix

## Table of Contents

## A   PROOF OF THEORETICAL RESULTS

The overall pipeline to model the tractable solution to AWRM is given in Fig. 6.

In the proof section, we replace the notation of $\mathbb{E}$ with an integral for brevity. Now we rewrite the original objective $\bar{L}(\rho_{M^*}^\beta, M)$ as:

$$\min_{M \in \mathcal{M}} \max_{\beta \in \Pi} \int_{\mathcal{X},\mathcal{A}} \rho_{M^*}^\mu(x,a)\omega(x,a) \int_{\mathcal{X}} M^*(x'|x,a)\left(-\log M(x'|x,a)\right) \mathrm{d}x'\mathrm{d}a\mathrm{d}x - \frac{\alpha}{2}\|\rho_{M^*}^\beta(\cdot,\cdot)\|_2^2,$$

(9)

where $\omega(x,a) = \frac{\rho_{M^*}^\beta(x,a)}{\rho_{M^*}^\mu(x,a)}$ and $\|\rho_{M^*}^\beta(\cdot,\cdot)\|_2^2 = \int_{\mathcal{X},\mathcal{A}} \rho_{M^*}^\beta(x,a)^2 \mathrm{d}a\mathrm{d}x$, which is the squared $l_2$-norm. In an MDP, given any policy $\pi$, $\rho_{M^*}^\pi(x,a,x') = \rho_{M^*}^\pi(x)\pi(a|x)M^*(x'|x,a)$ where $\rho_{M^*}^\pi(x)$ denotes the occupancy measure of $x$ for policy $\pi$, which can be defined (Sutton & Barto, 1998; Ho & Ermon, 2016) as $\rho_{M^*}^\pi(x) := (1-\gamma)\mathbb{E}_{x_0 \sim \rho_0}\left[\sum_{t=0}^\infty \gamma^t \Pr(x_t = x|x_0, M^*)\right]$ where $\Pr^\pi[x_t = x|x_0, M^*]$ is the state visitation probability that $\pi$ starts at state $x_0$ in model $M^*$ and receive $x$ at timestep $t$ and $\gamma \in [0,1]$ is the discount factor.

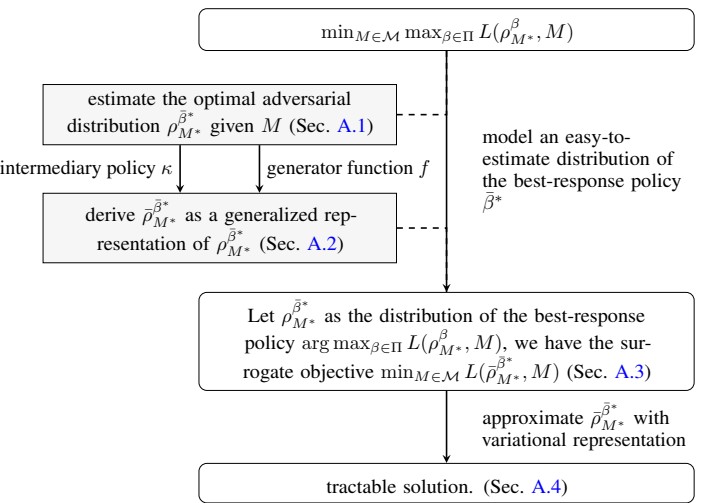

Figure 6: The overall pipeline to model the tractable solution to AWRM. $f$ is a generator function defined by $f$-divergence (Nowozin et al., 2016). $\kappa$ is an intermediary policy introduced in the estimation.

## A.1 Proof of Lemma 4.3

For better readability, we first rewrite Lemma 4.3 as follows:

**Lemma A.1.** *Given any $M$ in $\bar{L}(\rho_{M^*}^{\beta}, M)$, the distribution of the ideal best-response policy $\bar{\beta}^*$ satisfies:*

$$\rho_{M^*}^{\bar{\beta}^*}(x,a) = \frac{1}{\alpha_M}(D_{KL}(M^*(\cdot|x,a), M(\cdot|x,a)) + H_{M^*}(x,a)), \tag{10}$$

*where $D_{KL}(M^*(\cdot|x,a), M(\cdot|x,a))$ is the Kullback-Leibler (KL) divergence between $M^*(\cdot|x,a)$ and $M(\cdot|x,a)$, $H_{M^*}(x,a)$ denotes the entropy of $M^*(\cdot|x,a)$, where $D_{KL}(M^*(\cdot|x,a), M(\cdot|x,a))$ is the Kullback-Leibler (KL) divergence between $M^*(\cdot|x,a)$ and $M(\cdot|x,a)$, $H_{M^*}(x,a)$ denotes the entropy of $M^*(\cdot|x,a)$, and $\alpha_M$ is the regularization coefficient $\alpha$ in Eq. 9 and also as a normalizer.*

*Proof.* Given a transition function $M$ of an MDP, the distribution of the best-response policy $\beta^*$ satisfies:

$$\rho_{M^*}^{\beta^*} = \arg\max_{\rho_{M^*}^{\beta}} \int_{\mathcal{X},\mathcal{A}} \rho_{M^*}^{\mu}(x,a)\omega(x,a) \int_{\mathcal{X}} M^*(x'|x,a)\left(-\log M(x'|x,a)\right)\mathrm{d}x'\mathrm{d}a\mathrm{d}x - \frac{\alpha}{2}\|\rho_{M^*}^{\beta}(\cdot,\cdot)\|_2^2$$

$$= \arg\max_{\rho_{M^*}^{\beta}} \int_{\mathcal{X},\mathcal{A}} \rho_{M^*}^{\beta}(x,a) \underbrace{\int_{\mathcal{X}} M^*(x'|x,a)\left(-\log M(x'|x,a)\right)\mathrm{d}x'}_{g(x,a)}\mathrm{d}a\mathrm{d}x - \frac{\alpha}{2}\|\rho_{M^*}^{\beta}(\cdot,\cdot)\|_2^2$$

$$= \arg\max_{\rho_{M^*}^{\beta}} \frac{2}{\alpha}\int_{\mathcal{X},\mathcal{A}} \rho_{M^*}^{\beta}(x,a)g(x,a)\mathrm{d}a\mathrm{d}x - \|\rho_{M^*}^{\beta}(\cdot,\cdot)\|_2^2$$

$$= \arg\max_{\rho_{M^*}^{\beta}} \frac{2}{\alpha}\int_{\mathcal{X},\mathcal{A}} \rho_{M^*}^{\beta}(x,a)g(x,a)\mathrm{d}a\mathrm{d}x - \|\rho_{M^*}^{\beta}(\cdot,\cdot)\|_2^2 - \frac{\|g(\cdot,\cdot)\|_2^2}{\alpha^2}$$

$$= \arg\max_{\rho_{M^*}^{\beta}} -\left(-2\int_{\mathcal{X},\mathcal{A}} \rho_{M^*}^{\beta}(x,a)\frac{g(x,a)}{\alpha}\mathrm{d}a\mathrm{d}x + \|\rho_{M^*}^{\beta}(\cdot,\cdot)\|_2^2 + \frac{\|g(\cdot,\cdot)\|_2^2}{\alpha^2}\right)$$

$$= \arg\max_{\rho_{M^*}^{\beta}} -\|\rho_{M^*}^{\beta}(\cdot,\cdot) - \frac{g(\cdot,\cdot)}{\alpha}\|_2^2.$$

We know that the occupancy measure $\rho_{M^*}^{\beta}$ is a density function with a constraint $\int_{\mathcal{X}} \int_{\mathcal{A}} \rho_{M^*}^{\beta}(x,a) \mathrm{d}a \mathrm{d}x = 1$. Assuming the occupancy measure $\rho_{M^*}^{\beta}$ has an upper bound $c$, that is $0 \leq \rho_{M^*}^{\beta}(x,a) \leq c, \forall a \in \mathcal{A}, \forall x \in \mathcal{X}$, constructing a regularization coefficient $\alpha_M = \int_{\mathcal{X}} \int_{\mathcal{A}} (D_{KL}(M^*(\cdot|x,a), M(\cdot|x,a)) + H_{M^*}(x,a)) \mathrm{d}x \mathrm{d}a$ as a constant value given any $M$, then we have

$$
\begin{aligned}
\rho_{M^*}^{\beta^*}(x,a) &= \frac{g(x,a)}{\alpha_M} \\
&= \frac{\int_{\mathcal{X}} M^*(x'|x,a) \log \frac{M^*(x'|x,a)}{M(x'|x,a)} \mathrm{d}x - \int_{\mathcal{X}} M^*(x'|x,a) \log M^*(x'|x,a) \mathrm{d}x}{\alpha_M} \\
&= \frac{D_{KL}(M^*(\cdot|x,a), M(\cdot|x,a)) + H_{M^*}(x,a)}{\alpha_M} \\
&\propto \left( D_{KL}(M^*(\cdot|x,a), M(\cdot|x,a)) + H_{M^*}(x,a) \right),
\end{aligned}
$$

which is the optimal density function of Eq. 9 with $\alpha = \alpha_M$.

Note that in some particular $M^*$, we still cannot construct a $\beta$ that can generate an occupancy specified by $g(x,a)/\alpha_M$ for any $M$. We can only claim the distribution of the ideal best-response policy $\bar{\beta}^*$ satisfies:

$$
\rho_{M^*}^{\bar{\beta}^*}(x,a) = \frac{1}{\alpha_M}(D_{KL}(M^*(\cdot|x,a), M(\cdot|x,a)) + H_{M^*}(x,a)), \tag{11}
$$

where $\alpha_M$ is a normalizer that $\alpha_M = \int_{\mathcal{X}} \int_{\mathcal{A}} (D_{KL}(M^*(\cdot|x,a), M(\cdot|x,a)) + H_{M^*}(x,a)) \mathrm{d}x \mathrm{d}a$. We give a discussion of the rationality of the ideal best-response policy $\bar{\beta}^*$ as a replacement of the real best-response policy $\beta^*$ in Remark A.2.

$\square$

**Remark A.2.** *The optimal solution Eq. 11 relies on $g(x,a)$. In some particular $M^*$, it is intractable to derive a $\beta$ that can generate an occupancy specified by $g(x,a)/\alpha_M$. Consider the following case: a state $x_1$ in $M^*$ might be harder to reach than another state $x_2$, e.g., $M^*(x_1|x,a) < M^*(x_2|x,a), \forall x \in \mathcal{X}, \forall a \in \mathcal{A}$, then it is impossible to find a $\beta$ that the occupancy satisfies $\rho_{M^*}^{\beta}(x_1,a) > \rho_{M^*}^{\beta}(x_2,a)$. In this case, Eq. 11 can be a sub-optimal solution. Since this work focuses on task-agnostic solution derivation while the solution to the above problem should rely on the specific description of $M^*$, we leave it as future work. However, we point out that Eq. 11 is a reasonable re-weighting term even as a sub-optimum: $\rho_{M^*}^{\bar{\beta}^*}$ gives larger densities on the data where the distribution distance between the approximation model and the real model (i.e., $D_{KL}(M^*, M)$) is larger or the stochasticity of the real model (i.e., $H_{M^*}$) is larger.*

## A.2 Proof of Eq. 6

The integral process of $D_{KL}$ in Eq. 5 is intractable in the offline setting as it explicitly requires the conditional probability function of $M^*$. Our motivation for the tractable solution is utilizing the offline dataset $\mathcal{D}_{\mathrm{real}}$ as the empirical *joint* distribution $\rho_{M^*}^{\mu}(x,a,x')$ and adopting practical techniques for distance estimation on two joint distributions, like GAN (Goodfellow et al., 2014; Nowozin et al., 2016), to approximate Eq. 5. To adopt that solution, we should first transform Eq. 5 into a form under joint distributions. Without loss of generality, we introduce an intermediary policy $\kappa$, of which $\mu$ can be regarded as a specific instance. Then we have $M(x'|x,a) = \rho_M^{\kappa}(x,a,x')/\rho_M^{\kappa}(x,a)$ for any $M$ if $\rho_M^{\kappa}(x,a) > 0$. Assuming $\forall x \in \mathcal{X}, \forall a \in \mathcal{A}, \rho_{M^*}^{\kappa}(x,a) > 0$ if $\rho_{M^*}^{\bar{\beta}^*}(x,a) > 0$, which will hold when $\kappa$ overlaps with $\mu$, then Eq. 5 can transform to:

$$
\begin{aligned}
\rho_{M^*}^{\bar{\beta}^*}(x,a) =& \frac{D_{KL}(M^*(\cdot|x,a), M(\cdot|x,a)) + H_{M^*}(x,a)}{\alpha_M} \\
=& \frac{1}{\alpha_M} \int_{\mathcal{X}} M^*(x'|x,a) \left( \log \frac{M^*(x'|x,a)}{M(x'|x,a)} - \log M^*(x'|x,a) \right) \mathrm{d}x' \\
=& \frac{1}{\alpha_M \rho_{M^*}^{\kappa}(x,a)} \int_{\mathcal{X}} \rho_{M^*}^{\kappa}(x,a) M^*(x'|x,a) \left( \log \frac{M^*(x'|x,a)}{M(x'|x,a)} - \log M^*(x'|x,a) \right) \mathrm{d}x'
\end{aligned}
$$
$$(12)$$

$$
\begin{aligned}
=& \frac{1}{\alpha_M \rho_{M^*}^{\kappa}(x,a)} \int_{\mathcal{X}} \rho_{M^*}^{\kappa}(x,a,x') \left( \log \frac{\rho_{M^*}^{\kappa}(x,a,x')}{\rho_M^{\kappa}(x,a,x')} + \log \frac{\rho_M^{\kappa}(x,a)}{\rho_{M^*}^{\kappa}(x,a)} - \log M^*(x'|x,a) \right) \mathrm{d}x' \\
=& \frac{1}{\alpha_M \rho_{M^*}^{\kappa}(x,a)} \left( \int_{\mathcal{X}} \rho_{M^*}^{\kappa}(x,a,x') \log \frac{\rho_{M^*}^{\kappa}(x,a,x')}{\rho_M^{\kappa}(x,a,x')} \mathrm{d}x' - \right. \\
& \rho_{M^*}^{\kappa}(x,a) \log \frac{\rho_{M^*}^{\kappa}(x,a)}{\rho_M^{\kappa}(x,a)} \underbrace{\int_{\mathcal{X}} M^*(x'|x,a) \mathrm{d}x'}_{=1} - \left. \rho_{M^*}^{\kappa}(x,a) \int_{\mathcal{X}} M^*(x'|x,a) \log M^*(x'|x,a) \mathrm{d}x' \right) \\
=& \frac{1}{\alpha_0(x,a)} \left( \int_{\mathcal{X}} \rho_{M^*}^{\kappa}(x,a,x') \log \frac{\rho_{M^*}^{\kappa}(x,a,x')}{\rho_M^{\kappa}(x,a,x')} \mathrm{d}x' - \rho_{M^*}^{\kappa}(x,a) \log \frac{\rho_{M^*}^{\kappa}(x,a)}{\rho_M^{\kappa}(x,a)} + \rho_{M^*}^{\kappa}(x,a) H_{M^*}(x,a) \right)
\end{aligned}
$$
$$(13)$$

where $\alpha_0(x,a) = \alpha_M \rho_{M^*}^{\kappa}(x,a)$.

**Definition A.3** ($f$-divergence). *Given two distributions $P$ and $Q$, two absolutely continuous density functions $p$ and $q$ with respect to a base measure $\mathrm{d}x$ defined on the domain $\mathcal{X}$, we define the $f$-divergence (Nowozin et al., 2016),*

$$
D_f(P\|Q) = \int_{\mathcal{X}} q(x) f\left( \frac{p(x)}{q(x)} \right) \mathrm{d}x,
$$
$$(14)$$

*where the generator function $f : \mathbb{R}_+ \to \mathbb{R}$ is a convex, lower-semicontinuous function.*

We notice that the terms $\rho_{M^*}^{\kappa}(x,a,x') \log \frac{\rho_{M^*}^{\kappa}(x,a,x')}{\rho_M^{\kappa}(x,a,x')}$ and $\rho_{M^*}^{\kappa}(x,a) \log \frac{\rho_{M^*}^{\kappa}(x,a)}{\rho_M^{\kappa}(x,a)}$ are the integrated functions in reverse KL divergence, which is an instance of $f$ function in $f$-divergence (See Reverse-KL divergence of Tab.1 in (Nowozin et al., 2016) for more details). Replacing that form $q \log \frac{q}{p}$ with $q f(\frac{p}{q})$, we obtain a generalized representation of $\rho_{M^*}^{\bar{\beta}^*}$:

$$
\bar{\rho}_{M^*}^{\bar{\beta}^*} := \frac{1}{\alpha_0(x,a)} \left( \int_{\mathcal{X}} \rho_{M^*}^{\kappa}(x,a,x') f\left( \frac{\rho_M^{\kappa}(x,a,x')}{\rho_{M^*}^{\kappa}(x,a,x')} \right) \mathrm{d}x' - \rho_{M^*}^{\kappa}(x,a) \left( f\left( \frac{\rho_M^{\kappa}(x,a)}{\rho_{M^*}^{\kappa}(x,a)} \right) - H_{M^*}(x,a) \right) \right),
$$
$$(15)$$

## A.3 PROOF OF THM. 4.4

We first introduce several useful lemmas for the proof.

**Lemma A.4.** *Rearrangement inequality The rearrangement inequality states that, for two sequences $a_1 \geq a_2 \geq \ldots \geq a_n$ and $b_1 \geq b_2 \geq \ldots \geq b_n$, the inequalities*

$$
a_1 b_1 + a_2 b_2 + \cdots + a_n b_n \geq a_1 b_{\pi(1)} + a_2 b_{\pi(2)} + \cdots + a_n b_{\pi(n)} \geq a_1 b_n + a_2 b_{n-1} + \cdots + a_n b_1
$$

*hold, where $\pi(1), \pi(2), \ldots, \pi(n)$ is any permutation of $1, 2, \ldots, n$.*

**Lemma A.5.** *For two sequences $a_1 \geq a_2 \geq \ldots \geq a_n$ and $b_1 \geq b_2 \geq \ldots \geq b_n$, the inequalities*

$$
\sum_{i=1}^{n} \frac{1}{n} a_i b_i \geq \sum_{i=1}^{n} \frac{1}{n} a_i \sum \frac{1}{n} b_i
$$

*hold.*

*Proof.* By rearrangement inequality, we have

$$\sum_{i=1}^{n} a_i b_i \geq a_1 b_1 + a_2 b_2 + \cdots + a_n b_n$$

$$\sum_{i=1}^{n} a_i b_i \geq a_1 b_2 + a_2 b_3 + \cdots + a_n b_1$$

$$\sum_{i=1}^{n} a_i b_i \geq a_1 b_3 + a_2 b_4 + \cdots + a_n b_2$$

$$\vdots$$

$$\sum_{i=1}^{n} a_i b_i \geq a_1 b_n + a_2 b_1 + \cdots + a_n b_{n-1}$$

Then we have

$$n \sum_{i=1}^{n} a_i b_i \geq \sum_{i=1}^{n} a_i \sum_{i=1}^{n} b_i$$

$$\sum_{i=1}^{n} \frac{1}{n} a_i b_i \geq \sum_{i=1}^{n} \frac{1}{n} a_i \sum \frac{1}{n} b_i$$

$\square$

Now we extend Lemma A.5 into the continuous integral scenario:

**Lemma A.6.** *Given $\mathcal{X} \subset \mathbb{R}$, for two functions $f : \mathcal{X} \to \mathbb{R}$ and $g : \mathcal{X} \to \mathbb{R}$ that $f(x) \geq f(y)$ if and only if $g(x) \geq g(y)$, $\forall x, y \in \mathcal{X}$, the inequality*

$$\int_{\mathcal{X}} p(x) f(x) g(x) \mathrm{d}x \geq \int_{\mathcal{X}} p(x) f(x) \mathrm{d}x \int_{\mathcal{X}} p(x) g(x) \mathrm{d}x$$

*holds, where $p : \mathcal{X} \to \mathbb{R}$ and $p(x) > 0, \forall x \in \mathcal{X}$ and $\int_{\mathcal{X}} p(x) \mathrm{d}x = 1$.*

*Proof.* Since $(f(x) - f(y))(g(x) - g(y)) \geq 0, \forall x, y \in \mathcal{X}$, we have

$$\int_{x \in \mathcal{X}} \int_{y \in \mathcal{X}} p(x) p(y) (f(x) - f(y))(g(x) - g(y)) \mathrm{d}y \mathrm{d}x \geq 0$$

$$\int_{x \in \mathcal{X}} \int_{y \in \mathcal{X}} p(x)p(y)f(x)g(x) + p(x)p(y)f(y)g(y) - p(x)p(y)f(x)g(y) - p(x)p(y)f(y)g(x) \mathrm{d}y \mathrm{d}x \geq 0$$

$$\int_{x \in \mathcal{X}} \int_{y \in \mathcal{X}} p(x)p(y)f(x)g(x) + p(x)p(y)f(y)g(y) \mathrm{d}y \mathrm{d}x \geq \int_{x \in \mathcal{X}} \int_{y \in \mathcal{X}} p(x)p(y)f(x)g(y) + p(x)p(y)f(y)g(x) \mathrm{d}y \mathrm{d}x$$

$$\int_{x \in \mathcal{X}} \left( \int_{y \in \mathcal{X}} p(x)p(y)f(x)g(x) \mathrm{d}y + \int_{y \in \mathcal{X}} p(x)p(y)f(y)g(y) \mathrm{d}y \right) \mathrm{d}x \geq \int_{x \in \mathcal{X}} \int_{y \in \mathcal{X}} p(x)p(y)f(x)g(y) + p(x)p(y)f(y)g(x) \mathrm{d}y \mathrm{d}x$$

$$\int_{x \in \mathcal{X}} \left( p(x)f(x)g(x) + \int_{y \in \mathcal{X}} p(x)p(y)f(y)g(y) \mathrm{d}y \right) \mathrm{d}x \geq \int_{x \in \mathcal{X}} \int_{y \in \mathcal{X}} p(x)p(y)f(x)g(y) + p(x)p(y)f(y)g(x) \mathrm{d}y \mathrm{d}x$$

$$\int_{x \in \mathcal{X}} p(x)f(x)g(x) \mathrm{d}x + \int_{x \in \mathcal{X}} \int_{y \in \mathcal{X}} p(x)p(y)f(y)g(y) \mathrm{d}y \mathrm{d}x \geq \int_{x \in \mathcal{X}} \int_{y \in \mathcal{X}} p(x)p(y)f(x)g(y) + p(x)p(y)f(y)g(x) \mathrm{d}y \mathrm{d}x$$

$$\int_{x \in \mathcal{X}} p(x)f(x)g(x) \mathrm{d}x + \int_{y \in \mathcal{X}} p(y)f(y)g(y) \mathrm{d}y \geq \int_{x \in \mathcal{X}} \int_{y \in \mathcal{X}} p(x)p(y)f(x)g(y) + p(x)p(y)f(y)g(x) \mathrm{d}y \mathrm{d}x$$

$$2 \int_{x \in \mathcal{X}} p(x)f(x)g(x) \mathrm{d}x \geq 2 \int_{y \in \mathcal{X}} \int_{x \in \mathcal{X}} p(x)p(y)f(x)g(y) \mathrm{d}y \mathrm{d}x$$

$$2 \int_{x \in \mathcal{X}} p(x)f(x)g(x) \mathrm{d}x \geq 2 \int_{x \in \mathcal{X}} p(x)f(x) \mathrm{d}x \int_{x \in \mathcal{X}} p(x)g(x) \mathrm{d}x$$

$$\int_{x \in \mathcal{X}} p(x)f(x)g(x) \mathrm{d}x \geq \int_{x \in \mathcal{X}} p(x)f(x) \mathrm{d}x \int_{x \in \mathcal{X}} p(x)g(x) \mathrm{d}x$$

$\square$

**Corollary A.7.** *Let* $g(\frac{p(x)}{q(x)}) = -\log\frac{p(x)}{q(x)}$ *where* $p(x) > 0, \forall x \in \mathcal{X}$ *and* $q(x) > 0, \forall x \in \mathcal{X}$, *for* $\upsilon > 0$, *the inequality*

$$\int_{\mathcal{X}} q(x)f(\upsilon\frac{p(x)}{q(x)})g(\frac{p(x)}{q(x)})\mathrm{d}x \geq \int_{\mathcal{X}} q(x)f(\upsilon\frac{p(x)}{q(x)})\mathrm{d}x \int_{\mathcal{X}} q(x)g(\frac{p(x)}{q(x)})\mathrm{d}x,$$

*holds if* $f'(x) \leq 0, \forall x \in \mathcal{X}$. *It is not always satisfied for* $f$ *functions of* $f$-*divergence. We list a comparison of* $f$ *on that condition in Tab. 2.*

*Proof.* $g'(x) = -\log x = -\frac{1}{x} < 0, \forall x \in \mathcal{X}$. Suppose $f'(x) \leq 0, \forall x \in \mathcal{X}$, we have $f(x) \geq f(y)$ if and only if $g(x) \geq g(y)$, $\forall x, y \in \mathcal{X}$ holds. Thus $f(\upsilon\frac{p(x)}{q(x)}) \geq f(\upsilon\frac{p(y)}{q(y)})$ if and only if $g(\frac{p(x)}{q(x)}) \geq g(\frac{p(y)}{q(y)})$, $\forall x, y \in \mathcal{X}$ holds for all $\upsilon > 0$. By defining $F(x) = f(\upsilon\frac{p(x)}{q(x)}))$ and $G(x) = g(\frac{p(x)}{q(x)})$ and using Lemma A.6, we have:

$$\int_{\mathcal{X}} q(x)F(x)G(x)\mathrm{d}x \geq \int_{\mathcal{X}} q(x)F(x)\mathrm{d}x \int_{\mathcal{X}} q(x)G(x)\mathrm{d}x.$$

Then we know

$$\int_{\mathcal{X}} q(x)f(\upsilon\frac{p(x)}{q(x)})g(\frac{p(x)}{q(x)})\mathrm{d}x \geq \int_{\mathcal{X}} q(x)f(\upsilon\frac{p(x)}{q(x)})\mathrm{d}x \int_{\mathcal{X}} q(x)g(\frac{p(x)}{q(x)})\mathrm{d}x$$

holds. $\square$

Table 2: Properties of $f'(x) \leq 0, \forall x \in \mathcal{X}$ for $f$-divergences.

| Name | Generator function $f(x)$ | If $f'(x) \leq 0, \forall x \in \mathcal{X}$ |
|------|---------------------------|---------------------------------------------|
| Kullback-Leibler | $x\log x$ | False |
| Reverse KL | $-\log x$ | True |
| Pearson $\chi^2$ | $(x-1)^2$ | False |
| Squared Hellinger | $(\sqrt{x}-1)^2$ | False |
| Jensen-Shannon | $-(x+1)\log\frac{1+x}{2} + x\log x$ | False |
| GAN | $x\log x - (x+1)\log(x+1)$ | True |

Now, we prove Thm. 4.4. For better readability, we first rewrite Thm. 4.4 as follows:

**Theorem A.8.** *Let* $\bar{\rho}_{M^*}^{\bar{\beta}^*}$ *as the data distribution of the best-response policy* $\bar{\beta}^*$ *in Eq. 4 under model* $M_\theta$ *parameterized by* $\theta$, *then we can find the optimal* $\theta^*$ *of* $\min_\theta \max_{\beta \in \Pi} \bar{L}(\rho_{M^*}^\beta, M_\theta)$ *(Eq. 4) via iteratively optimizing the objective* $\theta_{t+1} = \min_\theta \bar{L}(\bar{\rho}_{M^*}^{\bar{\beta}^*}, M_\theta)$, *where* $\bar{\rho}_{M^*}^{\bar{\beta}^*}$ *is approximated via the last-iteration model* $M_{\theta_t}$. *Based on Corollary A.7, we have an upper bound objective for* $\min_\theta \bar{L}(\bar{\rho}_{M^*}^{\bar{\beta}^*}, M_\theta)$ *and derive the following objective*

$$\theta_{t+1} = \arg\max_\theta \mathbb{E}_{\rho_{M^*}^\kappa}\left[\frac{1}{\alpha_0(x,a)}\log M_\theta(x'|x,a)\underbrace{\left(f\left(\frac{\rho_{M_{\theta_t}}^\kappa(x,a,x')}{\rho_{M^*}^\kappa(x,a,x')}\right) - f\left(\frac{\rho_{M_{\theta_t}}^\kappa(x,a)}{\rho_{M^*}^\kappa(x,a)}\right) + H_{M^*}(x,a)\right)}_{W(x,a,x')}\right],$$

*where* $\alpha_0(x,a) = \alpha_{M_{\theta_t}}\rho_{M^*}^\kappa(x,a)$, $\mathbb{E}_{\rho_{M^*}^\kappa}[\cdot]$ *denotes* $\mathbb{E}_{x,a,x'\sim\rho_{M^*}^\kappa}[\cdot]$, $f$ *is the generator function in* $f$-*divergence which satisfies* $f'(x) \leq 0, \forall x \in \mathcal{X}$, *and* $\theta$ *is the parameters of* $M$. $M_{\theta_t}$ *denotes a probability function with the same parameters as the learned model (i.e.,* $\bar{\theta} = \theta$*) but the parameter is fixed and only used for sampling.*

*Proof.* Let $\bar{\rho}_{M^*}^{\bar{\beta}^*}$ as the data distribution of the best-response policy $\bar{\beta}^*$ in Eq. 4 under model $M_\theta$ parameterized by $\theta$, then we can find the optimal $\theta_{t+1}$ of $\min_\theta \max_{\beta \in \Pi} \bar{L}(\rho_{M^*}^\beta, M_\theta)$ (Eq. 4) via

iteratively optimizing the objective $\theta_{t+1} = \min_\theta \bar{L}(\bar{\rho}_{M^*}^{\bar{\beta}^*}, M_\theta)$, where $\bar{\rho}_{M^*}^{\bar{\beta}^*}$ is approximated via the last-iteration model $\bar{M}_{\theta_t}$:

$$\theta_{t+1} = \min_\theta \int_{\mathcal{X},\mathcal{A}} \bar{\rho}_{M^*}^{\bar{\beta}^*}(x,a) \int_{\mathcal{X}} M^*(x'|x,a) \left(-\log M_\theta(x'|x,a)\right) \mathrm{d}x'\mathrm{d}a\mathrm{d}x \tag{16}$$

$$= \min_\theta \int_{\mathcal{X},\mathcal{A}} \frac{1}{\alpha_0(x,a)} \left( \int_{\mathcal{X}} \rho_{M^*}^\kappa(x,a,x') f\left(\frac{\rho_{M_{\theta_t}}^\kappa(x,a,x')}{\rho_{M^*}^\kappa(x,a,x')}\right) \mathrm{d}x' \int_{\mathcal{X}} M^*(x'|x,a)(-\log M_\theta(x'|x,a))\mathrm{d}x' \right.$$
$$\left. - \rho_{M^*}^\kappa(x,a) \left( f\left(\frac{\rho_{M_{\theta_t}}^\kappa(x,a)}{\rho_{M^*}^\kappa(x,a)}\right) - H_{M^*}(x,a) \right) \int_{\mathcal{X}} M^*(x'|x,a)(-\log M_\theta(x'|x,a))\mathrm{d}x' \right) \mathrm{d}a\mathrm{d}x$$

$$= \min_\theta \int_{\mathcal{X},\mathcal{A}} \frac{1}{\alpha_0(x,a)} \left( \int_{\mathcal{X}} \rho_{M^*}^\kappa(x,a,x') f\left(\frac{\rho_{M_{\theta_t}}^\kappa(x,a,x')}{\rho_{M^*}^\kappa(x,a,x')}\right) \mathrm{d}x' \left( \int_{\mathcal{X}} M^*(x'|x,a)(-\log \frac{M_\theta(x'|x,a)}{M^*(x'|x,a)})\mathrm{d}x' + H_{M^*}(x,a) \right) \right.$$
$$\left. - \rho_{M^*}^\kappa(x,a) \left( f\left(\frac{\rho_{M_{\theta_t}}^\kappa(x,a)}{\rho_{M^*}^\kappa(x,a)}\right) - H_{M^*}(x,a) \right) \int_{\mathcal{X}} M^*(x'|x,a)(-\log M_\theta(x'|x,a))\mathrm{d}x' \right) \mathrm{d}a\mathrm{d}x$$

$$\leq \min_\theta \int_{\mathcal{X},\mathcal{A}} \frac{1}{\alpha_0(x,a)} \left( \underbrace{\rho_{M^*}^\kappa(x,a) \int_{\mathcal{X}} M^*(x'|x,a) f\left(\frac{\rho_{M_{\theta_t}}^\kappa(x,a,x')}{\rho_{M^*}^\kappa(x,a,x')}\right) (-\log \frac{M_\theta(x'|x,a)}{M^*(x'|x,a)})\mathrm{d}x'}_{\text{based on Corollary } A.7} \right.$$
$$\left. - \rho_{M^*}^\kappa(x,a) \left( f\left(\frac{\rho_{M_{\theta_t}}^\kappa(x,a)}{\rho_{M^*}^\kappa(x,a)}\right) - H_{M^*}(x,a) \right) \int_{\mathcal{X}} M^*(x'|x,a)(-\log M_\theta(x'|x,a))\mathrm{d}x' \right) \mathrm{d}a\mathrm{d}x$$

$$= \min_\theta \int_{\mathcal{X},\mathcal{A}} \frac{1}{\alpha_0(x,a)} \left( \rho_{M^*}^\kappa(x,a) \int_{\mathcal{X}} \left( M^*(x'|x,a) f\left(\frac{\rho_{M_{\theta_t}}^\kappa(x,a,x')}{\rho_{M^*}^\kappa(x,a,x')}\right) (-\log M_\theta(x'|x,a)) \right) \mathrm{d}x' \right.$$
$$\left. - \rho_{M^*}^\kappa(x,a) \left( f\left(\frac{\rho_{M_{\theta_t}}^\kappa(x,a)}{\rho_{M^*}^\kappa(x,a)}\right) - H_{M^*}(x,a) \right) \int_{\mathcal{X}} M^*(x'|x,a)(-\log M_\theta(x'|x,a))\mathrm{d}x' \right) \mathrm{d}a\mathrm{d}x$$

$$= \max_\theta \int_{\mathcal{X},\mathcal{A},\mathcal{X}} \frac{1}{\alpha_0(x,a)} \rho_{M^*}^\kappa(x,a,x') \log M_\theta(x'|x,a) \left( f\left(\frac{\rho_{M_{\theta_t}}^\kappa(x,a,x')}{\rho_{M^*}^\kappa(x,a,x')}\right) - f\left(\frac{\rho_{M_{\theta_t}}^\kappa(x,a)}{\rho_{M^*}^\kappa(x,a)}\right) + H_{M^*}(x,a) \right) \mathrm{d}x'\mathrm{d}a\mathrm{d}x, \tag{17}$$

where $M_{\theta_t}$ is introduced to approximate the term $\bar{\rho}_{M^*}^{\bar{\beta}^*}$ and fixed when optimizing $\theta$. In Eq. 16, $\|\rho_{M^*}^\beta(\cdot,\cdot)\|_2^2$ for Eq. 9 is eliminated as it does not contribute to the gradient of $\theta$. Assume $f'(x) \leq 0, \forall x \in \mathcal{X}$, let $\upsilon(x,a) := \frac{\rho_{M_{\theta_t}}^\kappa(x,a)}{\rho_{M^*}^\kappa(x,a)} > 0$, $p(x'|x,a) = M_\theta(x'|x,a)$, and $q(x'|x,a) = M^*(x'|x,a)$, the first inequality can be derived by adopting Corollary A.7 and eliminating the first $H_{M^*}$ since it does not contribute to the gradient of $\theta$.

$\square$

## A.4 PROOF OF THE TRACTABLE SOLUTION

Now we are ready to prove the tractable solution:

*Proof.* The core challenge is that the term $f(\frac{\rho_{M_{\theta_t}}^\kappa(x,a,x')}{\rho_{M^*}^\kappa(x,a,x')}) - f(\frac{\rho_{M_{\theta_t}}^\kappa(x,a)}{\rho_{M^*}^\kappa(x,a)})$ is still intractable. In the following, we give a tractable solution to Thm. 4.4. First, we resort to the first-order approximation. Given some $u \in (1-\xi, 1+\xi), \xi > 0$, we have

$$f(u) \approx f(1) + f'(u)(u - 1), \tag{18}$$

where $f'$ is the first-order derivative of $f$. By Taylor's formula and the fact that $f'(u)$ of the generator function $f$ is bounded in $(1-\xi, 1+\xi)$, the approximation error is no more than $\mathcal{O}(\xi^2)$. Substituting $u$ with $\frac{p(x)}{q(x)}$ in Eq. 18, the pattern $f(\frac{p(x)}{q(x)})$ in Eq. 17 can be converted to $\frac{p(x)}{q(x)} f'(\frac{p(x)}{q(x)}) - f'(\frac{p(x)}{q(x)}) + f(1)$,

then we have:

$$
\theta_{t+1} = \arg\max_\theta \frac{1}{\alpha_0(x,a)} \int_{\mathcal{X},\mathcal{A}} \left( \rho_{M^*}^\kappa(x,a) \int_{\mathcal{X}} M^*(x'|x,a) f\left( \frac{\rho_{M_{\theta_t}}^\kappa(x,a,x')}{\rho_{M^*}^\kappa(x,a,x')} \right) \log M_\theta(x'|x,a) \mathrm{d}x' - \right.
$$

$$
\rho_{M^*}^\kappa(x,a) f\left( \frac{\rho_{M_{\theta_t}}^\kappa(x,a)}{\rho_{M^*}^\kappa(x,a)} \right) \int_{\mathcal{X}} M^*(x'|x,a) \log M_\theta(x'|x,a) \mathrm{d}x' +
$$

$$
\left. \rho_{M^*}^\kappa(x,a) H_{M^*}(x,a) \int_{\mathcal{X}} M^*(x'|x,a) \log M_\theta(x'|x,a) \mathrm{d}x' \right) \mathrm{d}a \mathrm{d}x
$$

$$
\approx \arg\max_\theta \int_{\mathcal{X},\mathcal{A}} \left( \rho_{M_{\theta_t}}^\kappa(x,a) \int_{\mathcal{X}} M_{\theta_t}(x'|x,a) f'\left( \frac{\rho_{M_{\theta_t}}^\kappa(x,a,x')}{\rho_{M^*}^\kappa(x,a,x')} \right) \log M_\theta(x'|x,a) \mathrm{d}x' - \right.
$$

$$
\rho_{M^*}^\kappa(x,a) \int_{\mathcal{X}} M^*(x'|x,a) \left( f'\left( \frac{\rho_{M_{\theta_t}}^\kappa(x,a,x')}{\rho_{M^*}^\kappa(x,a,x')} \right) - f(1) \right) \log M_\theta(x'|x,a) \mathrm{d}x' -
$$

$$
\rho_{M_{\theta_t}}^\kappa(x,a) f'\left( \frac{\rho_{M_{\theta_t}}^\kappa(x,a)}{\rho_{M^*}^\kappa(x,a)} \right) \int_{\mathcal{X}} M^*(x'|x,a) \log M_\theta(x'|x,a) \mathrm{d}x' +
$$

$$
\rho_{M^*}^\kappa(x,a) \left( f'\left( \frac{\rho_{M_{\theta_t}}^\kappa(x,a)}{\rho_{M^*}^\kappa(x,a)} \right) - f(1) \right) \int_{\mathcal{X}} M^*(x'|x,a) \log M_\theta(x'|x,a) \mathrm{d}x' +
$$

$$
\left. \rho_{M^*}^\kappa(x,a) H_{M^*}(x,a) \int_{\mathcal{X}} M^*(x'|x,a) \log M_\theta(x'|x,a) \mathrm{d}x' \right) \mathrm{d}a \mathrm{d}x
$$

$$
= \arg\max_\theta \int_{\mathcal{X},\mathcal{A},\mathcal{X}} \frac{1}{\alpha_0(x,a)} \rho_{M_{\theta_t}}^\kappa(x,a,x) \left( f'\left( \frac{\rho_{M_{\theta_t}}^\kappa(x,a,x')}{\rho_{M^*}^\kappa(x,a,x')} \right) - f'\left( \frac{\rho_{M_{\theta_t}}^\kappa(x,a)}{\rho_{M^*}^\kappa(x,a)} \right) \right) \log M_\theta(x'|x,a) \mathrm{d}x' \mathrm{d}a \mathrm{d}x +
$$

$$
\int_{\mathcal{X},\mathcal{A},\mathcal{X}} \frac{1}{\alpha_0(x,a)} \rho_{M^*}^\kappa(x,a,x') \left( f'\left( \frac{\rho_{M_{\theta_t}}^\kappa(x,a)}{\rho_{M^*}^\kappa(x,a)} \right) - f'\left( \frac{\rho_{M_{\theta_t}}^\kappa(x,a,x')}{\rho_{M^*}^\kappa(x,a,x')} \right) + H_{M^*}(x,a) \right) \log M_\theta(x'|x,a) \mathrm{d}x' \mathrm{d}a \mathrm{d}x.
$$

Note that the part $\rho_{M^*}^\kappa(x,a)$ in $\rho_{M^*}^\kappa(x,a,x')$ can be canceled because of $\alpha_0(x,a) = \alpha_{M_{\theta_t}} \rho_{M^*}^\kappa(x,a)$, but we choose to keep it and ignore $\alpha_0(x,a)$. The benefit is that we can estimate $\rho_{M^*}^\kappa(x,a,x')$ from an empirical data distribution through data collected by $\kappa$ in $M^*$ directly, rather than from a uniform distribution which is harder to be generated. Although keeping $\rho_{M^*}^\kappa(x,a)$ incurs extra bias in theory, the results in our experiments show that it has not made significant negative effects in practice. We leave this part of modeling in future work. In particular, by ignoring $\alpha_0(x,a)$, we have:

$$
\theta_{t+1} = \arg\max_\theta \int_{\mathcal{X},\mathcal{A},\mathcal{X}} \rho_{M_{\theta_t}}^\kappa(x,a,x) \left( f'\left( \frac{\rho_{M_{\theta_t}}^\kappa(x,a,x')}{\rho_{M^*}^\kappa(x,a,x')} \right) - f'\left( \frac{\rho_{M_{\theta_t}}^\kappa(x,a)}{\rho_{M^*}^\kappa(x,a)} \right) \right) \log M_\theta(x'|x,a) \mathrm{d}x' \mathrm{d}a \mathrm{d}x +
$$

$$
\tag{19}
$$

$$
\int_{\mathcal{X},\mathcal{A},\mathcal{X}} \rho_{M^*}^\kappa(x,a,x') \left( f'\left( \frac{\rho_{M_{\theta_t}}^\kappa(x,a)}{\rho_{M^*}^\kappa(x,a)} \right) - f'\left( \frac{\rho_{M_{\theta_t}}^\kappa(x,a,x')}{\rho_{M^*}^\kappa(x,a,x')} \right) + H_{M^*}(x,a) \right) \log M_\theta(x'|x,a) \mathrm{d}x' \mathrm{d}a \mathrm{d}x.
$$

$$
\tag{20}
$$

We can estimate $f'\left( \frac{\rho_{M_{\theta_t}}^\kappa(x,a)}{\rho_{M^*}^\kappa(x,a)} \right)$ and $f'\left( \frac{\rho_{M_{\theta_t}}^\kappa(x,a,x')}{\rho_{M^*}^\kappa(x,a,x')} \right)$ through Lemma A.9.

**Lemma A.9** ($f'(\frac{p}{q})$ estimation (Nguyen et al., 2010))**.** *Given a function $T_\varphi : \mathcal{X} \to \mathbb{R}$ parameterized by $\varphi \in \Phi$, if $f$ is convex and lower semi-continuous, by finding the maximum point of $\varphi$ in the following objective:*

$$
\varphi^* = \arg\max_\varphi \mathbb{E}_{x\sim p(x)} \left[ T_\varphi(x) \right] - \mathbb{E}_{x\sim q(x)} \left[ f^*(T_\varphi(x)) \right],
$$

*we have $f'(\frac{p(x)}{q(x)}) = T_{\varphi^*}(x)$. $f^*$ is Fenchel conjugate of $f$ (Hiriart-Urruty & Lemaréchal, 2001).*

In particular,

$$
\varphi_0^* = \arg\max_{\varphi_0} \mathbb{E}_{x,a,x'\sim\rho_{M^*}^\kappa} \left[ T_{\varphi_0}(x,a,x') \right] - \mathbb{E}_{x,a,x'\sim\rho_{M_{\theta_t}}^\kappa} \left[ f^*(T_{\varphi_0}(x,a,x')) \right]
$$

$$
\varphi_1^* = \arg\max_{\varphi_1} \mathbb{E}_{x,a\sim\rho_{M^*}^\kappa} \left[ T_{\varphi_1}(x,a) \right] - \mathbb{E}_{x,a\sim\rho_{M_{\theta_t}}^\kappa} \left[ f^*(T_{\varphi_1}(x,a)) \right],
$$

then we have $f'\left(\frac{\rho^\kappa_{M_{\theta_t}}(x,a,x')}{\rho^\kappa_{M^*}(x,a,x')}\right) \approx T_{\varphi_0^*}(x,a,x')$ and $f'\left(\frac{\rho^\kappa_{M_{\theta_t}}(x,a)}{\rho^\kappa_{M^*}(x,a)}\right) \approx T_{\varphi_1^*}(x,a)$. Given $\varphi_0^*$ and $\varphi_1^*$, let $A_{\varphi_0^*,\varphi_1^*}(x,a,x') = T_{\varphi_0^*}(x,a,x') - T_{\varphi_1^*}(x,a)$, then we can optimize $\theta$ via:

$$
\begin{aligned}
\theta_{t+1} &= \arg\max_\theta \int_{\mathcal{X},\mathcal{A},\mathcal{X}} \rho^\kappa_{M_{\theta_t}}(x,a,x) \left(T_{\varphi_0^*}(x,a,x') - T_{\varphi_1^*}(x,a)\right) \log M_\theta(x'|x,a)\mathrm{d}x'\mathrm{d}a\mathrm{d}x + \\
&\qquad \int_{\mathcal{X},\mathcal{A},\mathcal{X}} \rho^\kappa_{M^*}(x,a,x') \left(T_{\varphi_1^*}(x,a) - T_{\varphi_0^*}(x,a,x') + H_{M^*}(x,a)\right) \log M_\theta(x'|x,a)\mathrm{d}x'\mathrm{d}a\mathrm{d}x \\
&= \arg\max_\theta \int_{\mathcal{X},\mathcal{A},\mathcal{X}} \rho^\kappa_{M_{\theta_t}}(x,a,x) A_{\varphi_0^*,\varphi_1^*}(x,a,x') \log M_\theta(x'|x,a)\mathrm{d}x'\mathrm{d}a\mathrm{d}x + \\
&\qquad \int_{\mathcal{X},\mathcal{A},\mathcal{X}} \rho^\kappa_{M^*}(x,a,x')(-A_{\varphi_0^*,\varphi_1^*}(x,a,x') + H_{M^*}(x,a)) \log M_\theta(x'|x,a)\mathrm{d}x'\mathrm{d}a\mathrm{d}x.
\end{aligned}
$$

Based on the specific $f$-divergence, we can represent $T$ and $f^*(T)$ with a discriminator $D_\varphi$. It can be verified that $f(u) = u \log u - (u+1)\log(u+1)$, $T_\varphi(u) = \log D_\varphi(u)$, and $f^*(T_\varphi(u)) = -\log(1 - D_\varphi(u))$ proposed in Nowozin et al. (2016) satisfies the condition $f'(x) \leq 0, \forall x \in \mathcal{X}$ (see Tab. 2). We select the former in the implementation and convert the tractable solution to:

$$
\begin{aligned}
\theta_{t+1} &= \arg\max_\theta \mathbb{E}_{\rho^\kappa_{M_{\theta_t}}} \left[A_{\varphi_0^*,\varphi_1^*}(x,a,x') \log M_\theta(x'|x,a)\right] + \mathbb{E}_{\rho^\kappa_{M^*}} \left[(H_{M^*}(x,a) - A_{\varphi_0^*,\varphi_1^*}(x,a,x')) \log M_\theta(x'|x,a)\right] \\
s.t. \quad \varphi_0^* &= \arg\max_{\varphi_0} \mathbb{E}_{\rho^\kappa_{M^*}} \left[\log D_{\varphi_0}(x,a,x')\right] + \mathbb{E}_{\rho^\kappa_{M_{\theta_t}}} \left[\log(1 - D_{\varphi_0}(x,a,x'))\right] \\
\varphi_1^* &= \arg\max_{\varphi_1} \mathbb{E}_{\rho^\kappa_{M^*}} \left[\log D_{\varphi_1}(x,a)\right] + \mathbb{E}_{\rho^\kappa_{M_{\theta_t}}} \left[\log(1 - D_{\varphi_1}(x,a))\right],
\end{aligned}
$$

(21)

where $A_{\varphi_0^*,\varphi_1^*}(x,a,x') = \log D_{\varphi_0^*}(x,a,x') - \log D_{\varphi_1^*}(x,a)$, $\mathbb{E}_{\rho^\kappa_{M_{\theta_t}}}[\cdot]$ is a simplification of $\mathbb{E}_{x,a,x' \sim \rho^\kappa_{M_{\theta_t}}}[\cdot]$.

$\square$

# B    DISCUSSION OF THE THEORETICAL RESULTS

We summarize the limitations of current theoretical results and future work as follows:

1. As discussed in Remark A.2, the solution Eq. 11 relies on $\rho^\beta_{M^*}(x,a) \in [0,c], \forall a \in \mathcal{A}, \forall x \in \mathcal{X}$. In some particular $M^*$, it is intractable to derive a $\beta$ that can generate an occupancy specified by $g(x,a)/\alpha_M$. If more knowledge of $M^*$ or $\beta^*$ is provided or some mild assumptions can be made on the properties of $M^*$ or $\beta^*$, we may model $\rho$ in a more sophisticated approach to alleviating the above problem.

2. In the tractable solution derivation, we ignore the term $\alpha_0(x,a) = \alpha_{M_{\theta_t}} \rho^\kappa_{M^*}(x,a)$ (See Eq. 20). The benefit is that $\rho^\kappa_{M^*}(x,a,x')$ in the tractable solution can be estimated through offline datasets directly. Although the results in our experiments show that it does not produce significant negative effects in these tasks, ignoring $\rho^\kappa_{M^*}(x,a)$ indeed incurs extra bias in theory. In future work, techniques for estimating $\rho^\kappa_{M^*}(x,a)$ (Liu et al., 2020) can be incorporated to correct the bias. On the other hand, $\alpha_{M_{\theta_t}}$ is also ignored in the process. $\alpha_{M_{\theta_t}}$ can be regarded as a global rescaling term of the final objective Eq. 20. Intuitively, it constructs an adaptive learning rate for Eq. 20, which increases the step size when the model is better fitted and decreases the step size otherwise. It can be considered to further improve the learning process in future work, e.g., cooperating with empirical risk minimization by balancing the weights of the two objectives through $\alpha_{M_{\theta_t}}$.

# C    SOCIETAL IMPACT

This work studies a method toward counterfactual environment model learning. Reconstructing an accurate environment of the real world will promote the wide adoption of decision-making policy optimization methods in real life, enhancing our daily experience. We are aware that decision-making

policy in some domains like recommendation systems that interact with customers may have risks of causing price discrimination and misleading customers if inappropriately used. A promising way to reduce the risk is to introduce fairness into policy optimization and rules to constrain the actions (Also see our policy design in Sec. G.1.3). We are involved in and advocating research in such directions. We believe that business organizations would like to embrace fair systems that can ultimately bring long-term financial benefits by providing a better user experience.

## D   AWRM-ORACLE PSEUDOCODE

We list the pseudocode of AWRM-oracle in Alg. 1.

---

**Algorithm 1** AWRM with Oracle Counterfactual Datasets

---
**Input**:
$\Phi$: policy space; $N$: total iterations
**Process**:

1: Generate counterfactual datasets $\{\mathcal{D}_{\pi_\phi}\}$ for all adversarial policies $\pi_\phi, \phi \in \Phi$
2: Initialize an environment model $M_\theta$
3: **for** i = 1:N **do**
4:     Select $\mathcal{D}_{\pi_\phi}$ with worst prediction errors through $M_\theta$ from $\{\mathcal{D}_{\pi_\phi}\}$
5:     Optimize $M_\theta$ with standard supervised learning based on $\mathcal{D}_{\pi_\phi}$
6: **end for**

---

## E   IMPLEMENTATION

### E.1   DETAILS OF THE GALILEO IMPLEMENTATION

The approximation of Eq. 18 holds only when $\frac{p(x)}{q(x)}$ is close to 1, which might not be satisfied. To handle the problem, we inject a standard supervised learning loss

$$\arg\max_\theta \mathbb{E}_{\rho_{M^*}^\kappa} \left[ \log M_\theta(x'|x, a) \right] \tag{22}$$

to replace the second term of the above objective when the output probability of $D$ is far away from $0.5$ ($f'(1) = \log 0.5$).

In the offline model-learning setting, we only have a real-world dataset $\mathcal{D}$ collected by the behavior policy $\mu$. We learn a policy $\hat\mu \approx \mu$ via behavior cloning with $\mathcal{D}$ (Pomerleau, 1991; Ho & Ermon, 2016) and let $\hat\mu$ be the policy $\kappa$. We regard $\mathcal{D}$ as the empirical data distribution of $\rho_{M^*}^\kappa$ and the trajectories collected by $\hat\mu$ in the model $M_{\theta_t}$ as the empirical data distribution of $\rho_{M_{\theta_t}}^\kappa$. But the assumption $\forall x \in \mathcal{X}, \forall a \in \mathcal{A}, \mu(a|x) > 0$ might not be satisfied. In behavior cloning, we model $\hat\mu$ with a Gaussian distribution and constrain the lower bound of the variance with a small value $\epsilon_\mu > 0$ to keep the assumption holding. Besides, we add small Gaussian noises $\mathbf{u} \sim \mathcal{N}(0, \epsilon_D)$ to the inputs of $D_\varphi$ to handle the mismatch between $\rho_{M^*}^\mu$ and $\rho_{M^*}^{\hat\mu}$ due to $\epsilon_\mu$. In particular, for $\varphi_0$ and $\varphi_1$ learning, we have:

$$\varphi_0^* = \arg\max_{\varphi_0} \mathbb{E}_{\rho_{M^*}^\kappa, \mathbf{u}} \left[ \log D_{\varphi_0}(x + u_x, a + u_a, x' + u_{x'}) \right] + \mathbb{E}_{\rho_{M_{\theta_t}}^\kappa, \mathbf{u}} \left[ \log(1 - D_{\varphi_0}(x + u_x, a + u_a, x' + u_{x'})) \right]$$

$$\varphi_1^* = \arg\max_{\varphi_1} \mathbb{E}_{\rho_{M^*}^\kappa, \mathbf{u}} \left[ \log D_{\varphi_1}(x + u_x, a + u_a) \right] + \mathbb{E}_{\rho_{M_{\theta_t}}^\kappa, \mathbf{u}} \left[ \log(1 - D_{\varphi_1}(x + u_x, a + u_a)) \right],$$

where $\mathbb{E}_{\rho_{M_{\theta_t}}^\kappa, \mathbf{u}}[\cdot]$ is a simplification of $\mathbb{E}_{x,a,x' \sim \rho_{M_{\theta_t}}^\kappa, \mathbf{u} \sim \mathcal{N}(0, \epsilon_D)}[\cdot]$ and $\mathbf{u} = [u_x, u_a, u_{x'}]$.

On the other hand, we notice that the first term in Eq. 21 is similar to the objective of GAIL (Ho & Ermon, 2016) by regarding $M_\theta$ as the policy to learn and $\kappa$ as the environment to generate data. For better capability in sequential environment model learning, here we introduce some practical tricks inspired by GAIL for model learning (Shi et al., 2019; Shang et al., 2019): we introduce an MDP for $\kappa$ and $M_\theta$, where the reward is defined by the discriminator $D$, i.e., $r(x, a, x') = \log D(x, a, x')$. $M_\theta$ is learned to maximize the cumulative rewards. With advanced policy gradient methods (Schulman et al., 2015; 2017), the objective is converted to $\max_\theta \left[ A_{\varphi_0^*, \varphi_1^*}(x, a, x') \log M_\theta(x, a, x') \right]$, where $A = Q_{M_{\theta_t}}^\kappa - V_{M_{\theta_t}}^\kappa$,

---

**Algorithm 2** GALILEO pseudocode

---

**Input**:

$\mathcal{D}_{\text{real}}$: offline dataset sampled from $\rho^{\mu}_{M^*}$ where $\mu$ is the behavior policy;

$N$: total iterations;

**Process**:

1: Approximate a behavior policy $\hat{\mu}$ via behavior cloning
2: Initialize an environment model $M_{\theta_1}$
3: **for** $t = 1 : N$ **do**
4:     Use $\hat{\mu}$ to generate a dataset $\mathcal{D}_{\text{gen}}$ with the model $M_{\theta_t}$
5:     Update the discriminators $D_{\varphi_0}$ and $D_{\varphi_1}$ via Eq. 26 and Eq. 27 respectively, where $\rho^{\hat{\mu}}_{M_{\theta_t}}$ is estimated by $\mathcal{D}_{\text{gen}}$ and $\rho^{\mu}_{M^*}$ is estimated by $\mathcal{D}_{\text{real}}$
6:     Update $Q$ and $V$ via Eq. 24 and Eq. 25 through $\mathcal{D}_{\text{gen}}$, $D_{\varphi_0}$, and $D_{\varphi_1}$
7:     Update the model $M_{\theta_t}$ via the first term of Eq. 23, which is implemented with a standard policy gradient method like TRPO (Schulman et al., 2015) or PPO (Schulman et al., 2017). Record the policy gradient $g_{\text{pg}}$
8:     **if** $p_0 < \mathbb{E}_{\mathcal{D}_{\text{gen}}}\left[D_{\varphi_0}(x_t, a_t, x_{t+1})\right] < p_1$ **then**
9:         Compute the gradient of $M_{\theta_t}$ via the second term of Eq. 23 and record it as $g_{\text{sl}}$
10:     **else**
11:         Compute the gradient of $M_{\theta_t}$ via Eq. 22 and record it as $g_{\text{sl}}$
12:     **end if**
13:     Rescale $g_{\text{sl}}$ via Eq. 28
14:     Update the model $M_{\theta_t}$ via the gradient $g_{\text{sl}}$ and obtain $M_{\theta_{t+1}}$
15: **end for**

---

$Q^{\kappa}_{M_{\bar{\theta}}}(x, a, x') = \mathbb{E}\left[\sum_{t=0}^{\infty} \gamma^t r(x_t, a_t, x_{t+1}) \mid (x_t, a_t, x_{t+1}) = (x, a, x'), \kappa, M_{\theta_t}\right]$, and $V^{\kappa}_{M_{\bar{\theta}}}(x, a) = \mathbb{E}_{M_{\bar{\theta}}}\left[Q^{\kappa}_{M_{\bar{\theta}}}(x, a, x')\right]$. $A$ in Eq. 21 can also be constructed similarly. Although it looks unnecessary in theory since the one-step optimal model $M_{\theta}$ is the global optimal model in this setting, the technique is helpful in practice as it makes $A$ more sensitive to the compounding effect of one-step prediction errors: we would consider the cumulative effects of prediction errors induced by multi-step transitions in environments. In particular, to consider the cumulative effects of prediction errors induced by multi-step of transitions in environments, we overwrite function $A_{\varphi_0^*, \varphi_1^*}$ as $A_{\varphi_0^*, \varphi_1^*} = Q^{\kappa}_{M_{\theta_t}} - V^{\kappa}_{M_{\theta_t}}$, where $Q^{\kappa}_{M_{\theta_t}}(x, a, x') = \mathbb{E}\left[\sum_t^{\infty} \gamma^t \log D_{\varphi_0^*}(x_t, a_t, x_{t+1}) \mid (x_t, a_t, x_{t+1}) = (x, a, x'), \kappa, M_{\theta_t}\right]$ and $V^{\kappa}_{M_{\theta_t}}(x, a) = \mathbb{E}\left[\sum_t^{\infty} \gamma^t \log D_{\varphi_1^*}(x_t, a_t) \mid (x_t, a_t) = (x, a), \kappa, M_{\theta_t}\right]$. To give an algorithm for single-step environment model learning, we can just set $\gamma$ in $Q$ and $V$ to 0.

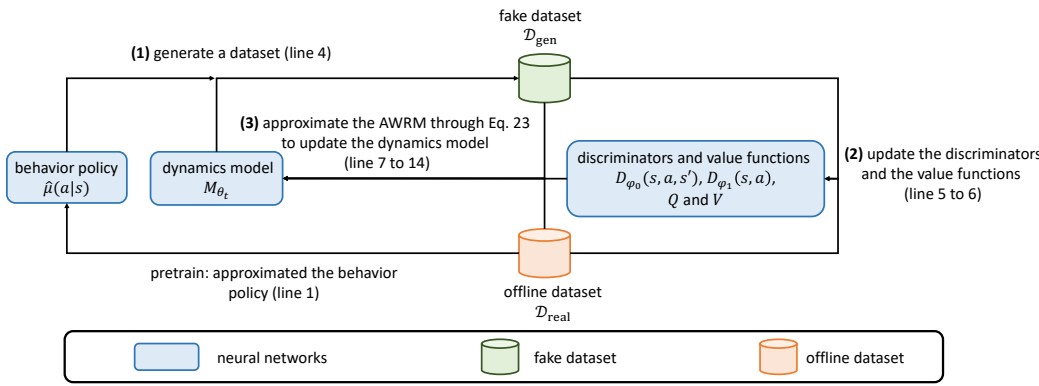

Figure 7: Illustration of the workflow of the GALILEO algorithm.

By adopting the above implementation techniques, we convert the objective into the following formulation

$$\theta_{t+1} = \arg \max_{\theta} \mathbb{E}_{\rho^{\kappa}_{M_{\theta_t}}} \left[ A_{\varphi_0^*, \varphi_1^*}(x, a, x') \log M_{\theta}(x'|x, a) \right] + \mathbb{E}_{\rho^{\kappa}_{M^*}} \left[ (H_{M^*}(x, a) - A_{\varphi_0^*, \varphi_1^*}(x, a, x')) \log M_{\theta}(x'|x, a) \right]$$
(23)

$$s.t. \quad Q^{\kappa}_{M_{\theta_t}}(x, a, x') = \mathbb{E} \left[ \sum_{t}^{\infty} \gamma^t \log D_{\varphi_0^*}(x_t, a_t, x_{t+1}) | (x_t, a_t, x_{t+1}) = (x, a, x'), \kappa, M_{\theta_t} \right] \quad (24)$$

$$V^{\kappa}_{M_{\theta_t}}(x, a) = \mathbb{E} \left[ \sum_{t}^{\infty} \gamma^t \log D_{\varphi_1^*}(x_t, a_t) | (x_t, a_t) = (x, a), \kappa, M_{\theta_t} \right] \quad (25)$$

$$\varphi_0^* = \arg \max_{\varphi_0} \mathbb{E}_{\rho^{\kappa}_{M^*}, \mathbf{u}} \left[ \log D_{\varphi_0}(x + u_x, a + u_a, x' + u_{x'}) \right] + \mathbb{E}_{\rho^{\kappa}_{M_{\theta_t}}, \mathbf{u}} \left[ \log(1 - D_{\varphi_0}(x + u_x, a + u_a, x' + u_{x'})) \right]$$
(26)

$$\varphi_1^* = \arg \max_{\varphi_1} \mathbb{E}_{\rho^{\kappa}_{M^*}, \mathbf{u}} \left[ \log D_{\varphi_1}(x + u_x, a + u_a) \right] + \mathbb{E}_{\rho^{\kappa}_{M_{\theta_t}}, \mathbf{u}} \left[ \log(1 - D_{\varphi_1}(x + u_x, a + u_a)) \right],$$
(27)

where $A_{\varphi_0^*, \varphi_1^*}(x, a, x') = Q^{\kappa}_{M_{\theta}}(x, a, x') - V^{\kappa}_{M_{\theta}}(x, a)$. In practice, GALILEO optimizes the first term of Eq. 23 with conservative policy gradient algorithms (e.g., PPO (Schulman et al., 2017) or TRPO (Schulman et al., 2015)) to avoid unreliable gradients for model improvements. Eq. 26 and Eq. 27 are optimized with supervised learning. The second term of Eq. 23 is optimized with supervised learning with a re-weighting term $-A_{\varphi_0^*, \varphi_1^*} + H_{M^*}$. Since $H_{M^*}$ is unknown, we use $H_{M_{\theta}}$ to estimate it. When the mean output probability of a batch of data is larger than $0.6$ or small than $0.4$, we replace the second term of Eq. 23 with a standard supervised learning in Eq. 22. Besides, unreliable gradients also exist in the process of optimizing the second term of Eq. 23. In our implementation, we use the scale of policy gradients to constrain the gradients of the second term of Eq. 23. In particular, we first compute the $l_2$-norm of the gradient of the first term of Eq. 23 via conservative policy gradient algorithms, named $||g_{pg}||_2$. Then we compute the $l_2$-norm of the gradient of the second term of Eq. 23, name $||g_{sl}||_2$. Finally, we rescale the gradients of the second term $g_{sl}$ by

$$g_{sl} \leftarrow g_{sl} \frac{||g_{pg}||_2}{\max\{||g_{pg}||_2, ||g_{sl}||_2\}}. \quad (28)$$

For each iteration, Eq. 23, Eq. 26, and Eq. 27 are trained with certain steps (See Tab. 5) following the same framework as GAIL. Based on the above techniques, we summarize the pseudocode of GALILEO in Alg. 2, where $p_0$ and $p_1$ are set to $0.4$ and $0.6$ in all of our experiments. The overall architecture is shown in Fig. 7.                                                              to Reviewer bcJS

### E.2 CONNECTION WITH PREVIOUS ADVERSARIAL ALGORITHMS

Standard GAN (Goodfellow et al., 2014) can be regarded as a partial implementation including the first term of Eq. 23 and Eq. 26 by degrading them into the single-step scenario. In the context of GALILEO, the objective of GAN is

$$\theta_{t+1} = \arg \max_{\theta} \mathbb{E}_{\rho^{\kappa}_{M_{\theta_t}}} \left[ A_{\varphi^*}(x, a, x') \log M_{\theta}(x'|x, a) \right]$$

$$s.t. \quad \varphi* = \arg \max_{\varphi} \mathbb{E}_{\rho^{\kappa}_{M^*}} \left[ \log D_{\varphi}(x, a, x') \right] + \mathbb{E}_{\rho^{\kappa}_{M_{\theta_t}}} \left[ \log(1 - D_{\varphi}(x, a, x')) \right],$$

where $A_{\varphi^*}(x, a, x') = \log D_{\varphi^*}(x, a, x')$. In the single-step scenario, $\rho^{\kappa}_{M_{\theta_t}}(x, a, x') = \rho_0(x)\kappa(a|x)M_{\theta_t}(x'|a, x)$. The term $\mathbb{E}_{\rho^{\kappa}_{M_{\theta_t}}} \left[ A_{\varphi^*}(x, a, x') \log M_{\theta}(x'|x, a) \right]$ can convert to $\mathbb{E}_{\rho^{\kappa}_{M_{\theta}}} \left[ \log D_{\varphi^*}(x, a, x') \right]$ by replacing the gradient of $M_{\theta_t}(x'|x, a)\nabla_{\theta} \log M_{\theta}(x'|x, a)$ with $\nabla_{\theta} M_{\theta}(x'|x, a)$ (Sutton & Barto, 1998). Previous algorithms like GANITE (Yoon et al., 2018) and SCIGAN (Bica et al., 2020) can be regarded as variants of the above training framework.

The first term of Eq. 23 and Eq. 26 are similar to the objective of GAIL by regarding $M_{\theta}$ as the "policy" to imitate and $\hat{\mu}$ as the "environment" to collect data. In the context of GALILEO, the

objective of GAIL is:

$$\theta_{t+1} = \arg\max_{\theta} \mathbb{E}_{\rho^{\kappa}_{M_{\theta_t}}} \left[ A_{\varphi^*}(x, a, x') \log M_{\theta}(x'|x, a) \right]$$

$$s.t. \quad Q^{\kappa}_{M_{\theta_t}}(x, a, x') = \mathbb{E} \left[ \sum_{t}^{\infty} \gamma^t \log D_{\varphi^*}(x_t, a_t, x_{t+1}) | (x_t, a_t, x_{t+1}) = (x, a, x'), \kappa, M_{\theta_t} \right]$$

$$\varphi^* = \arg\max_{\varphi} \mathbb{E}_{\rho^{\kappa}_{M^*}} \left[ \log D_{\varphi}(x, a, x') \right] + \mathbb{E}_{\rho^{\kappa}_{M_{\theta_t}}} \left[ \log(1 - D_{\varphi}(x, a, x')) \right],$$

where $A_{\varphi^*}(x, a, x') = Q^{\kappa}_{M_{\theta}}(x, a, x') - V^{\kappa}_{M_{\theta}}(x, a)$ and $V^{\kappa}_{M_{\theta_t}}(x, a) = \mathbb{E}_{M_{\theta_t}(x, a)} \left[ Q^{\kappa}(x, a, x') \right]$.

## F  ADDITIONAL RELATED WORK

to Reviewer j5p5 and rQ79

Our primitive objective is inspired by weighted empirical risk minimization (WERM) based on inverse propensity score (IPS). WERM is originally proposed to solve the generalization problem of domain adaptation in machine learning literature. For instance, we would like to train a predictor $M(y|x)$ in a domain with distribution $P_{\text{train}}(x)$ to minimize the prediction risks in the domain with distribution $P_{\text{test}}(x)$, where $P_{\text{test}} \neq P_{\text{test}}$. To solve the problem, we can train a weighted objective with $\max_M \mathbb{E}_{x \sim P_{\text{train}}}[\frac{P_{\text{test}}(x)}{P_{\text{train}}(x)} \log M(y|x)]$, which is called weighted empirical risk minimization methods (Ben-David et al., 2006; 2010; Cortes et al., 2010; Byrd & Lipton, 2019; Quinonero-Candela et al., 2008). These results have been extended and applied to causal inference, where the predictor is required to be generalized from the data distribution in observational studies (source domain) to the data distribution in randomized controlled trials (target domain) (Shimodaira, 2000; Assaad et al., 2021; Hassanpour & Greiner, 2019; Jung et al., 2020; Johansson et al., 2018). In this case, the input features include a state $x$ (a.k.a. covariates) and an action $a$ (a.k.a. treatment variable) which is sampled from a policy. We often assume the distribution of $x$, $P(x)$ is consistent between the source domain and the test domain, then we have $\frac{P_{\text{test}}(x)}{P_{\text{train}}(x)} = \frac{P(x)\beta(a|x)}{P(x)\mu(a|x)} = \frac{\beta(a|x)}{\mu(a|x)}$, where $\mu$ and $\beta$ are the policies in source and target domains respectively. In Shimodaira (2000); Assaad et al. (2021); Hassanpour & Greiner (2019), the policy in randomized controlled trials is modeled as a uniform policy, then $\frac{P_{\text{test}}(x)}{P_{\text{train}}(x)} = \frac{P(x)\beta(a|x)}{P(x)\mu(a|x)} = \frac{\beta(a|x)}{\mu(a|x)} \propto \frac{1}{\mu(a|x)}$. $\frac{1}{\mu(a|x)}$ is also known as inverse propensity score (IPS). In Johansson et al. (2018), it assumes that the policy in the target domain is predefined as $\beta(a|x)$ before environment model learning, then it uses $\frac{\beta}{\mu}$ as the IPS. The differences between AWRM and previous works are fallen in two aspects: (1) We consider the distribution-shift problem in the sequential decision-making scenario. In this scenario, we not only consider the action distribution mismatching between the behavior policy $\mu$ and the policy to evaluation $\beta$, but also the follow-up effects of policies to the state distribution; (2) For faithful offline policy optimization, we require the environment model to have generalization ability in numerous different policies. The objective of AWRM is proposed to guarantee the generalization ability of $M$ in numerous different policies instead of a specific policy.

On a different thread, there are also studies that bring counterfactual inference techniques of causal inference into model-based RL (Buesing et al., 2019; Pitis et al., 2020; Sontakke et al., 2021). These works consider that the transition function is relevant to some hidden noise variables and use Pearl-style structural causal models (SCMs), which is a directed acyclic graphs to define the causality of nodes in an environment, to handle the problem. SCMs can help RL in different ways: Buesing et al. (2019) approximate the posterior of the noise variables based on the observation of data, and environment models are learned based on the inferred noises. The generalization ability is improved if we can infer the correct value of the noise variables. Pitis et al. (2020) discover several local causal structural models of a global environment model, then data augmentation strategies by leveraging these local structures to generate counterfactual experiences. Sontakke et al. (2021) proposes a representation learning technique for causal factors, which is an instance of the hidden noise variables, in partially observable Markov decision processes (POMDPs). With the learned representation of causal factors, the performance of policy learning and transfer in downstream tasks will be improved. Instead of considering the hidden noise variables in the environments, our study considers the environment model learning problem in the fully observed setting and focuses on unbiased causal effect estimation in the offline dataset under behavior policies collected with selection bias.

to Reviewer j5p5

In offline model-based RL, the problem is called distribution shift (Yu et al., 2020; Levine et al., 2020; Chen et al., 2021) which has received great attentions. However, previous algorithms do not handle the model learning challenge directly but propose techniques to suppress policy sampling and learning in risky regions (Yu et al., 2020; Kidambi et al., 2020). Although these algorithms have made great progress in offline policy optimization in many tasks, so far, how to learn a better environment model in this scenario has rarely been discussed.

## G  EXPERIMENT DETAILS

### G.1  SETTINGS

#### G.1.1  GENERAL NEGATIVE FEEDBACK CONTROL (GNFC)

The design of GNFC is inspired by a classic type of scenario that behavior policies $\mu$ have selection bias and easily lead to counterfactual risks: For some internet platforms, we would like to allocate budgets to a set of targets (e.g., customers or cities) to increase the engagement of the targets in the platforms. Our task is to train a model to predict targets' feedback on engagement given targets' features and allocated budgets.

In these tasks, for better benefits, the online working policy (i.e., the behavior policy) will tend to cut down the budgets if targets have better engagement, otherwise, the budgets might be increased. The risk of counterfactual environment model learning in the task is that: the object with better historical engagement will be sent to smaller budgets because of the selection bias of the behavior policies, then the model might exploit this correlation for learning and get a conclusion that: increasing budgets will reduce the targets' engagement, which violates the real causality. We construct an environment and a behavior policy to mimic the above process. In particular, the behavior policy $\mu_{GNFC}$ is

$$\mu_{GNFC}(x) = \frac{(62.5 - \text{mean}(x))}{15} + \epsilon,$$

where $\epsilon$ is a sample noise, which will be discussed later. The environment includes two parts:

(1) response function $M_1(y|x, a)$:

$$M_1(y|x, a) = \mathcal{N}(\text{mean}(x) + a, 2)$$

(2) mapping function $M_2(x'|x, y)$:

$$M_2(x'|x, a, y) = y - \text{mean}(x) + x$$

The transition function $M^*$ is a composite of $M^*(x'|x, a) = M_2(x'|x, a, M_1(y|x, a))$. The behavior policies have selection bias: the actions taken are negatively correlated with the states, as illustrated in Fig. 8(a) and Fig. 8(b). We control the difficulty of distinguishing the correct causality of $x$, $a$, and $y$ by designing different strategies of noise sampling on $\epsilon$. In principle, with a larger number or more pronounced disturbances, there are more samples violating the correlation between $x$ and $a$, then more samples can be used to find the correct causality. Therefore, we can control the difficulty of counterfactual environment model learning by controlling the strength of disturbance. In particular, we sample $\epsilon$ from a uniform distribution $U(-e, e)$ with probability $p$. That is, $\epsilon = 0$ with probability $1 - p$ and $\epsilon \sim U(-e, e)$ with probability $p$. Then with larger $p$, there are more samples in the dataset violating the negative correlation (i.e., $\mu_{GNFC}$), and with larger $e$, the difference of the feedback will be more obvious. By selecting different $e$ and $p$, we can construct different tasks to verify the effectiveness and ability of the counterfactual environment model learning algorithm.

#### G.1.2  THE CANCER GENOMIC ATLAS (TCGA)

The Cancer Genomic Atlas (TCGA) is a project that has profiled and analyzed large numbers of human tumors to discover molecular aberrations at the DNA, RNA, protein, and epigenetic levels. The resulting rich data provide a significant opportunity to accelerate our understanding of the molecular basis of cancer. We obtain features, $\mathbf{x}$, from the TCGA dataset and consider three continuous treatments as done in SCIGAN (Bica et al., 2020). Each treatment, $a$, is associated with a set of parameters, $\mathbf{v}_1$, $\mathbf{v}_2$, $\mathbf{v}_3$, that are sampled randomly by sampling a vector from a standard normal distribution and scaling it with its norm. We assign interventions by sampling a treatment, $a$, from

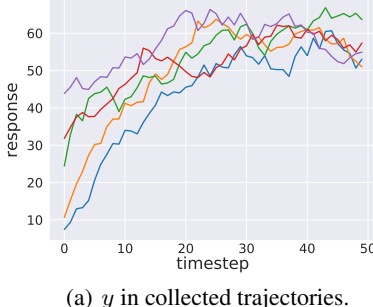
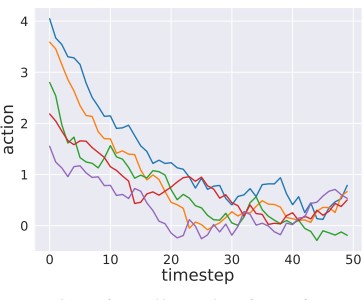

(a) $y$ in collected trajectories.

(b) $a$ in collected trajectories.

Figure 8: Illustration of information about the collected dataset in GNFC. Each color of the line denotes one of the collected trajectories. The X-axis denotes the timestep of a trajectory.

a beta distribution, $a \mid \mathbf{x} \sim \text{Beta}(\alpha, \beta)$. $\alpha \geq 1$ controls the sampling bias and $\beta = \frac{\alpha-1}{a^*} + 2 - \alpha$, where $a^*$ is the optimal treatment. This setting of $\beta$ ensures that the mode of $\text{Beta}(\alpha, \beta)$ is $a^*$.

The calculation of treatment response and optimal treatment are shown in Table 3.

Table 3: Treatment response used to generate semi-synthetic outcomes for patient features $\mathbf{x}$. In the experiments, we set $C = 10$.

| Treatment | Treatment Response | Optimal treatment |
|---|---|---|
| 1 | $f_1(\mathbf{x}, a_1) = C\left(\left(\mathbf{v}_1^1\right)^T \mathbf{x} + 12\left(\mathbf{v}_2^1\right)^T \mathbf{x} a_1 - 12\left(\mathbf{v}_3^1\right)^T \mathbf{x} a_1^2\right)$ | $a_1^* = \frac{\left(\mathbf{v}_2^1\right)^T \mathbf{x}}{2\left(\mathbf{v}_3^1\right)^T \mathbf{x}}$ |
| 2 | $f_2(\mathbf{x}, a_2) = C\left(\left(\mathbf{v}_1^2\right)^T \mathbf{x} + \sin\left(\pi\left(\frac{\mathbf{v}_2^{2T}\mathbf{x}}{\mathbf{v}_3^{2T}\mathbf{x}}\right) a_2\right)\right)$ | $a_2^* = \frac{\left(\mathbf{v}_3^2\right)^T \mathbf{x}}{2\left(\mathbf{v}_2^2\right)^T \mathbf{x}}$ |
| 3 | $f_3(\mathbf{x}, a_3) = C\left(\left(\mathbf{v}_1^3\right)^T \mathbf{x} + 12a_3(a_3 - b)^2\right)$, where $b = 0.75\frac{\left(\mathbf{v}_2^3\right)^T \mathbf{x}}{\left(\mathbf{v}_3^3\right)^T \mathbf{x}}$ | $\frac{3}{b}$ if $b \geq 0.75$, 1 if $b < 0.75$ |

We conduct experiments on three different treatments separately and change the value of bias $\alpha$ to assess the robustness of different methods to treatment bias. When the bias of treatment is large, which means $\alpha$ is large, the training set contains data with a strong bias on treatment so it would be difficult for models to appropriately predict the treatment responses out of the distribution of training data.

### G.1.3 BUDGET ALLOCATION TASK TO THE TIME PERIOD (BAT)

We deploy GALILEO in a real-world large-scale food-delivery platform. The platform contains various food stores, and food delivery clerks. The overall workflow is as follows: the platform presents the nearby food stores to the customers and the customers make orders, i.e., purchase take-out foods from some stores on the platform. The food delivery clerks can select orders from the platform to fulfill. After an order is selected to fulfill, the delivery clerks will take the ordered take-out foods from the stores and then send the food to the customers. The platform will pay the delivery clerks (mainly in proportion to the distance between the store and the customers' location) once the orders are fulfilled. An illustration of the workflow can be found in Fig. 9.

However, there is an imbalance problem between the demanded orders from customers and the supply of delivery clerks to fulfill these orders. For example, at peak times like lunchtime, there will be many more demanded orders than at other times, and the existed delivery clerks might not be able to fulfill all of these orders timely. The goal of the Budget Allocation task to the Time period (BAT) is to handle the imbalance problem in time periods by sending reasonable allowances to different time periods. More precisely, the goal of BAT is to make all orders (i.e., the demand) sent in different time periods can be fulfilled (i.e., the supply) timely.

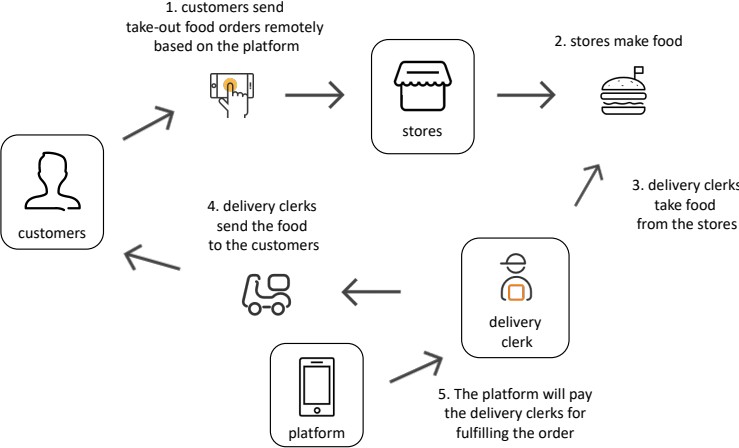

Figure 9: Illustration of the workflow of the food-delivery platform.

To handle the imbalance problem in different time periods, in the platform, the orders in different time periods $t \in [0, 1, 2..., 23]$ will be allocated with different allowances $c \in \mathcal{N}^+$. For example, at 10 A.M. (i.e., $t = 10$), we add 0.5\$ (i.e., $c = 0.5$) allowances to all of the demanded orders. From 10 A.M. to 11 A.M., the delivery clerks who take orders and send food to customers will receive extra allowances. Specifically, if the platform pays the delivery clerks 2\$ for fulfilling the order, now he/she will receive 2.5\$. For each day, the budget of allowance $C$ is fixed. We should find the best budget allocation policy $\pi^*(c|t)$ of the limited budget $C$ to make as many orders as possible can be taken timely.

To find the policy, we first learn a model to reconstruct the response of allowance for each delivery clerk $\hat{M}(y_{t+1}|s_t, p_t, c_t)$, where $y_{t+1}$ is the taken orders of the delivery clerks in state $s_t$, $c_t$ is the allowances, $p_t$ denotes static features of the time period $t$. In particular, the state $s_t$ includes historical order-taken information of the delivery clerks, current orders information, the feature of weather, city information, and so on. Then we use a rule-based mapping function $f$ to fill the complete next time-period states, i.e., $s_{t+1} = f(s_t, p_t, c_t, y_{t+1})$. Here we define the composition of the above functions $\hat{M}$ and $f$ as $\hat{M}_f$. Finally, we learn a budget allocation policy based on the learned model. For each day, the policy we would like to find is:

$$\max_{\pi} \mathbb{E}_{s_0 \sim \mathcal{S}} \left[ \sum_{t=0}^{23} y_t | \hat{M}_f, \pi \right],$$
$$\text{s.t.,} \sum_{t,s \in \mathcal{S}} c_t y_t \leq C$$

In our experiment, we evaluate the degree of balancing between demand and supply by computing the averaged five-minute order-taken rate, that is the percentage of orders picked up within five minutes. Note that the behavior policy is fixed for the long term in this application. So we directly use the data replay with a small scale of noise (See Tab. 5) to reconstruct the behavior policy for model learning in GALILEO.

**Also note that although we model the response for each delivery clerk, for fairness, the budget allocation policy is just determining the allowance of each time period $t$ and keeps the allowance to each delivery clerk $s$ the same.**

## G.2 BASELINE ALGORITHMS

The algorithm we compared are: (1) Supervised Learning (SL): training a environment model to minimize the expectation of prediction error, without considering the counterfactual risks; (2) inverse propensity weighting (IPW) (Spirtes, 2010): a practical way to balance the selection bias by re-weighting. It can be regarded as $\omega = \frac{1}{\hat{\mu}}$, where $\hat{\mu}$ is another model learned to approximate

the behavior policy; (3) SCIGAN: a recent proposed adversarial algorithm for model learning for continuous-valued interventions (Bica et al., 2020). All of the baselines algorithms are implemented with the same capacity of neural networks (See Tab. 5).

### G.2.1 SUPERVISED LEARNING (SL)

As a baseline, we train a multilayer perceptron model to directly predict the response of different treatments, without considering the counterfactual risks. We use mean square error to estimate the performance of our model so that the loss function can be expressed as $MSE = \frac{1}{n} \sum_{i=1}^{n} (y_i - \hat{y}_i)^2$, where $n$ is the number of samples, $y$ is the true value of response and $\hat{y}$ is the predicted response. In practice, we train our SL models using Adam optimizer and the initial learning rate $3e^{-4}$ on both datasets TCGA and GNFC. The architecture of the neural networks is listed in Tab. 5.

### G.2.2 INVERSE PROPENSITY WEIGHTING (IPW)

Inverse propensity weighting (Spirtes, 2010) is an approach where the treatment outcome model uses sample weights to balance the selection bias by re-weighting. The weights are defined as the inverse propensity of actually getting the treatment, which can be expressed as $\frac{1}{\hat{\mu}(a|x)}$, where $x$ stands for the feature vectors in a dataset, $a$ is the corresponding action and $\hat{\mu}(a|x)$ indicates the action taken probability of $a$ given the features $x$ within the dataset. $\hat{\mu}$ is learned with standard supervised learning. Standard IPW leads to large weights for the points with small sampling probabilities and finally makes the learning process unstable. We solve the problem by clipping the propensity score: $\hat{\mu} \leftarrow \min(\hat{\mu}, 0.05)$, which is common used in existing studies (Ionides, 2008). The loss function can thus be expressed as $\frac{1}{n} \sum_{i=1}^{n} \frac{1}{\hat{\mu}(a_i|x_i)} (y_i - \hat{y}_i)^2$. The architecture of the neural networks is listed in Tab. 5.

### G.2.3 SCIGAN

SCIGAN (Bica et al., 2020) is a model that uses generative adversarial networks to learn the data distribution of the counterfactual outcomes and thus generate individualized response curves. SCIGAN does not place any restrictions on the form of the treatment-does response functions and is capable of estimating patient outcomes for multiple treatments, each with an associated parameter. SCIGAN first trains a generator to generate response curves for each sample within the training dataset. The learned generator can then be used to train an inference network using standard supervised methods. For fair comparison, we increase the number of parameters for the the open-source version of SCIGAN so that the SCIGAN model can have same order of magnitude of network parameters as GALILEO. In addition, we also finetune the hyperparameters (Tab. 4) of the enlarged SCIGAN to realize its full strength. We set num_dosage_samples 9 and $\lambda = 10$.

Table 4: Table of hyper-parameters for SCIGAN.

| Parameter | Values |
|---|---|
| Number of samples | 3, 5, 7, 9, 11 |
| $\lambda$ | 0.1, 1, 10, 20 |

### G.3 HYPER-PARAMETERS

We list the hyper-parameter of GALILEO in Tab. 5.

### G.4 COMPUTATION RESOURCES

We use one Tesla V100 PCIe 32GB GPU and a 32-core Intel(R) Xeon(R) Gold 5118 CPU @ 2.30GHz to train all of our model.

Table 5: Table of hyper-parameters for all of the tasks.

| Parameter | GNFC | TAGC | MuJoCo | BAT |
|---|---|---|---|---|
| hidden layers of all neural networks | 4 | 4 | 5 | 5 |
| hidden units of all neural networks | 256 | 256 | 512 | 512 |
| collect samples for each time of model update | 5000 | 5000 | 40000 | 96000 |
| batch size of discriminators | 5000 | 5000 | 40000 | 80000 |
| horizon | 50 | 1 | 100 | 48 (half-hours) |
| $\epsilon_\mu$ (also $\epsilon_D$) | 0.005 | 0.01 | 0.05 (0.1 for walker2d) | 0.05 |
| times for discriminator update | 2 | 2 | 1 | 5 |
| times for model update | 1 | 1 | 2 | 20 |
| times for supervised learning update | 1 | 1 | 4 | 20 |
| learning rate for supervised learning | 1e-5 | 1e-5 | 3e-4 | 1e-5 |
| $\gamma$ | 0.99 | 0.0 | 0.99 | 0.99 |
| clip-ratio | NAN | NAN | NAN | 0.1 |
| max $D_{KL}$ | 0.001 | 0.001 | 0.001 | NAN |
| optimization algorithm (the first term of Eq. 23) | TRPO | TRPO | TRPO | PPO |

## H  ADDITIONAL RESULTS

### H.1  ALL OF THE RESULT TABLE

We give the result of CNFC in Tab. 8, TCGA in Tab. 9, BAT in Tab. 7, and MuJoCo in Tab. 6.

### H.2  AVERAGED RESPONSE CURVES

We give the averaged responses for all of the tasks and the algorithms in Fig. 16 to Fig. 23. We randomly select 20% of the states in the dataset and equidistantly sample actions from the action space for each sampled state, and plot the averaged predicted feedback of each action. The real response is slightly different among different figure as the randomly-selected states for testing is different. We sample 9 points in GNFC tasks and 33 points in TAGC tasks for plotting.

### H.3  ABLATION STUDIES

In Sec. 4.3 and Appx. E.1, we introduce several techniques to develop a practical GALILEO algorithm. Based on task `e0.2_p0.05` in GNFC, we give the ablation studies to investigate the effects of these techniques. We first compare two variants that do not handle the assumptions violation problems: (1) `NO_INJECT_NOISE`: set $\epsilon_\mu$ and $\epsilon_D$ to zero, which makes the overlap assumption not satisfied;; (2) `SINGLE_SL`: without replacing the second term in Eq. 8 with standard supervised learning even when the output probability of $D$ is far away from 0.5. Besides, we introduced several tricks inspired by GAIL and give a comparison of these tricks and GAIL: (3) `ONE_STEP`: use one-step reward instead of cumulative rewards (i.e., Q and V; see Eq. 24 and Eq. 25) for re-weighting, which is implemented by set $\gamma$ to 0; (4) `SINGE_DIS`: remove $T_{\varphi_1^*}(x,a)$ and replace it with $\mathbb{E}_{M_\theta}\left[T_{\varphi_0^*}(x,a,x')\right]$, which is inspired by GAIL that uses a value function as a baseline instead of using another discriminator; (5) `PURE_GAIL`:

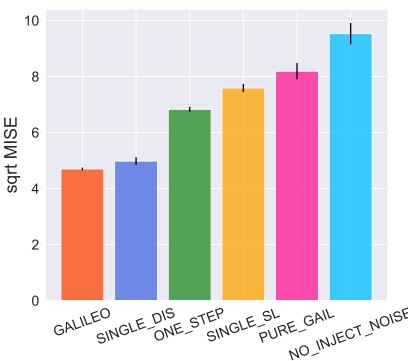

Figure 10: Illustration of the ablation studies. The error bars are the standard error.

remove the second term in Eq. 8. It can be regarded as a naive adoption of GAIL and a partial implementation of GALILEO.

We summarize the results in Fig. 10. Based on the results of `NO_INJECT_NOISE` and `SINGLE_SL`, we can see that handling the assumption violation problems is important and will increase the ability

Table 6: The root mean square errors on MuJoCo tasks. We bold the lowest error for each task. "medium" dataset **is used for training**, while "expert" and "medium-replay" datasets **are just used for testing**. $\pm$ follows the standard deviation of three seeds.

| TASK | HalfCheetah | | |
|---|---|---|---|
| DATASET | medium (train) | expert (test) | medium-replay (test) |
| GALILEO | **0.378** $\pm$ 0.003 | **2.287** $\pm$ 0.005 | **1.411** $\pm$ 0.037 |
| OFF-SL | 0.404 $\pm$ 0.001 | 3.311 $\pm$ 0.055 | 2.246 $\pm$ 0.016 |
| TASK | Walker2d | | |
| DATASET | medium (train) | expert (test) | medium-replay (test) |
| GALILEO | 0.49 $\pm$ 0.00 | **1.514** $\pm$ 0.002 | **0.968** $\pm$ 0.004 |
| OFF-SL | **0.467** $\pm$ 0.004 | 1.825 $\pm$ 0.061 | 1.239 $\pm$ 0.004 |
| TASK | Hopper | | |
| DATASET | medium (train) | expert (test) | medium-replay (test) |
| GALILEO | 0.037 $\pm$ 0.002 | **0.322** $\pm$ 0.036 | **0.408** $\pm$ 0.003 |
| OFF-SL | **0.034** $\pm$ 0.001 | 0.464 $\pm$ 0.021 | 0.574 $\pm$ 0.008 |

on counterfactual queries. The results of PURE_GAIL tell us that the partial implementation of GALILEO is not enough to give stable predictions on counterfactual data; On the other hand, the result of ONE_STEP also demonstrates that embedding the cumulative error of one-step prediction is helpful for GALILEO training; Finally, we also found that SINGLE_DIS nearly has almost no effect on the results. It suggests that, empirically, we can use $\mathbb{E}_{M_\theta}\left[T_{\varphi_0^*}(x,a,x')\right]$ as a replacement for $T_{\varphi_1^*}(x,a)$, which can reduce the computation costs of the extra discriminator training.

## H.4 WORST-CASE PREDICTION ERROR

In theory, GALILEO increases the generalization ability by focusing on the worst-case samples' training to achieve AWRM. To demonstrate the property, we propose a new metric named Mean-Max Square Error (MMSE): $\mathbb{E}\left[\max_{a\in\mathcal{A}}\left(M^*(x'|x,a) - M(x'|x,a)\right)^2\right]$ and give the results of MMSE for GNFC in Tab. 10 and for TCGA in Tab. 11.

## H.5 DETAILED RESULTS IN THE MUJOCO TASKS

We select 3 environments from D4RL (Fu et al., 2020) to construct our model learning tasks. We compare it with a typical transition model learning algorithm used in the previous offline model-based RL algorithms (Yu et al., 2020; Kidambi et al., 2020), which is a variant of standard supervised learning. We name the method OFF-SL. We train models in datasets HalfCheetah-medium, Walker2d-medium, and Hopper-medium, which are collected by a behavior policy with 1/3 performance to the expert policy, then we test them in the corresponding expert dataset. We plot the converged results and learning curves of GALILEO and OFF-SL in three MuJoCo tasks in Tab. 6 and Fig. 11 respectively.

In Fig. 11, we can see that both OFF-SL and GALILEO perform well in the training datasets. OFF-SL can even reach a bit lower error in halfcheetah and walker2d. However, when we verify the models through "expert" and "medium-replay" datasets, which are collected by other policies, the performance of GALILEO is significantly more stable and better than OFF-SL. As the training continues, OFF-SL even gets worse and worse. In summary, GALILEO reaches significantly better performances in the expert dataset: the averaged declines of root MSE in three environments are 56.5%, 49.2%, and 34.8%. However, whether in GALILEO or OFF-SL, the performance for testing is at least 2x worse than in the training dataset. The phenomenon indicates that although GALILEO can make better performances for counterfactual queries, the risks of using the models are still large and still challenging to be further solved.

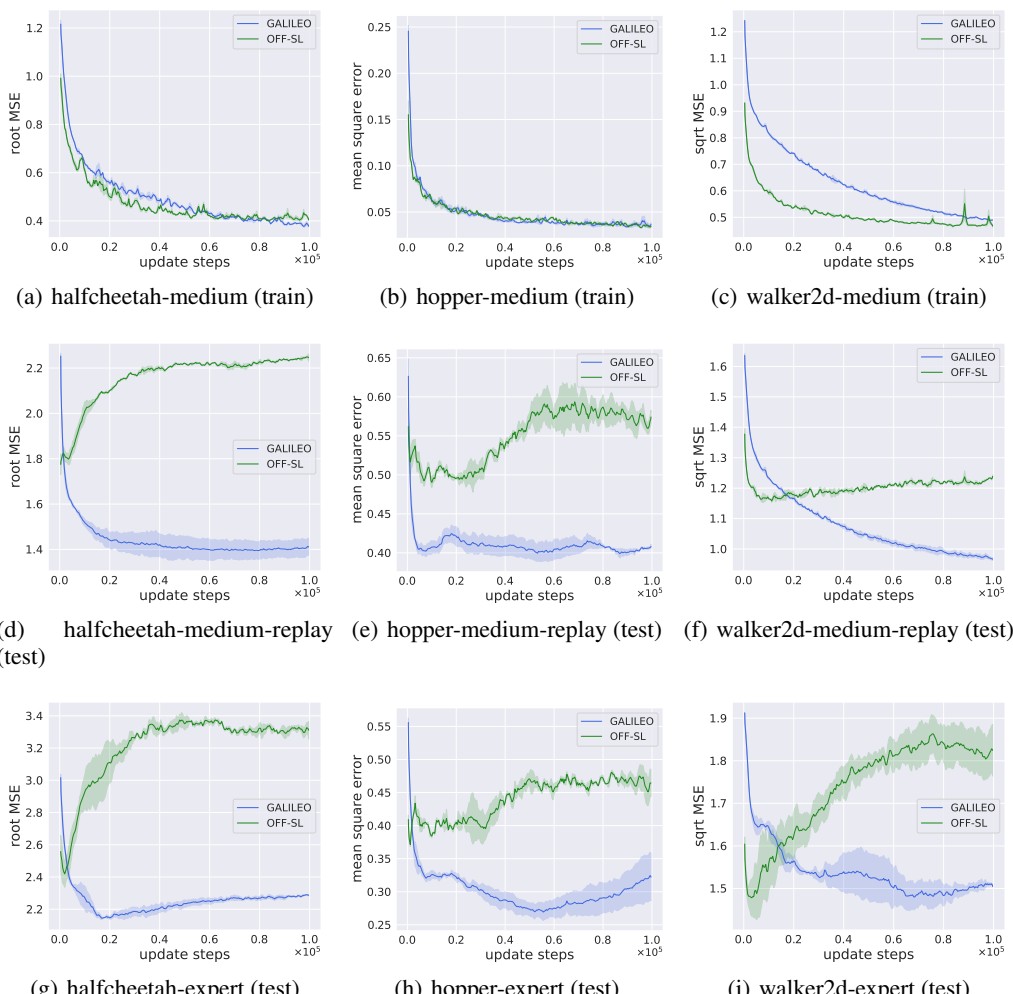

Figure 11: Illustration of learning curves of the MuJoCo Tasks. The X-axis record the steps of the environment model update, and the Y-axis is the corresponding prediction error. The figures with titles ending in "(train)" means the dataset is used for training while the titles ending in "(test)" means the dataset is *just used for testing*. The solid curves are the mean reward and the shadow is the standard error of three seeds.

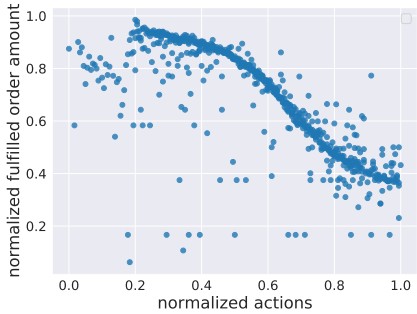

Figure 12: Illustration of relationship between user feedback and the actions of the offline dataset in the real-world food-delivery platform.

## H.6    DETAILED RESULTS IN THE BAT TASK

The core challenge of the environment model learning in BAT tasks is similar to the challenge in Fig. 1. Specifically, the behavior policy in BAT tasks is a human-expert policy, which will tend to increase the budget of allowance in the time periods with a lower supply of delivery clerks, otherwise will decrease the budget (Fig. 12 gives an instance of this phenomenon in the real data).

Since there is no oracle environment model for querying, we have to describe the results with other metrics.

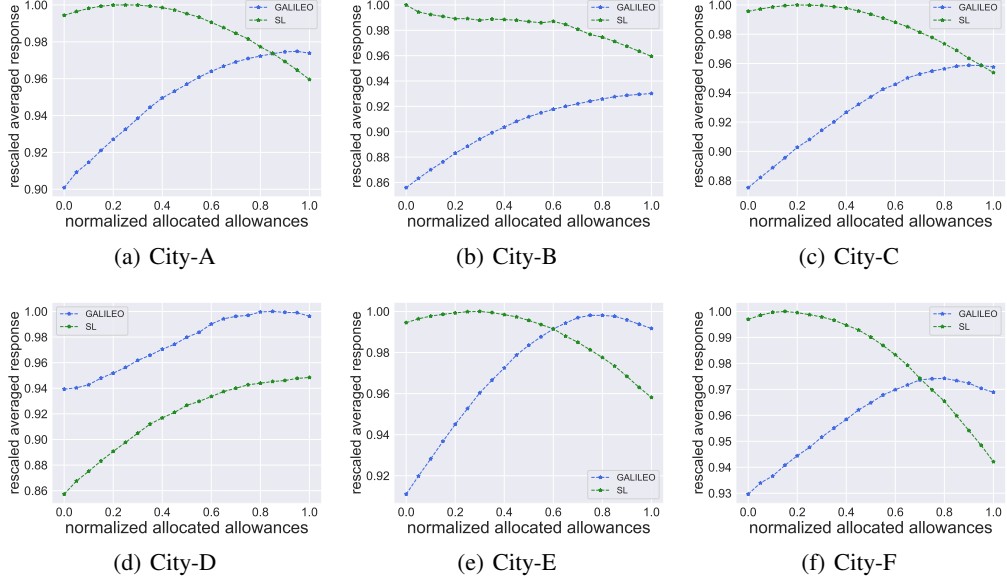

Figure 13: Illustration of the response curves in the 6 cities. Although the ground-truth curves are unknown, through human expert knowledge, *we know that it is expected to be monotonically increasing* .

First, we review whether the tendency of the response curve is consistent. In this application, with a larger budget of allowance, the supply will not be decreased. As can be seen in Fig. 13, the tendency of GALILEO's response is valid in 6 cities but almost all of the models of SL give opposite directions to the response. If we learn a policy through the model of SL, the optimal solution is canceling all of the allowances, which is obviously incorrect in practice.

Second, we conduct randomized controlled trials (RCT) in one of the testing cities. Using the RCT samples, we can evaluate the correctness of the sort order of the model predictions via Area Under

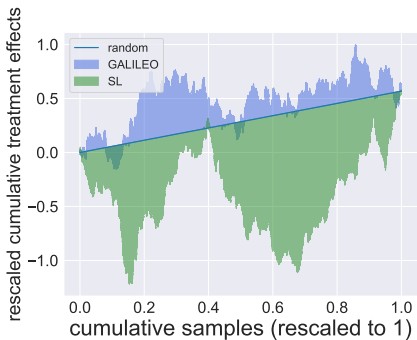

Figure 14: Illustration of the AUUC result for BAT.

the Uplift Curve (AUUC) (Betlei et al., 2020). To plot AUUC, we first sort the RCT samples based on the predicted treatment effects. Then the cumulative treatment effects are computed by scanning the sorted sample list. If the sort order of the model predictions is better, the sample with larger treatment effects will be computed early. Then the area of AUUC will be larger than the one via a random sorting strategy. The result of AUUC show GALILEO gives a reasonable sorting to the RCT samples (see Fig. 14).

Finally, we search for the optimal policy via the cross-entropy method planner (Hafner et al., 2019) based on the learned model. We test the online supply improvement in 6 cities. The algorithm compared is a human-expert policy, which is also the behavior policy of the offline datasets. We conduct online A/B tests for each of the cities. For each test, we randomly split a city into two partitions, one is for deploying the optimal policy learned from the GALILEO model, and the other is as a control group, which keeps the human-expert policy as before. Before the intervention, we collect 10 days' observation data and compute the averaged five-minute order-taken rates as the baselines of the treatment and control group, named $b^t$ and $b^c$ respectively. Then we start intervention and observe the five-minute order-taken rate in the following 14 days for the two groups. The results of the treatment and control groups are $y_i^t$ and $y_i^c$ respectively, where $i$ denotes the $i$-th day of the deployment. The percentage points of the supply improvement are computed via difference-in-difference (DID):

$$\frac{\sum_i^T (y_i^t - b^t) - (y_i^c - b^c)}{T} \times 100,$$

where $T$ is the total days of the intervention and $T = 14$ in our experiments.

Table 7: Results on BAT. We use City-X to denote the experiments on different cities. "pp" is an abbreviation of percentage points on the supply improvement.

| target | City-A | City-B | City-C |
|---|---|---|---|
| supply improvement | +1.63pp | +0.79pp | +0.27pp |

| target | City-D | City-E | City-F |
|---|---|---|---|
| supply improvement | +0.2pp | +0.14pp | +0.41pp |

The results are summarized in Tab. 7. The online experiment is conducted in 14 days and the results show that the policy learned with GALILEO can make better (the supply improvements are from **0.14 to 1.63** percentage points) budget allocation than the behavior policies in **all the testing cities**. We give detailed results which record the supply difference between the treatment group and the control group in Fig. 15.

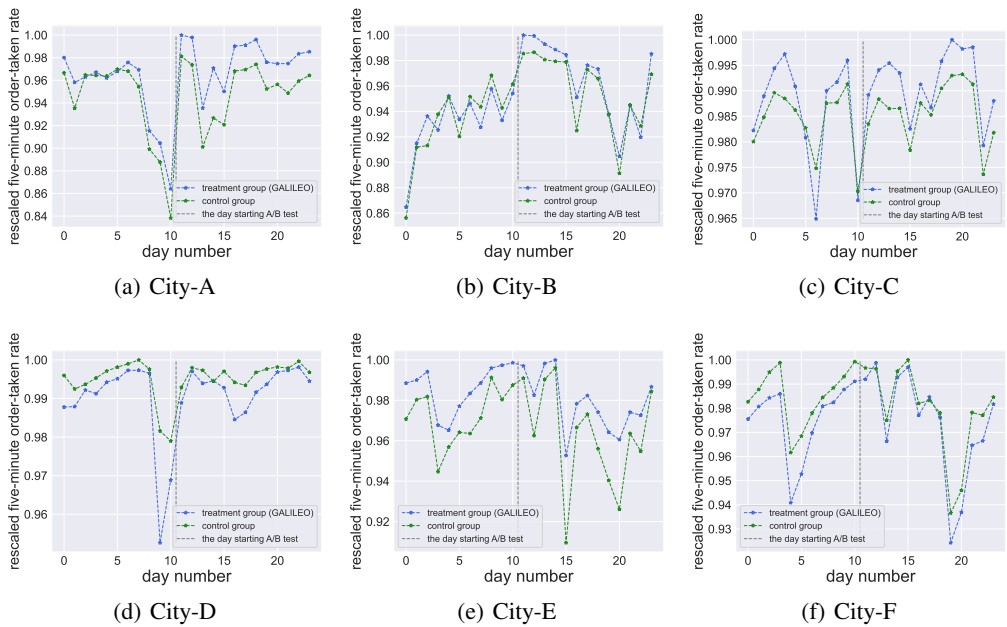

(a) City-A         (b) City-B         (c) City-C

(d) City-D         (e) City-E         (f) City-F

Figure 15: Illustration of the daily responses in the A/B test in the 6 cities.

Table 8: $\sqrt{MISE}$ results on GNFC. We bold the lowest error for each task. $\pm$ is the standard deviation of three random seeds.

|  | e1_p1 | e0.2_p1 | e0.05_p1 |
|---|---|---|---|
| GALILEO | $5.17 \pm 0.06$ | $\mathbf{4.73 \pm 0.13}$ | $\mathbf{4.70 \pm 0.02}$ |
| SL | $\mathbf{5.15 \pm 0.23}$ | $4.73 \pm 0.31$ | $23.64 \pm 4.86$ |
| IPW | $5.22 \pm 0.09$ | $5.50 \pm 0.01$ | $5.02 \pm 0.07$ |
| SCIGAN | $7.05 \pm 0.52$ | $6.58 \pm 0.58$ | $18.55 \pm 3.50$ |
|  | e1_p0.2 | e0.2_p0.2 | e0.05_p0.2 |
| GALILEO | $\mathbf{5.03 \pm 0.09}$ | $\mathbf{4.72 \pm 0.05}$ | $\mathbf{4.87 \pm 0.15}$ |
| SL | $5.21 \pm 0.63$ | $6.74 \pm 0.15$ | $33.52 \pm 1.32$ |
| IPW | $5.27 \pm 0.05$ | $5.69 \pm 0.00$ | $20.23 \pm 0.45$ |
| SCIGAN | $16.07 \pm 0.27$ | $12.07 \pm 1.93$ | $19.27 \pm 10.72$ |
|  | e1_p0.05 | e0.2_p0.05 | e0.05_p0.05 |
| GALILEO | $\mathbf{5.23 \pm 0.41}$ | $\mathbf{5.01 \pm 0.08}$ | $\mathbf{6.17 \pm 0.33}$ |
| SL | $5.89 \pm 0.88$ | $14.25 \pm 3.48$ | $37.50 \pm 2.29$ |
| IPW | $5.21 \pm 0.01$ | $5.52 \pm 0.44$ | $31.95 \pm 0.05$ |
| SCIGAN | $11.50 \pm 7.76$ | $13.05 \pm 4.19$ | $25.74 \pm 8.30$ |

Table 9: $\sqrt{MISE}$ results on TCGA. We bold the lowest error for each task. $\pm$ is the standard deviation of three random seeds.

|  | t0_bias_2.0 | t0_bias_20.0 | t0_bias_50.0 |
|---|---|---|---|
| GALILEO | $\mathbf{0.34 \pm 0.05}$ | $\mathbf{0.67 \pm 0.13}$ | $\mathbf{2.04 \pm 0.12}$ |
| SL | $0.38 \pm 0.13$ | $1.50 \pm 0.31$ | $3.06 \pm 0.65$ |
| IPW | $6.57 \pm 1.16$ | $6.88 \pm 0.30$ | $5.84 \pm 0.71$ |
| SCIGAN | $0.74 \pm 0.05$ | $2.74 \pm 0.35$ | $3.19 \pm 0.09$ |
|  | t1_bias_2.0 | t1_bias_6.0 | t1_bias_8.0 |
| GALILEO | $\mathbf{0.43 \pm 0.05}$ | $\mathbf{0.25 \pm 0.02}$ | $\mathbf{0.21 \pm 0.04}$ |
| SL | $0.47 \pm 0.05$ | $1.33 \pm 0.97$ | $1.18 \pm 0.73$ |
| IPW | $3.67 \pm 2.37$ | $0.54 \pm 0.13$ | $2.69 \pm 1.17$ |
| SCIGAN | $0.45 \pm 0.25$ | $1.08 \pm 1.04$ | $1.01 \pm 0.77$ |
|  | t2_bias_2.0 | t2_bias_6.0 | t2_bias_8.0 |
| GALILEO | $1.46 \pm 0.09$ | $\mathbf{0.85 \pm 0.04}$ | $\mathbf{0.46 \pm 0.01}$ |
| SL | $0.81 \pm 0.14$ | $3.74 \pm 2.04$ | $3.59 \pm 0.14$ |
| IPW | $2.94 \pm 1.59$ | $1.24 \pm 0.01$ | $0.99 \pm 0.06$ |
| SCIGAN | $\mathbf{0.73 \pm 0.15}$ | $1.20 \pm 0.53$ | $2.13 \pm 1.75$ |

Table 10: $\sqrt{MMSE}$ results on GNFC. We bold the lowest error for each task. $\pm$ is the standard deviation of three random seeds.

|  | e1_p1 | e0.2_p1 | e0.05_p1 |
|---|---|---|---|
| GALILEO | **3.86 ± 0.03** | **3.99 ± 0.01** | **4.07 ± 0.03** |
| SL | 5.73 ± 0.33 | 5.80 ± 0.28 | 18.78 ± 3.13 |
| IPW | 4.02 ± 0.05 | 4.15 ± 0.12 | 22.66 ± 0.33 |
| SCIGAN | 8.84 ± 0.54 | 12.62 ± 2.17 | 24.21 ± 5.20 |
|  | e1_p0.2 | e0.2_p0.2 | e0.05_p0.2 |
| GALILEO | **4.13 ± 0.10** | **4.11 ± 0.15** | **4.21 ± 0.15** |
| SL | 5.87 ± 0.43 | 7.44 ± 1.13 | 29.13 ± 3.44 |
| IPW | 4.12 ± 0.02 | 6.12 ± 0.48 | 30.96 ± 0.17 |
| SCIGAN | 12.87 ± 3.02 | 14.59 ± 2.13 | 24.57 ± 3.00 |
|  | e1_p0.05 | e0.2_p0.05 | e0.05_p0.05 |
| GALILEO | **4.39 ± 0.20** | **4.34 ± 0.20** | **5.26 ± 0.29** |
| SL | 6.12 ± 0.43 | 14.88 ± 4.41 | 30.81 ± 1.69 |
| IPW | 13.60 ± 7.83 | 26.27 ± 2.67 | 32.55 ± 0.12 |
| SCIGAN | 9.19 ± 1.04 | 15.08 ± 1.26 | 17.52 ± 0.02 |

Table 11: $\sqrt{MMSE}$ results on TCGA. We bold the lowest error for each task. $\pm$ is the standard deviation of three random seeds.

|  | t0_bias_2.0 | t0_bias_20.0 | t0_bias_50.0 |
|---|---|---|---|
| GALILEO | **1.56 ± 0.04** | **1.96 ± 0.53** | **3.16 ± 0.13** |
| SL | 1.92 ± 0.67 | 2.31 ± 0.19 | 5.11 ± 0.66 |
| IPW | 7.42 ± 0.46 | 5.36 ± 0.96 | 5.38 ± 1.24 |
| SCIGAN | 2.11 ± 0.47 | 5.23 ± 0.27 | 5.59 ± 1.02 |
|  | t1_bias_2.0 | t1_bias_6.0 | t1_bias_8.0 |
| GALILEO | 1.43 ± 0.06 | 1.09 ± 0.05 | **1.36 ± 0.36** |
| SL | **1.12 ± 0.15** | 3.65 ± 1.91 | 3.96 ± 1.81 |
| IPW | 1.14 ± 0.11 | **0.90 ± 0.09** | 2.04 ± 0.99 |
| SCIGAN | 3.32 ± 0.88 | 4.74 ± 2.12 | 5.17 ± 2.42 |
|  | t2_bias_2.0 | t2_bias_6.0 | t2_bias_8.0 |
| GALILEO | 3.77 ± 0.35 | 3.99 ± 0.40 | **2.08 ± 0.60** |
| SL | **2.70 ± 0.67** | 8.33 ± 5.05 | 9.70 ± 3.12 |
| IPW | 2.92 ± 0.15 | 3.90 ± 0.17 | 4.47 ± 2.16 |
| SCIGAN | 3.82 ± 2.12 | **1.83 ± 1.49** | 3.62 ± 4.9 |

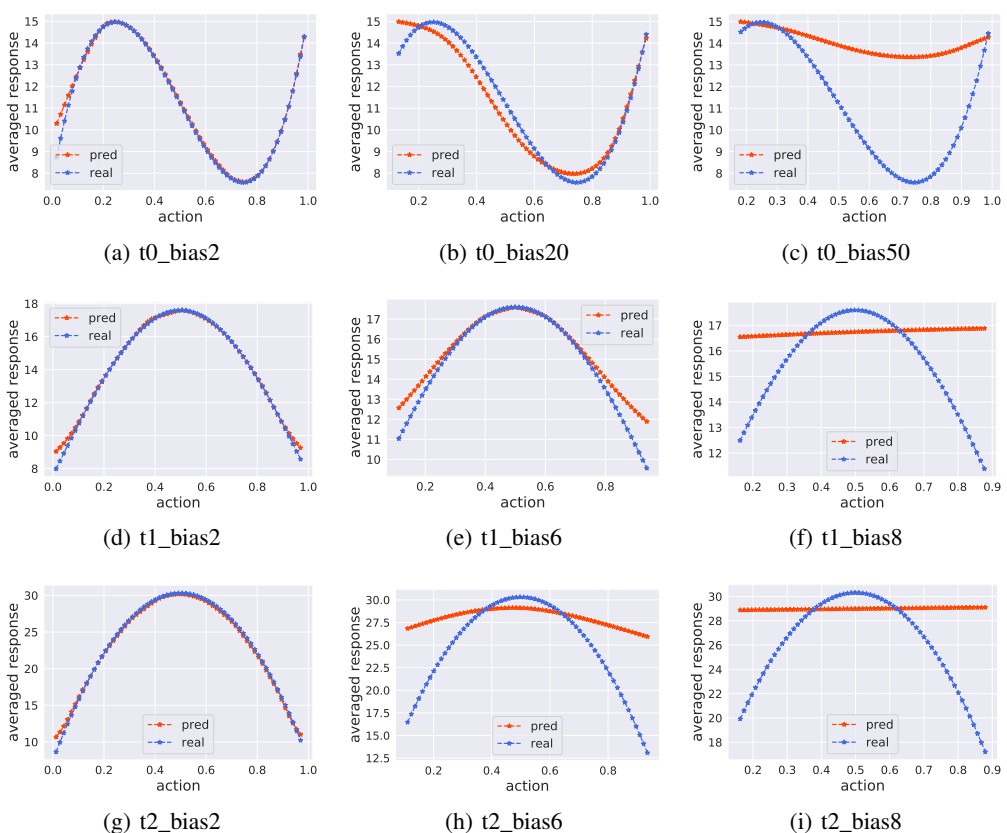

Figure 16: Illustration of the averaged response curves of Supervised Learning (SL) in TCGA.

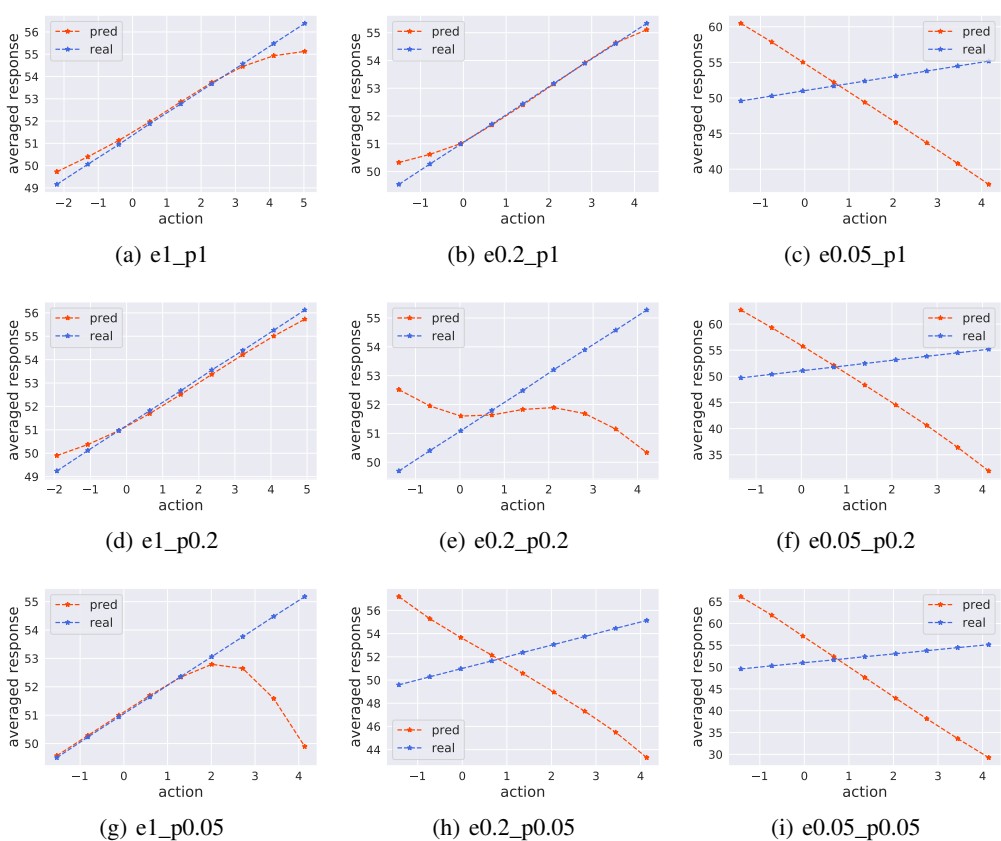

Figure 17: Illustration of the averaged response curves of Supervised Learning (SL) in GNFC.

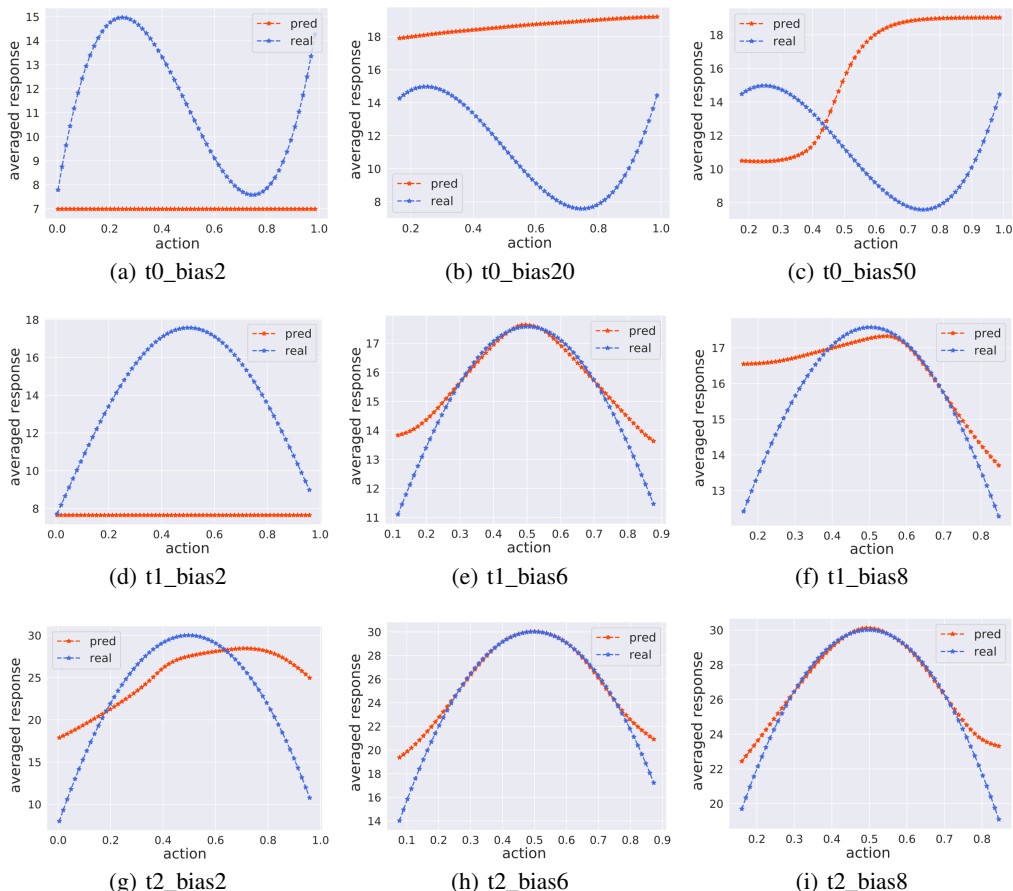

Figure 18: Illustration of the averaged response curves of Inverse Propensity Weighting (IPW) in TCGA.

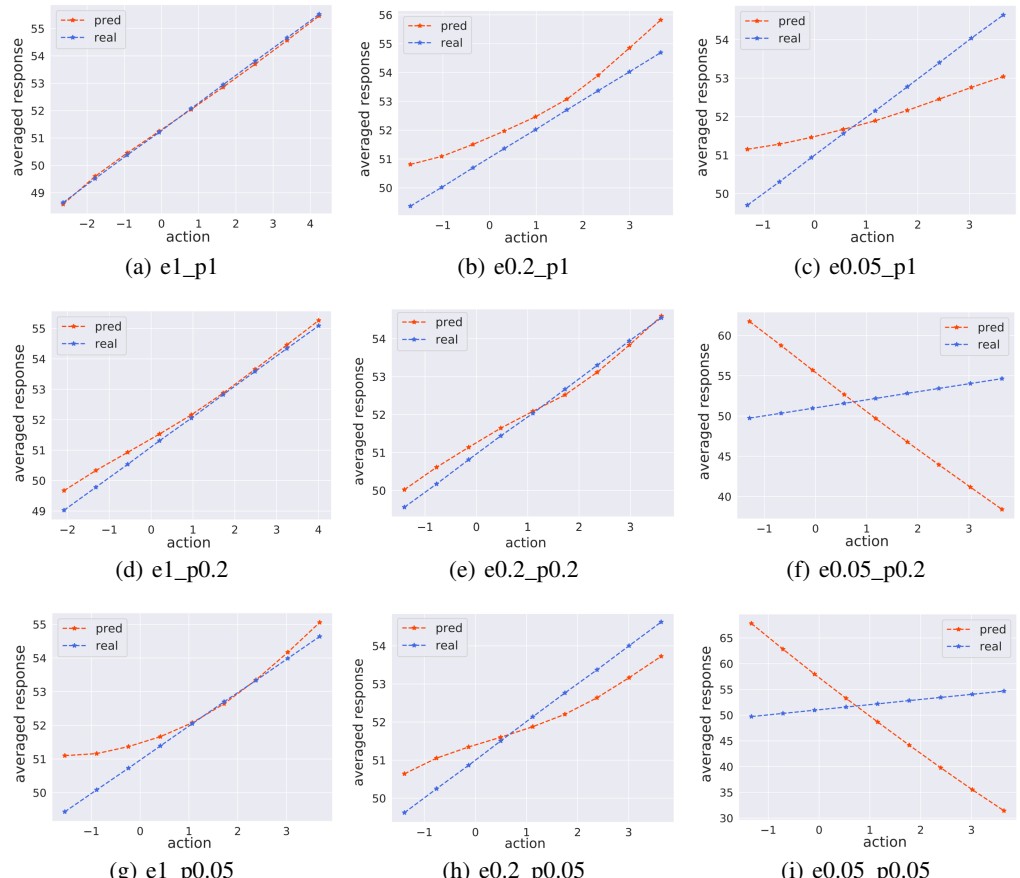

Figure 19: Illustration of the averaged response curves of Inverse Propensity Weighting (IPW) in GNFC.

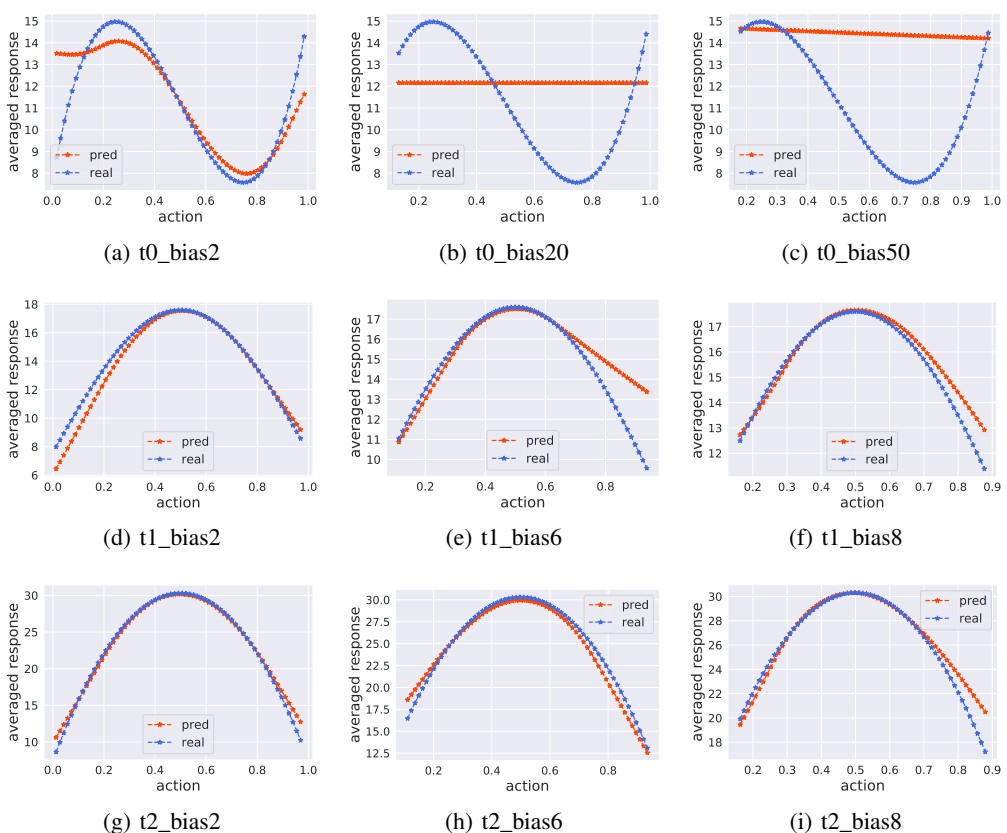

Figure 20: Illustration of the averaged response curves of SCIGAN in TCGA.

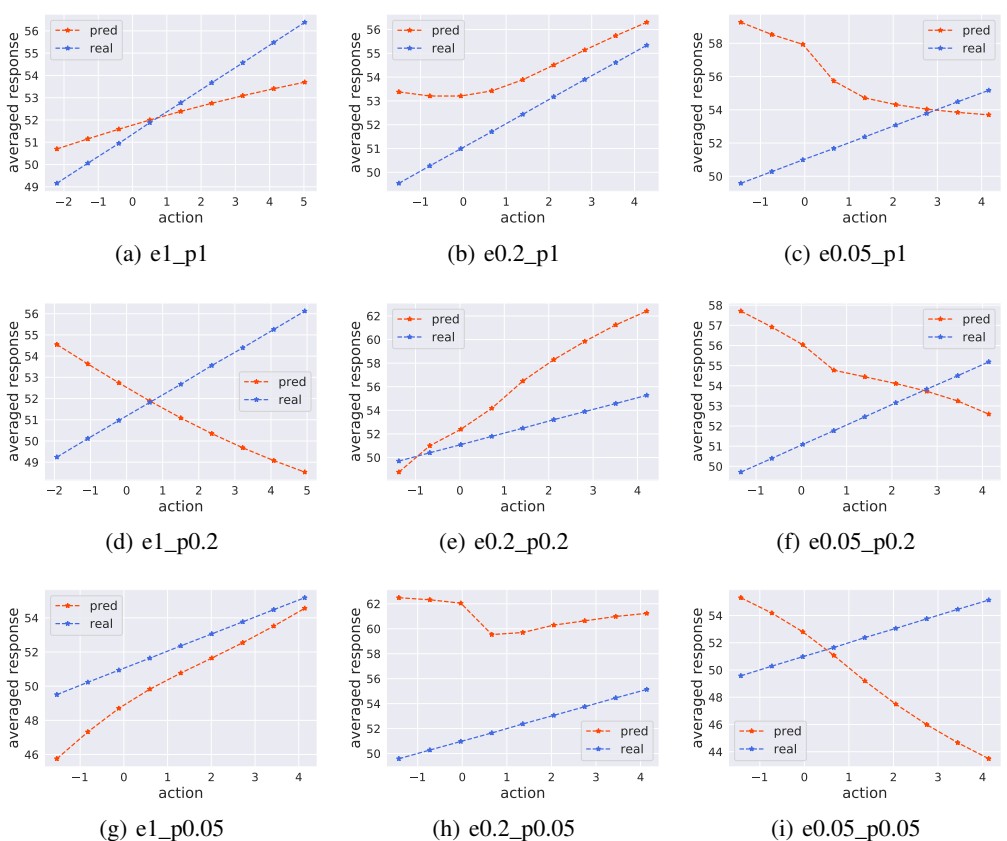

Figure 21: Illustration of the averaged response curves of SCIGAN in GNFC.

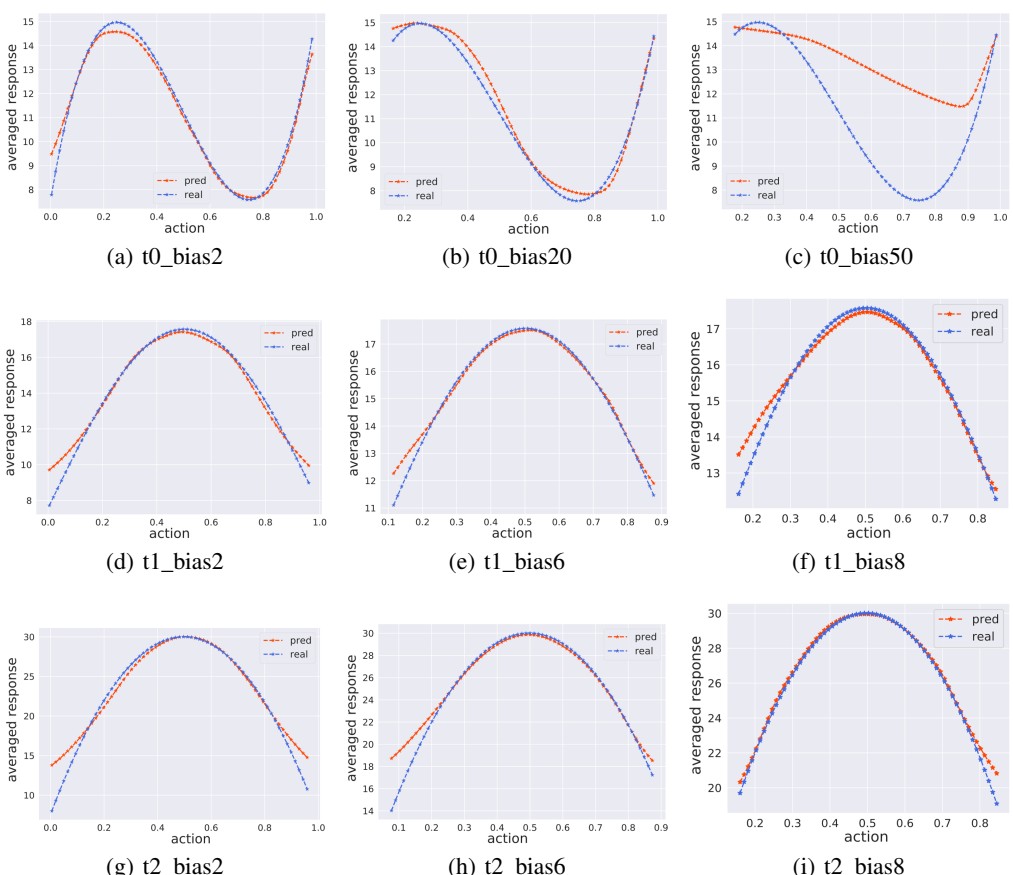

Figure 22: Illustration of the averaged response curves of GALILEO in TCGA.

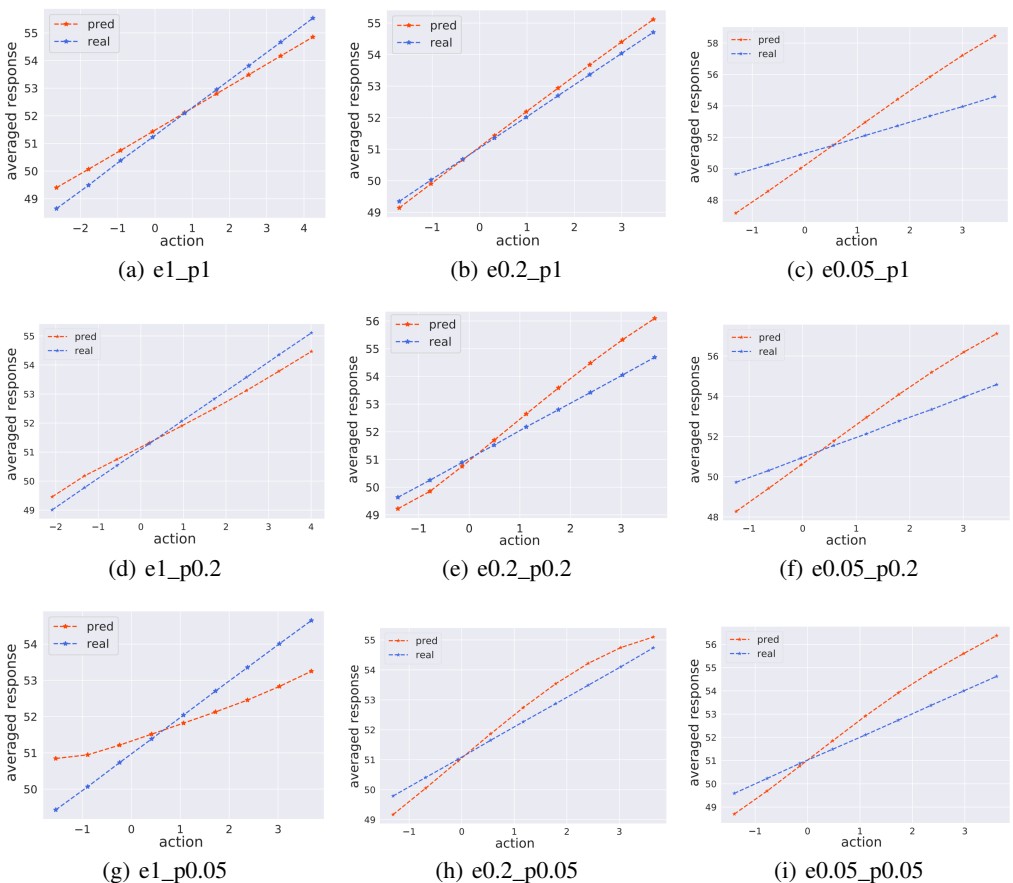

Figure 23: Illustration of the averaged response curves of GALILEO in GNFC.

