# OpenReview forum: "Adversarial Counterfactual Environment Model Learning"
_ICLR.cc/2023/Conference — Submitted to ICLR 2023_

### Official Review · Reviewer_rQ79 · 2022-10-23

**Confidence:** 3
**Correctness:** 1
**Technical Novelty And Significance:** 2
**Empirical Novelty And Significance:** 2
**Recommendation:** 3

**Clarity, Quality, Novelty And Reproducibility:**

Reproducibility: As an experiment, the authors deploy their policy in a "real-world large-scale food-delivery platform", which is clearly not reproducible.

**Strength And Weaknesses:**

Strengths:
- addresses an important problem
- applies model based RL to offline RL, a promising avenue of research
- applies adversarial worst-case ideas, which should in theory result in better

Major Concerns:

Clarity: It took a lot of reading and re-reading for me to understand what this paper is about. The main thing that made it hard for me to understand is that it's sold by the authors as a RL paper, when in fact their method is more a Domain Adaptation method, where they take data from one domain (observations of a transition function under one policy) and try to make predictions in another domain (behavior of a transition function under another policy). This is confusing, since what one is actually interested in is off policy evaluation, knowing how well a an new policy will work based on observation of an old policy.

Quality: I do not thing the authors claims are supported by experimental evidence. The problem with the presented experimental evidence is three-fold: First, they don't compare to other state of the art domain adaptation methods, even though it's a domain adaption paper. Second, their experiments only talk about prediction accuracy of the transition function. It's totally possible to predict accurately enough an environment, but be very inaccurate at predicting a policy's performance in the environment. Last, the authors claim their method's "policy significantly improves performance" but only show it for a non-reproducible setup. In RL, the whole point is to improve a policies performance, and the authors can't demonstrate that their method indeed improves performance.

To be frank, since the authors were unable, in my view, to support their claims with evidence I didn't carefully study every mathematical detail of their method.



**Summary Of The Paper:**

In this paper, the authors address the age old offline RL problem of counterfactual prediction, i.e. what would have happened if an unobserved action would have happened at an unobserved state. To do this, they use a weighted empirical risk minimization framework, which is extended with an adversarial framework to learn a model under a worst-case assumption.

They verify that their model makes accurate counterfactual predictions on unseen data by evaluating on D4RL and a propriety real-world food delivery task.

**Summary Of The Review:**

The authors fail to show their method is relevant for reinforcement learning, as the method merely learns a prediction of the transition function, but they don't show this leads to better policies or better off-policy evaluation.

---

> ### Author Response · Authors · 2022-11-16
> **Official Response to Reviewer rQ79 (Part I)**
>
> Many thanks for the reviewer's constructive comments on our paper.
>
> After repeatedly reading the whole review, we found that there seem to be several misunderstandings on the background, related work and the primitive objective of this study, which finally misleads the assessment of the reviewer on our work. We first clarify some concepts of this study as follows:
> ### clarification
> **(1) model-based RL and offline RL**
>
> > quote:
> > "one strength of this work is applying model based RL to offline RL"
>
> This is neither the first work that applies model-based RL to offline RL nor the contribution of this article.
>
> Offline model-based RL is an existing field with many powerful techniques [1,7,8]. The selection bias can be regarded as an instance of the problem called ``distributional shift'' in offline model-based RL, which has also received great attention [2,3,4].  However, previous methods where *naive supervised learning* is used for environment model learning, ignore the selection-bias problem in environment model learning and handle the problem by suppressing the policy exploration and learning in risky regions. Although these methods have made great progress in many tasks, so far, how to learn a better environment model that can alleviate the selection bias remains open.
>
>  We point out that this paper discloses, for the first time in offline RL, that the supervised-learned model can lead to a wrong learning direction. This issue was completely missed in previous studies. This is also one of the reasons why we focus on developing and evaluating the environment model learning technique. The related description is in Appx. F in the previous version, and we have moved them to Sec.1 in the revised paper (refer to the red lines).

---

> > ### Author Response · Authors · 2022-11-16
> > **Official Response to Reviewer rQ79 (Part II)**
> >
> >
> >
> > **(2) relation between the model error and the value gap of policies.**
> >
> > > quote:
> > >
> > > "The authors fail to show their method is relevant for reinforcement learning, as the method merely learns a prediction of the transition function, the method merely learns a prediction of the transition function, but they don't show this leads to better policies or better off-policy evaluation"
> > >
> > > "I do not think the authors claims are supported by experimental evidence, ..., (because) their experiments only talk about prediction accuracy of the transition function. It's totally possible to predict accurately enough an environment, but be very inaccurate at predicting a policy's performance in the environment. "
> > >
> > > "they take data from one domain (observations of a transition function under one policy) and try to make predictions in another domain (behavior of a transition function under another policy). This is confusing, since what one is actually interested in is off policy evaluation, knowing how well a an new policy will work based on observation of an old policy."
> >
> > Another misunderstanding of the reviewer is that the accuracy of the policy's value estimation is the one and only correct metric to evaluate the performance of an offline RL algorithm, so improving the accuracy of the learned transition function means nothing to increase the accuracy of the value estimation, thus the experiment cannot support the statement in this paper.
> >
> > However, the fact is that many existing studies in offline model-based RL have revealed the relationship between the error of transition prediction and the gaps in the policy's values [2,5]. Taking the result in [2] as an example, Lemma 4.1 and Eq. (4) tell us that:
> >
> > Given a true MDP with transition model $M^*$ and the learned dynamics model $M_\theta$ , for a policy $\beta$ , we have:
> >
> > $
> > |\eta_{M^*}(\beta)-\eta_{M_\theta}(\beta)| \le \frac{r_{\max}\gamma}{\sqrt{2}(1-\gamma)} \sqrt{\underbrace{\mathbb{E}_{(x,a)\sim \rho^{M^*}\_{\beta}} [D\_{\rm KL}(M^*(\cdot|x,a),M_\theta(\cdot|x,a))]}\_{\epsilon\_m} }
> > $
> >
> > where $\eta_M(\beta)$  is the expected discounted return of policy $\beta$ in the model $M$,  $r_{\rm max}$ is the maximal one-step reward, $|\eta_{M^*}(\beta)-\eta_{M_\theta}(\beta)|$ is the estimation gap of policy's values through the model $M_{\theta}$, and $\epsilon_m$ is also known as model error of $M_\theta$ under the data distribution $\rho^{M^*}_{\beta}$ collected by $\beta$.
> >
> > So it is natural to claim that we can generally reduce the value gaps $|\eta_{M^*}(\beta)-\eta_{M_\theta}(\beta)| $ by increasing the accuracy of the learned transition function, i.e., reducing $\epsilon_m$.
> >
> > Based on the above analysis, we also point out that the statement "It's totally possible to predict accurately enough an environment, but be very inaccurate at predicting a policy's performance in the environment" is unexact. If we have an accurate enough environment model, i.e., for any $\beta$,  $\epsilon_{m}$ is small enough, then we have, for any $\beta, |\eta_{M^*}(\beta)-\eta_{M_\theta}(\beta)|\approx 0$.
> >
> > In summary, the relationship between the prediction error of transition functions and the value gaps has been revealed in many works. *This is the reason why we focus on comparing the accuracy of transition functions queried by varied policies .*
> >
> > We also point out that the above inequality just gives an upper bound of the value estimation error.  If $\epsilon_m$ is not small enough on $\beta$, it could happen that the actual value estimation error increases when the model error decreases. Thus, verifying the performance improvement of the policy learned from the dynamics model is also a valuable and the most straightforward thing to do. We have added the related experiments to the revised version (refer to Sec. 5.2). The analysis of the results is also posted in the general response.

---

> > > ### Author Response · Authors · 2022-11-16
> > > **Official Response to Reviewer rQ79 (Part III)**
> > >
> > >
> > >
> > > **(3) relation between domain adaptation and our proposed AWRM.**
> > >
> > > > quote:
> > > >
> > > > "their method is more a Domain Adaptation method, where they take data from one domain (observations of a transition function under one policy) and try to make predictions in another domain (behavior of a transition function under another policy)"
> > > >
> > > > "To do this, they use a weighted empirical risk minimization framework, which is extended with an adversarial framework to learn a model under a worst-case assumption"
> > >
> > > Our method is neither a domain adaptation method nor a model learning algorithm under "a worst-case assumption."
> > >
> > > Domain adaptation methods usually make the model learned in the source domain have the ability to generalize to the target domain, where, in our setting, the source and target domains are the state-action distributions collected by the behavior policy $\mu$ and the target policy $\beta$ respectively. The weighted empirical risk minimization (WERM) based on the inverse propensity score (IPS) is a domain adaptation method in the selection bias scenario (refer to the blue lines in Appx. F for a detailed discussion).
> > >
> > > However, in this work, the primitive objective AWRM does not assume a worst-case target domain or other predefined domains for adaptation. Instead, AWRM uses the adversarial model learning framework to *iteratively* change the target distribution, which maximizes the model's prediction error, for model learning to achieve the generalization ability to unknown target policies $\beta \in \Pi$. In fact, it is more in line with domain generalization [6].
> > >
> > > Please note that our foremost contribution is the discovery that supervised learning of the model can be completely wrong. The proposed AWRM stands on the shoulders of giants, i.e., an adoption and modifications from WERM based on IPS. The modifications include considering sequential decision-making and the uncertain target policy to be evaluated, which are novel and were not presented in previous methods. We also do not think this could weaken our contribution.

---

> > > > ### Author Response · Authors · 2022-11-16
> > > > **Official Response to Reviewer rQ79 (Part Ⅳ)**
> > > >
> > > > ### Answer to the questions
> > > >
> > > > We sincerely hope the above clarifications can correct the misunderstandings of the reviewer about our study. In the following, we answer the reviewer's questions:
> > > >
> > > > **Q1: They don't compare to other state of the art domain adaptation methods, even though it's a domain adaption paper.**
> > > >
> > > > We indeed compared the state-of-the-art domain adaptation method in the synthetic tasks.
> > > >
> > > > As clarified in (3), WERM based on IPS is a special case of domain adaptation in the selection-bias scenario. In our experiment, the compared algorithm IPW is one of the popular implementations of WERM based on IPS.  We have given the related descriptions of the compared algorithms in the first paragraph of Sec. 5 (refer to the red lines).
> > > >
> > > > Besides, in the revised version, we also implemented IPW in MuJoCo tasks and the related results are in Tab. 1 (also refer to the general response).
> > > >
> > > >
> > > >
> > > > **Q2: Their experiments only talk about prediction accuracy of the transition function. & The authors claim their method's "policy significantly improves performance". In RL, the whole point is to improve a policies performance, and the authors can't demonstrate that their method indeed improves performance.**
> > > >
> > > > As discussed above, we have pointed out that focusing on comparing the accuracy of the transition function queried by varied policies is reasonable and the most important thing for demonstrating the statement of the study, based on the existing conclusion in previous works. We also agree that verifying the performance improvement of the policy learned from the dynamics model is a valuable and the most straightforward thing to do. We have added the related experiments in the revised version.  The analysis of the results is also posted in the general response.
> > > >
> > > >
> > > >
> > > > **Q3: As an experiment, the authors deploy their policy in a "real-world large-scale food-delivery platform", which is clearly not reproducible.**
> > > >
> > > > These experiments indeed cannot be open-sourced because of the commercial use of the code.
> > > >
> > > > In the revised version, we have given the details of this experiment in Sec. 5.2 and Appx. H.6 to explain why the problem is suitable for GALILEO to solve, and the important intermediate indicators to demonstrate the correctness of the learned model and the policies. Besides, we provide experiments in synthetic tasks and MuJoCo tasks using code the same as the code we used for model learning in the real-world environment, which will be open-sourced. And we have given a list of hyperparameters to show the core differences in Tab. 5.
> > > >
> > > > We hope the results help the reviewer verify the rationality of the experiments.
> > > >
> > > >
> > > > ### Reference
> > > >
> > > > [1] Sergey Levine, Aviral Kumar, George Tucker, Justin Fu: Offline Reinforcement Learning: Tutorial, Review, and Perspectives on Open Problems. CoRR abs/2005.01643 (2020).
> > > >
> > > > [2] Tianhe Yu, Garrett Thomas, Lantao Yu, Stefano Ermon, James Y. Zou, Sergey Levine, Chelsea Finn, Tengyu Ma: MOPO: Model-based Offline Policy Optimization. NeurIPS 2020.
> > > >
> > > > [3] Rahul Kidambi, Aravind Rajeswaran, Praneeth Netrapalli, Thorsten Joachims: MOReL: Model-Based Offline Reinforcement Learning. NeurIPS 2020.
> > > >
> > > > [4] Tianhe Yu, Aviral Kumar, Rafael Rafailov, Aravind Rajeswaran, Sergey Levine, Chelsea Finn: COMBO: Conservative Offline Model-Based Policy Optimization. NeurIPS 2021: 28954-28967
> > > >
> > > > [5] Tian Xu, Ziniu Li, Yang Yu: Error Bounds of Imitating Policies and Environments. NeurIPS 2020.
> > > >
> > > > [6] Jindong Wang, Cuiling Lan, Chang Liu, Yidong Ouyang, Tao Qin: Generalizing to Unseen Domains: A Survey on Domain Generalization. IJCAI 2021: 4627-4635.
> > > >
> > > > [7] Rafael Figueiredo Prudencio, Marcos R. O. A. Máximo, Esther Luna Colombini: A Survey on Offline Reinforcement Learning: Taxonomy, Review, and Open Problems. CoRR abs/2203.01387 (2022).
> > > >
> > > > [8] Fan-Ming Luo, Tian Xu, Hang Lai, Xiong-Hui Chen, Weinan Zhang, and Yang Yu: A survey on model-based reinforcement learning. CoRR abs/2206.09328 (2022).

---

### Official Review · Reviewer_j5p5 · 2022-10-24

**Confidence:** 4
**Correctness:** 4
**Technical Novelty And Significance:** 3
**Empirical Novelty And Significance:** 3
**Recommendation:** 6

**Clarity, Quality, Novelty And Reproducibility:**

*Clarity*: Great for the most part (detailed concerns above)

*Quality*: Very Good

*Novelty*: There are some minor concerns about significance since the experimental results barely touch on how the improved models contribute to better policies (and only for the "real world" example). The lack of discussion about relevant causal environment modeling papers is of some concern.

*Reproducibility*: There is an extensive section in the Appendix that outlines how the methods were implemented with corresponding pseudocode.


**Strength And Weaknesses:**

**Strengths**

This is a very complete and thoroughly written paper. The extensive efforts made by the authors to clearly justify the steps they took in the development of the proposed GALILEO is appreciated. I do wish there was a more detailed "related work" section with more concrete discussion about how the proposed approach builds or differs from prior work. However, this concern is mitigated mostly by how well cited the rest of the paper is. Each addition and concept throughout the paper is properly attributed.

I found the step-by-step derivation of GALILEO through Section 4 to be really easy to follow. The writing was very clear in the development of the final use of AWRM. This was capped off with a great discussion about how the theoretical components of the method are implemented (Section 4.3). Unfortunately, this level of detail (and accessibility) was not carried through the experiments (more on that below).

The extent of the experiments provide some clear signals that the proposed method carries some "general" significance outside of the simple synthetic domains many papers use as a proof of concept. It's clear that the models learned with GALILEO are more accurate than those trained with other approaches.

**Weaknesses**

To quickly jump to the empirical justification. It's disappointing that the improved models learned with GALILEO aren't shown to improve the overall task performance of the individual domains the models were learned to assist with. I can understand that this is not the major focus of the paper but the conceptual justification for needing generalizable models for MBRL is to provide more robust policy learning. As such the reported results feel only partial in nature. It would be great to see what the knock-on effects of an improved model are when learning a policy in the environment. Is there an improved nature of sample efficiency? How about the performance in more diverse test scenarios?

It's also disappointing that there aren't very thorough baseline comparisons to contemporary MBRL approaches in the accuracy of the learned models. Aside from the simple synthetic tasks, there are not really any baseline comparisons with "competing" approaches. I suppose that the SL baseline is better than nothing but the experiments overlook contributions made by the prior work in learning adequate models. Without a discussion about how these approaches devolve into SL, it's hard to place any significance on the experiments.

At times the discussion is cast in very formal language that distracts from the actual usage (or motivation) of the concepts. It's technically correct to cast "error" as "risk" (especially since the paper pursues "risk minimization") but it comes across as unnecessarily formal in places. Especially when MSE is used as a metric... This isn't a major concern but it does belie some aspects of where the writing in the paper could be improved to facilitate greater clarity.

The "real world" example is too vague in the initial sections of the paper. There is a lot of weight placed on the contributions of GALILEO working on a real domain but there's not sufficient framing or discussion about why this experiment works or is necessarily best solved by GALILEO (esp. when not compared to relevant baselines).

There are missing references to prior work using counterfactuals to improve environment modeling. In particular, I would interested to hear from the authors why they omit contributions from prior work such as Sontakke, et al (ICML 2021).

```Sontakke, S. A., Mehrjou, A., Itti, L., & Schölkopf, B. (2021, July). Causal curiosity: RL agents discovering self-supervised experiments for causal representation learning. In International Conference on Machine Learning (pp. 9848-9858). PMLR.```



**Summary Of The Paper:**

This paper introduces an approach for learning a model of the environment for model-based RL (MBRL) based on counterfactual data. This enables more stable model performance outside of the training distribution. The added generalization is driven by adversarial training to mitigate selection bias influencing the learned relationships between observations and subsequent actions.

**Summary Of The Review:**

This is a good paper and one that warrants publication. I however reserve my enthusiastic recommendation because there are points where the discussion could be improved, specifically around the empirical justifications used to validate the development of GALILEO.

---

> ### Author Response · Authors · 2022-11-16
> **Official Response to Reviewer j5p5 (Part I).**
>
> Thanks for your appreciation of our paper and your constructive suggestions for it. Our answers to the questions are as follows:
>
> **Q1:  I do wish there was a more detailed "related work" section with more concrete discussion about how the proposed approach builds or differs from prior work. However, this concern is mitigated mostly by how well cited the rest of the paper is. && There are missing references to prior work using counterfactuals to improve environment modeling [1].**
>
> In the revised version, we added a detailed related work section in Appx. F (refer to the first paragraph).
>
> The mentioned paper [1] belongs to another thread of applying causal inference techniques into RL (also refer to the second paragraph of Appx. F). In these studies, we consider that the transition function is relevant to some hidden noise variables, where the concept of causal factors in [1] is a special instance of the hidden noise variable. These studies focus on reconstructing the representation of the noise variable, or discovering and estimating the effects of the noise variables. [1] proposes a representation learning technique for causal factors in Partially Observable Markov Decision Processes (POMDPs). With the learned representation of causal factors, the performance of policy learning and transfer in downstream tasks will be improved.
>
> Generally speaking, our studies and the previous studies using IPS focused on handling the unbiased causal effect estimation problem of actions in the offline dataset under behavior policies collected with selection bias. In this branch of studies, we consider the environment model learning problem in the fully observed setting thus the hidden noise variable does not exist.
>
>
> [1] Sontakke, S. A., Mehrjou, A., Itti, L., & Schölkopf, B. (2021, July). Causal curiosity: RL agents discovering self-supervised experiments for causal representation learning. In International Conference on Machine Learning (pp. 9848-9858). PMLR.
>
>
> **Q2: It would be great to see what the knock-on effects of an improved model are when learning a policy in the environment. Aside from the simple synthetic tasks, there are not really any baseline comparisons with "competing" approaches.**
>
>
>
> We agree that more baselines in MuJoCo tasks and showing the knock-on effects will be better to demonstrate our statement. We have added the related experiments to the revised version (refer to Sec. 5.2). The analysis of the results is also posted in the general response.
>
>
>
>
>
> **Q4: The "real world" example is too vague in the initial sections of the paper. there's not sufficient framing or discussion about why this experiment works or is necessarily best solved by GALILEO.**
>
> Thanks for the important reminder. We found that the motivation for using GALILEO to solve BAT tasks is missing in the main body. We have added related information in the revised version.
>
> Specifically, the goal of the BAT task is to handle the imbalance problem between the demanded orders from customers and the supply of delivery clerks in different time periods by allocating reasonable allowances to those time periods.  The core challenge of the environment model learning in BAT tasks is similar to the challenge in Fig. 1 (in fact, this work began with this task).  Specifically, the behavior policy in BAT tasks is a human-expert policy with selection bias, which tends to increase the budget of allowance in the time periods with a lower supply of delivery clerks, otherwise tends to decrease the budget (Fig. 12 gives a  real-data instance of this phenomenon).
>
> We first learn a model to predict the supply of delivery clerks (measured by fulfilled order amount) on given allowances.  We found that, although the SL model can easily fit the offline data, the tendency of the response curve is easily to be incorrect. As can be seen in Fig. 5(a), with a larger budget of allowance, the prediction of the supply is decreased in SL, which obviously goes against our prior knowledge. This is because, in the offline dataset, the corresponding supply will be smaller when the allowance is larger. *It is conceivable that if we learn a policy through the model of SL, the derived optimal solution is canceling all of the allowances, which is obviously incorrect in practice.* On the other hand, the tendency of GALILEO's response is correct. Fig.13 plots all the results in 6 cities. Then we use the AUUC curve (Fig. 5(b)) to further demonstrate the correctness of the sort order of the model’s prediction in a randomized controlled trials (RCT) dataset. The above experiment results give us enough evidence and confidence to support the real-world performance of the policy learned by the GALILEO model. We have added the above description in Sec. 5.2 (refer to the blue lines).

---

> ### Author Response · Authors · 2022-11-16
> **Official Response to Reviewer j5p5 (Part II).**
>
>
>
> **Q5  At times the discussion is cast in a very formal language that distracts from the actual usage (or motivation) of the concepts. It's technically correct to cast "error" as "risk" (especially since the paper pursues "risk minimization") but it comes across as unnecessarily formal in places. Especially when MSE is used as a metric.**
>
> Thanks for the advice.  Risk minimization is the general objective, and we measure the "counterfactual risks" with mean square error (MSE) in different datasets in practice. We have clarified that in the revised paper.

---

### Official Review · Reviewer_bcJS · 2022-10-24

**Confidence:** 2
**Correctness:** 3
**Technical Novelty And Significance:** 3
**Empirical Novelty And Significance:** 3
**Recommendation:** 6

**Clarity, Quality, Novelty And Reproducibility:**

The paper is mostly clear. The idea and technical contributions look novel to me.

**Strength And Weaknesses:**

Strengths:

1. The main idea of the paper are well-motivated and interesting.
2. I am not sure how novel and original the relaxation of the adversarial objective is since I am not familiar with the related work. But the technical insights are sound.
3. The paper is in general well-written with much details well-organized in the appendix.
4. The experimental results are convincing in terms of showing the major messages.

Weaknesses:

1. It is not clear how the entire algorithm works and how many networks are learned in total. A high-level summarization or even an architecture figure would be helpful.
2. Many implementation details are omitted in the main paper. How is $H_{M^*}(x,a)$ implemented?
3. The presentation could be improved. The notations are sometimes not very clear. Are $\kappa$ and $\hat{\mu}$ (in appendix) the same? The notation $a$ is used several times for different meanings. Also, the layout is too tight. It seems that a negative vspace is used very aggresively.
4. The experiments only focus on the model learning performance in the form of prediction errors. However, it would be better to demonstrate how such a model can help model-based RL (learn a policy from the fitted model).

**Summary Of The Paper:**

This paper tackles the model learning problem in sequential decision making. Traditional methods may learn a model that fails under counterfactual samples (unseen in offline training dataset). To imporve the model's generalizability, the authors introduce adversarial weighted empirical risk minimization (AWRM) which learns a model with adversarially selected test-time policies. Although the original formulation of AWRM is intractable, the paper proposes a surrogate objective with several approximations. The effectiveness of the proposed method is verified by experiments in a synthetic environment, 3 MuJoCo environments, as well as a real-world food-delivery platform.

**Summary Of The Review:**

This paper addresses the generalization challenge of model learning in sequential decision making problems. The proposed method is reasonable and sound, with theoretical analysis and empirical evidence.

---

> ### Author Response · Authors · 2022-11-16
> **Official Response to Reviewer bcJS**
>
> Thanks for your appreciation of our paper and your constructive comments on it. We have fixed the mentioned typos and our answers to the questions are as follows:
>
> **Q1:  How the entire algorithm works and how many networks are learned in total.  A high-level summarization or even an architecture figure would be helpful.**
>
> Thanks for the reminder. We have given a workflow figure in Fig. 7.
>
>
>
> **Q2: How is $H_{M^∗}(x,a)$ implemented?**
>
>
>
>  In Eq.(8), $H_{M^*}$ is unknown in advance. In practice, we use $H_{M_\theta}$ to estimate it. More specifically, the neural network of $M_\theta $ can be modeled with a Gaussian distribution. The variance of the Gaussian distribution is modeled with global variables $\Sigma$ for each dimension of output. We estimate $H_{M^*}$ with the closed-form solution of Gaussian entropy through $\Sigma$. The related description is placed in Appx. E.1 in the previous version, which might be omitted by readers. We have moved them to Sec. 4.3 for better readability (see the blue lines in Sec. 4.3).
>
>
>
> **Q3: Are $\kappa$ and $\hat \mu$  (in appendix) the same?**
>
>
> $\hat \mu$ is a special case of $\kappa$.
>
> Without loss of  generality, we first introduce a general policy $\kappa$ in the process of tractable AWRM  solution derivation (see the red lines in Sec. 4.2) for better analyzing the properties of the derived objective. In Sec 4.3, we take $\hat μ$ as a practical implementation of $\kappa$  (see the red lines in Sec. 4.3).
>
>
>
>
>
> **Q4: the layout is too tight.**
>
> Thanks for the suggestion. We have removed some vspace commands for better readability.
>
>
>
> **Q5: it would be better to demonstrate how such a model can help model-based RL (learn a policy from the fitted model).**
>
>
>
> We agree that showing the knock-on effects will be better to demonstrate our statement. We have added the related experiments to the revised version (refer to Sec. 5.2). The analysis of the results is also posted in the general response.

---

### Author Response · Authors · 2022-11-16
**General Response**

To all reviewers:

We sincerely thank the reviewers for their appreciation of our paper and all their constructive suggestions, which really improve the paper. We first would like to emphasize the background of this paper, under which the contributions should be evaluated. Learning world models is essential in model-based RL. It was widely known that world models are hard to learn. Previously, this difficulty is mainly formulated into the compounding error, i.e., the model error will be accumulated when rolling out a long trajectory in the learned model [1,2,3]. Consequently, the state-of-the-art MBRL methods commonly employ short-length roll-outs [3,4,5]. However, the compounding error is not the only issue for model learning. *This paper discloses, for the first time in offline RL, that the supervised-learned model can lead to wrong learning directions, even if using short-length roll-outs*. *Note that the state-of-the-art MBRL methods all employ supervised learning techniques to learn models. This issue was completely missed in previous studies.*

Besides, we agree that more baselines in MuJoCo tasks and showing the knock-on effects will be better to demonstrate our statement. We have added the related experiments to the revised version.

In particular, we verify the generalization ability of the models by adopting them into offline model-based RL. Instead of designing sophisticated tricks to suppress policy exploration and learning in risky regions as current offline model-based RL algorithms [3,4,5] do, we just use the standard SAC algorithm [6] to exploit the models for policy learning to strictly verify the ability of the models. Unfortunately, we found that the compounding error will still be inevitably large in the 1,000-step rollout, which is the standard horizon in MuJoCo tasks, leading all models to fail to derive a reasonable policy. To better verify the effects of models on policy optimization, we learn and evaluate the policies with three smaller horizons: $H \in \\{10,20,40\\}$.

The results are listed in the below table (refer to the next comment).

We first averaged the normalized return (refer to the column ''avg. norm.'') under each task, and we can see that the policy obtained by GALILEO is significantly higher than other models (the improvements are 24\% to 161\%).

At the same time, we found that SCIGAN performed better in policy learning, while IPW performed similarly to SL. This is in line with our expectations, since IPW only considers the uniform policy as the target policy for adaptation, while policy optimization requires querying a wide variety of policies. Minimizing the prediction risks only on uniform policy cannot yield a good environment model for policy optimization. Besides, in IPW, the cumulative effects of policy on the state distribution are ignored. On the other hand, SCIGAN, as a partial implementation of GALILEO (refer to Appx. E.2), also roughly achieves AWRM and considers the cumulative effects of policy on the state distribution, so its overall performance is better.


In addition, we find that GALILEO achieves significant improvement in 6 of the 9 tasks. But in HalfCheetah, IPW works slightly better. However, compared with MAX-RETURN, it can be found that all methods fail to derive reasonable policies because their performances are far away from the optimal policy. By further visualizing the trajectories, we found that all the learned policies just keep the cheetah standing in the same place or even going backward. This phenomenon is also similar to the results in MOPO [3]: In MOPO's experiment in the "medium" datasets, the truncated-rollout horizon used in Walker and Hopper for policy training is set to 5, while HalfCheetah has to be set to *the minimal value: 1*.  These phenomena indicate that HalfCheetah may still have unknown problems, resulting in the generalization bottleneck of the models.



[1] Michael Janner, Justin Fu, Marvin Zhang, Sergey Levine: When to Trust Your Model: Model-Based Policy Optimization. NeurIPS 2019: 12498-12509.

[2] Tian Xu, Ziniu Li, Yang Yu: Error Bounds of Imitating Policies and Environments. NeurIPS 2020.

[3] Yu, Tianhe, et al. "MOPO: Model-based offline policy optimization." Advances in Neural Information Processing Systems 33 (2020): 14129-14142.

[4] Kidambi, Rahul, et al. "Morel: Model-based offline reinforcement learning." Advances in neural information processing systems 33 (2020): 21810-21823.

[5] Yu, Tianhe, et al. "COMBO: Conservative offline model-based policy optimization." Advances in neural information processing systems 34 (2021): 28954-28967.

[6] Haarnoja, Tuomas, et al. "Soft actor-critic algorithms and applications." arXiv preprint arXiv:1812.05905 (2018).

---

> ### Author Response · Authors · 2022-11-16
> **Results of policy performance**
>
>
> > Results of policy performance optimized through standard SAC using the learned dynamics models and deployed in MuJoCo environments. MAX-RETURN is the policy performance of SAC in the MuJoCo environments, and ``avg. norm.'' is the averaged normalized return of the policies in the 9 tasks, where the returns are normalized to lie between 0 and 100, where a score of 0 corresponds to the worst policy, and 100 corresponds to MAX-RETURN.
>
> | Task       | Hopper                  | Hopper                  | Hopper                  | Walker2d                | Walker2d                | Walker2d                | HalfCheetah             | HalfCheetah            | HalfCheetah              | avg. norm.    |
> |------------|-------------------------|-------------------------|-------------------------|-------------------------|-------------------------|-------------------------|-------------------------|------------------------|--------------------------|---------------|
> | Horizon    | H=10                    | H=20                    | H=40                    | H=10                    | H=20                    | H=40                    | H=10                    | H=20                   | H=40                     | /            |
> | GALILEO    | **13.0 $\pm$ 0.1** | **33.2 $\pm$ 0.1** | **53.5 $\pm$ 1.2** | **11.7 $\pm$ 0.2** | **29.9 $\pm$ 0.3** | **61.2 $\pm$ 3.4** | 0.7 $\pm$ 0.2           | -1.1 $\pm$ 0.2         | -14.2 $\pm$ 1.4          | **51.1** |
> | SL         | 4.8 $\pm$ 0.5           | 3.0 $\pm$ 0.2           | 4.6 $\pm$ 0.2           | 10.7 $\pm$ 0.2          | 20.1 $\pm$ 0.8          | 37.5 $\pm$ 6.7          | 0.4 $\pm$ 0.5           | -1.1 $\pm$ 0.6         | -13.2 $\pm$ 0.3          | 21.1          |
> | IPW        | 5.9 $\pm$ 0.7           | 4.1 $\pm$ 0.6           | 5.9 $\pm$ 0.2           | 4.7 $\pm$ 1.1           | 2.8 $\pm$ 3.9           | 14.5 $\pm$ 1.4          | **1.6 $\pm$ 0.2** | **0.5 $\pm$ 0.8** | **-11.3 $\pm$ 0.9** | 19.7          |
> | SCIGAN     | 12.7 $\pm$ 0.1          | 29.2 $\pm$ 0.6          | 46.2 $\pm$ 5.2          | 8.4 $\pm$ 0.5           | 9.1 $\pm$ 1.7           | 1.0 $\pm$ 5.8           | 1.2 $\pm$ 0.3           | -0.3 $\pm$ 1.0         | -11.4 $\pm$ 0.3          | 41.8          |
> | MAX-RETURN | 13.2 $\pm$ 0.0          | 33.3 $\pm$ 0.2          | 71.0 $\pm$ 0.5          | 14.9 $\pm$ 1.3          | 60.7 $\pm$ 11.1         | 221.1 $\pm$ 8.9         | 2.6 $\pm$ 0.1           | 13.3 $\pm$ 1.1         | 49.1 $\pm$ 2.3           | 100.0         |

---

> ### Comment · Reviewer_j5p5 · 2022-12-05
> **Thank you for the thorough response!**
>
> I wanted to briefly mention that I'm happy with the detailed and thorough response provided by the authors. The concerns I've raised in my review have sufficiently been addressed. I'm grateful for the additional effort to benchmark the proposed approach. It's clear that the proposed GALILEO algorithm furthers Model-based results in offline RL.
>
> Thanks for your patience and effort to improve the paper.

---

> > ### Author Response · Authors · 2022-12-06
> > **Thank you for the feedback**
> >
> > We are honored that our responses have addressed your concerns, and sincerely thank you for the constructive suggestions for improving our work.

---

### Decision · Program_Chairs · 2023-01-20

**Decision:**

Reject

**Justification For Why Not Higher Score:**

Clarity of exposition

**Justification For Why Not Lower Score:**

N/A

**Metareview: Summary, Strengths And Weaknesses:**

The authors propose a new way for learning models within the model-based RL framework. They propose an adversarially weighted IPS ERM framework, that tries to find models that perform well under multiple target distributions. Even though the paper offers a promising approach with good empirical performance, the initial submission offered weak experimental evidence and was not clear on the positioning of the paper. The authors improved their submission on both fronts, but still seems the paper would be more appropriate for a re-submission.